# Finite Sample Analyses for Continuous-time Linear Systems: System Identification and Online Control

## Abstract

Real world evolves in continuous time but computations are done from finite samples. Therefore, we study algorithms using finite observations in continuous-time linear dynamical systems. We first study the system identification problem, and propose a first non-asymptotic error analysis with finite observations. Our algorithm identifies system parameters without needing integrated observations over certain time intervals, making it more practical for real-world applications. Further we propose a lower bound result that shows our estimator is provably optimal up to constant factors. Moreover, we apply the above algorithm to online control regret analysis for continuous-time linear system. Our system identification method allows us to explore more efficiently, enabling the swift detection of ineffective policies. We achieve a regret of $\mathcal{O}(\sqrt{T})$ over a single $T$-time horizon in a controllable system, requiring only $\mathcal{O}(T)$ observations of the system.

## 1 Introduction

Finding optimal control policies requires accurately modelling the system (Kirk, 2004). However, real-world environments often involve unknown system parameters. In such cases, estimating unknown parameters from exploration becomes essential to identify the unseen dynamics. This process is recognized as system identification, a fundamental tool employed in various research fields, including time-series analysis (Korenberg, 1989), control theory (Kumar, 1983), robotics (Johansson et al., 2000), and reinforcement learning (Ross & Bagnell, 2012).

The identification of linear systems has long been studied because linear systems, as one of the most fundamental systems in both theoretical frameworks and practical applications, has wide applications ranging from natural physical processes to robotics. Most classical results provide only *asymptotic* convergence guarantees for parameter estimation (Åström & Eykhoff, 1971; Ljung, 1998b; Campi & Kumar, 1998b).

On the other hand, with the rapid increase in data scale, there is a growing concern for statistical efficiency. Consequently, the non-asymptotic convergence of *discrete-time* linear system identification has emerged as another pivotal topic in this field. Investigations into this matter delve into understanding how estimation confidence is influenced by the sample complexity of trajectories (Dean et al., 2018), or the running time on a single trajectory (Simchowitz et al., 2018; Sarkar & Rakhlin, 2019). Furthermore, many of these studies operate under the common assumption of stochastic noise, there has been a parallel exploration into the identification of discrete-time linear dynamical systems with diverse setups. This includes scenarios where perturbations are adversarial (Hazan et al., 2020) or when only black-box access is available (Chen & Hazan, 2021).

In contrast to studies in discrete time system, there have been relatively fewer non-asymptotic results addressing parameter identification for *continuous-time systems*. Two problems exist for continuous time analysis. First, nonasymptotic analysis in continuous system without noise can be degenerate, as a short time interval can contain infinite pieces of information. Second, if we consider the non-degenerate case when finite noisy observations are available, then the analyses require concentration results that become known only as in (Simchowitz et al., 2018; Dean et al., 2018; Sarkar & Rakhlin, 2019). Recently Basei et al. (2022) provides novel analyses for estimating system parameters, which

relies on continuous data collection and interaction with the environment. Motivated by progress in these works, our first goal is to answer the question below: *Can we design a continuous-time stochastic system identification algorithm that provides nonasymptotic error bounds with only a finite number of samples?*

We will introduce our system identification algorithms tailored to meet the above requirements. As expected, we discretize time into small intervals, thereby reducing the problem to a discrete system. The interesting part involves ensuring that the discretization remains bijective and that the inversion is unbiased. Our algorithm identifies the continuous system using only a finite number of samples from the discrete system. We further propose a information theoretic lower bound that shows our algorithm is optimal.

As an application of our system identification methods, we study an online continuous-time linear control problem as introduced in (Shirani Faradonbeh & Shirani Faradonbeh, 2023). In this context, exploration is essential for estimating unknown parameters, with the goal of identifying a more optimal control policy that narrows the performance gap. The primary challenge involves finding the right balance between exploration and exploitation. Leveraging our identification method for more efficient parameter estimation allows us to effectively manage exploration and exploitation, achieving an expected regret of $\mathcal{O}(\sqrt{T})$ over a single trajectory with only $\mathcal{O}(T)$ samples in time horizon $T$. This surpasses the previously best known result of $\mathcal{O}(\sqrt{T}\log(T))$, which needs continuous data collection from the system.

We summarize our contributions below.

1. When the system can be stabilized by a known controller, we establish an algorithm with $\mathcal{O}(T)$ samples that achieves estimation error $\mathcal{O}(\sqrt{1/T})$ on a single trajectory with running time $T$, which is shown in Theorem 1. We also provide Theorem 2 which shows that the estimation error of our system identification method is optimal up to constant factors.

2. When a stable controller is not available, we can use $N$ independent short trajectories to obtain estimators with error $\mathcal{O}(\sqrt{1/N})$, as is shown in Theorem 3 .

3. We apply our system identification method to an online continuous linear control algorithm, which only requires $\mathcal{O}(T)$ samples and achieves $\mathcal{O}(\sqrt{T})$ regret on a single trajectory with lasting time $T$ (Theorem 5), improving upon the best known result $\mathcal{O}(\sqrt{T}\log(T))$ in (Shirani Faradonbeh & Shirani Faradonbeh, 2023).

## 2 RELATED WORKS

Control of both discrete and continuous linear dynamical systems have been extensively studied in various settings, such as linear quadratic optimal control (Mehrmann, 1991), $H_2$ stochastic control (Dragan et al., 2004), $H_\infty$ robust control (Stengel, 1994; Khalil et al., 1996) and system identification (Kumar, 1983; Ljung, 1998b). Below we introduce some of the important results on both system identification and optimal control for linear dynamical systems.

**System Identification** Earlier literature focused primarily on the asymptotic convergence of system identification (Campi & Kumar, 1998a; Ljung, 1998a). Recently, there has been a resurgence of interest in non-asymptotic system identification for *discrete-time* systems. Dean et al. (2018) studied the sample complexity of multiple trajectories, with $\mathcal{O}(\sqrt{1/N})$ estimation error on $N$ independent trajectories. For systems with dynamics $x_{t+1} = Ax_t + w_t$(without controllers), Simchowitz et al. (2018) established an analysis for $\mathcal{O}(\sqrt{1/T})$ estimation error on a single stable trajectory with running time $T$, while Faradonbeh et al. (2018) and Sarkar & Rakhlin (2019) extended to more general discrete-time systems.

Non-asymptotic analyses for continuous-time linear system are less studied. Recently, Basei et al. (2022) examined continuous-time linear quadratic control systems with standard brown noise and unknown system dynamics. Our algorithm is specifically designed for finite observations, achieving an error rate that cannot be attained through the direct discretization of integrals as done in (Basei et al., 2022).

**Regret Analysis of Online Control** In online control, if the system's parameters are known, achieving the optimal control policy in this setup can be straightforward (Stengel, 1994; Yong & Zhou, 1999). However, when the system parameters are unknown, identifying the system incurs regret. Abbasi-Yadkori & Szepesvári (2011) achieved an $\mathcal{O}(\sqrt{T})$ regret for discrete-time online linear control, which has been proven optimal in $T$ under that setting in Simchowitz & Foster (2023). Subsequent works have extended this setup, focusing on worst-case analysis with adversarial noise and cost, including (Mania et al., 2019; Cohen et al., 2019; Lale et al., 2020; Simchowitz & Foster, 2023). These analyses are limited to discrete systems. For continuous-time systems, works of Shirani Faradonbeh et al. (2022); Shirani Faradonbeh & Shirani Faradonbeh (2023) established algorithms for online continuous control that achieves $\mathcal{O}\left(T^{1/2}\log(T)\right)$ regret.

## 3 PROBLEM SETUPS AND NOTATIONS

In this section, we introduce the background and notation for system identification in linear systems. We then discuss optimal control problems to motivate the setup for online control.

### 3.1 LINEAR DYNAMICAL SYSTEMS

We first introduce discrete-time linear dynamical systems as follows: Let $x_k \in \mathbb{R}^d$ represent the state of the system at time $k$, and let $u_k \in \mathbb{R}^p$ denote the action at time $k$. Then, for some linear time-invariant dynamics characterized by $A \in \mathbb{R}^{d \times d}$ and $B \in \mathbb{R}^{d \times p}$, the transition of the system to the next state can be represented as:

$$x_{k+1} = Ax_k + Bu_k + w_k, \tag{1}$$

where $w_k \in \mathbb{R}^d$ are i.i.d. Gaussian random vectors with zero means and certain covariance.

Similarly, a continuous-time linear dynamical system with stochastic disturbance at time $t$ is defined by a differential equation, instead of a recurrence relation:

$$dX_t = AX_t dt + BU_t dt + dW_t. \tag{2}$$

In this context, we use $X_t$ and $U_t$ to represent the state and action in the continuous-time linear system, distinguishing them from $x_t$ and $u_t$ in discrete-time systems. $W_t$ denotes the stochastic noise, which is modeled by standard Brownian motion.

For a continuous control problem, an important question of a linear dynamical system is whether such system can be stably controlled. Below we define the concepts of stable dynamics and stabilizers.

**Definition 1.** For any square matrix $A$, define $\alpha(A) = \max_i\{\Re(\lambda_i)|\lambda_i \in \lambda(A)\}$, where $\Re(\lambda)$ represents the real part of complex number $\lambda$, $\lambda(A)$ is the set of all eigenvalues of $A$.

**Definition 2.** A matrix $A \in \mathbb{R}^{d \times d}$ is stable if $\alpha(A) < 0$. A control matrix $K \in \mathbb{R}^{p \times d}$ is said to be a stabilizer for system $(A, B)$ if $A + BK$ is stable.

Under the above definition, a stable dynamic guarantees that the state can automatically go to the origin when no external forces are added, while applying a stabilizer as the dynamic for controller will also ensure that the state does not diverge.

### 3.2 CONTINUOUS-TIME LQR PROBLEMS AND OPTIMAL CONTROL

For continuous-time linear systems disturbed by stochastic noise, as introduced in 3.1, we denote the strategy of applying control to such systems through a specific causal policy, $f : X \to U$. This policy maps states $X$ to control inputs $U$, where the policy at time $t$ can only depend on the states and actions prior to $t$.

The optimal controls in linear systems are often linear (Stengel, 1994; Yong & Zhou, 1999), which takes the following form

$$U_t = K_t X_t,$$

where $K_t \in \mathbb{R}^{p \times d}$ represents the linear parameterization at time $t$ under some policy $f(X) = KX$. Additionally, we define the cost function of applying the action $U_t = K_t X_t$ with linear quadratic regulator (LQR) control. Given predefined symmetric positive definite matrices $Q \in \mathbb{R}^{d \times d}$ and

$R \in \mathbb{R}^{p \times p}$, along with the initial state $X_0$, the cost during $t \in [0, T]$ is denoted by $J_T$, as represented in the following equation:

$$J_T = \mathbb{E}\left[\int_{t=0}^{T} \left(X_t^{\mathrm{T}} Q X_t + U_t^{\mathrm{T}} R U_t\right) dt\right]. \tag{3}$$

Here the expectation is taken over the randomness of $X_t$ and the choice of linear dynamic $K_t$.

Among all the polices there exists an optimal mapping $f_*$ which minimizes $J_T$. When the system is dominated by dynamics $(A, B)$, with the state transits according to equation 2, such optimal $K_t$ can be computed via the Lyapunov matrix $P_t$ that solves the Ricatti differential equation (Yong & Zhou, 1999):

$$\frac{d}{dt} P_t = P_t B R^{-1} B^{\mathrm{T}} P_t - A^{\mathrm{T}} P_t - P_t A - Q, P_T = 0, \tag{4}$$

Then, under $f_*$ the action dynamic is set to be $K_t = -R^{-1} B^{\mathrm{T}} P_t$.

When $T \to +\infty$, the starting dynamic $P_0$ converges to some special dynamic $P_*$ satisfying

$$P_* B R^{-1} B^{\mathrm{T}} P_* - A^{\mathrm{T}} P_* - P_* A - Q = 0, \tag{5}$$

and the optimal control policy for infinite time horizon is by setting $K_t = -R^{-1} B^{\mathrm{T}} P_* := K_*$ and apply the action by $U_t = K_* X_t$.

**Online Control Problems.** Online learning aims to find a strategy to output a sequence of controls $\{U_t\}$ that minimizes the cost $J_T$ without knowing the system parameters $A, B$. In this scenario, online learning algorithms must explore to obtain valuable information, such as estimators $(\hat{A}, \hat{B})$ for $(A, B)$, while simultaneously exploit gathered information to avoid large instantaneous cost.

To quantify the progress in an online learning problem with horizon $T$, one quantity of interest is the regret $R_T$, which quantifies the performance gap between the control taken $U_t = f(X_t)$ and a baseline policy which takes $U_t = K_t X_t$, where $K_t$ is defined in equation 4. Formally, by denoting $J_T$ be the expected cost under $f$, and $J_T^*$ be the expected cost under the baseline policy, the regret $R_T$ is represented as:

$$R_T = J_T - J_T^*. \tag{6}$$

We have seen that in LQR problems with an infinite-time horizon, the optimal policy uses a time-invariant control dynamic $K_*$, expressed by $K_* = -R^{-1} B^{\mathrm{T}} P_*$ equation 5. For finite horizon, the dynamic $K_t$ equation 4 corresponds to the optimal policy converges exponentially fast in $T$ to the fixed dynamic $K_*$. Therefore, in this work, also following Shirani Faradonbeh & Shirani Faradonbeh (2023), for any horizon $T$, we adopt the same baseline policy, which performs the control by setting the following:

$$U_t = K_* X_t = -R^{-1} B^{\mathrm{T}} P_* X_t.$$

**Other Notations** Denote the d-dimensional unit sphere $\mathcal{S}^{d-1} = \{v \in \mathbb{R}^d, \|v\|_2 = 1\}$, where $\|\cdot\|_2$ is the $L_2$ norm. For any matrix $A \in \mathbb{R}^{m \times n}$, denote $\|A\|$ be the spectral norm of $A$, or equivalently,

$$\|A\| = \sup_{v \in \mathcal{S}^{n-1}} \|Av\|_2 = \sup_{u \in \mathcal{S}^{m-1}, v \in \mathcal{S}^{n-1}} u^{\mathrm{T}} A v.$$

## 4 THE PROPOSED SYSTEM IDENTIFICATION METHOD

In this part we propose our system identification method. Under finite time and samples, we develop a strategy to construct sets of states and actions which transit according to equation 1 a discrete update rule, with intermediate dynamics $(A', B')$ (see Algorithm 1 and Algorithm 2).

In particular, we can establish a one-to-one mapping between the constructed discrete system and the original continuous-time system as in equation 10 so that we can estimate $(A, B)$. This avoids computing on an integration, which is never achieved in previous works. We then bound the estimation errors in Theorem 1 and Theorem 3. The results are consistent with current best convergence rates of discrete-time linear system identification (Simchowitz et al. (2018), Dean et al. (2018)). Moreover, we present Theorem 2, which establishes the lower bound of the estimation error in this continuous-time system identification problem. It suggests our algorithm is optimal in sample complexity.

## 4.1 IDENTIFYING CONTINUOUS-TIME SYSTEMS WITH FINITE OBSERVATIONS

We highlight the finite-sample requirement in our analyses of system identification for continuous systems. Under this setting, not only the running time is finite, but the number of observed states is also limited. In other words, for a trajectory with lasting time $T$, we can only get access to a finite set of states $\{X_{t_1}, X_{t_2}, ..., X_{t_k}\}$ instead of a continuous uncountable collection of trajectory $\{X_t\}$, where $t$ is in some interval. One key difference is that quantities such as $\int_I \phi(X_t, U_t)dt$ with any function $\phi$ cannot be evaluated without error over an interval $I$. In addressing the system identification problem, we underscore a key distinction between our methodology and the traditional techniques employed for integral approximation.

In previous approaches, computations have relied on the approximation $X_{t+\epsilon} \approx (I + \epsilon A)X_t + \epsilon BU_t + (W_{t+\epsilon} - W_t)$. However, this approximation does not accurately reflect the true dynamics as expressed in equation 7, leading to a systematic error between the approximated and actual dynamics. This discrepancy, characterized by the error term $\epsilon^{-1}(e^{\epsilon A} - I) - A$, is of the order of $\epsilon$. As a result, to achieve the desired error bound, a super-linear sampling complexity relative to the total running time $T$ is required, which significantly increases computational demands. Our method addresses this issue by utilizing the bijection between domains of $S = \{X \big| \|X\| \le \frac{1}{4}\}$ and their matrix exponentials, thereby overcoming the limitations of direct discretization.

## 4.2 ALGORITHMS FOR CONTINUOUS SYSTEM IDENTIFICATION

With the transition of the state of a continuous system, represented in equation 2, when we take observations of state with sampling gap $h$, the states transit as in equation 7:

$$X_{t+h} = e^{Ah}X_t + \int_{s=0}^{h} e^{A(h-s)}BU_{t+s}ds + \int_{s=0}^{h} e^{A(h-s)}dW_{t+s}, \tag{7}$$

This transition equation connects continuous-time and discrete-time systems. However, the matrix exponential and integration make identifying system parameters from this relationship challenging. We address this challenge by first applying appropriate controls to simplify the analysis and then proposing a novel method to estimate $A$ from the matrix exponential, followed by recovering $B$ using the estimate of $A$.

In our method, the whole trajectory is partitioned into intervals with proper determined length $h$. During time $t \in [kh, (k+1)h]$, we observe a state $x_k$ at time $t = kh$, and fix the action $U_t \equiv u_k$ in this interval. Then the set of observations $\{x_k|k = 0, 1, 2, ...\}$ and actions $\{u_k|k = 0, 1, 2, ...\}$ has the following relation:

$$x_{k+1} = e^{Ah}x_k + \left[\int_{s=0}^{h} e^{A(h-s)}ds\right]Bu_k + w_k, \tag{8}$$

Here $w_k \sim \mathcal{N}(0, \Sigma_h)$ with $\Sigma_h = \int_{s=0}^{h} e^{As}e^{A^{\mathrm{T}}s}ds$ is a sequence of independent random noise. Denoting $A' = e^{Ah}$ and $B' = \left[\int_{s=0}^{h} e^{A(h-s)}ds\right]B$, the observed state transitions follow the standard discrete-time linear dynamical system:

$$x_{k+1} = A'x_k + B'u_k + w_k.$$

Next, we show how to identify $(A, B)$ from observations $\{x_t\}, \{u_t\}$ which follow the transition law in 8. Different from classical discrete-time systems, continuous-time systems present new challenges. The crucial one is that knowing $e^{Ah}$ is not sufficient to determine $A$, because the matrix exponential function $f(X) = e^X$ is not one-to-one. This means we might obtain an incorrect estimator $\hat{A}$ by solving $e^{\hat{A}h} = M$, where $M$ is the estimate of $e^{Ah}$.

The key to overcoming this challenge is the observation that when $\|X\| \le \frac{1}{4}$, the map $f(X) = e^X$ becomes one-to-one. Furthermore, if $\|X\| \le \frac{1}{6}$, we have $\|e^X - I_d\| \le \frac{1}{5}$, which allows us to find a $\bar{X}$ using Taylor expansion (see 10), with $\|\bar{X}\| \le \frac{1}{4}$ and $e^{\bar{X}} = e^X$. This $\bar{X}$ is exactly $X$ due to the one-to-one property of $f$ in the restricted domain. This insight enables direct analysis of the matrix exponential $e^{Ah}$ under the condition that $\|A\|h$ is small. The detailed proof is provided in Lemma 7.

---

**Algorithm 1** System identification algorithm for stable system

---

**Input:** Running time $T$, sample interval $h$ satisfying the condition in Assumption 1.
Define the number of samples $T_0 = \lceil T/h \rceil$.
**for** $k = 0, \ldots, T_0 - 1$ **do**
    Sample the action $u_k \overset{\text{i.i.d.}}{\sim} \mathcal{N}(0, I_p)$.
    Use the action $U_t \equiv u_k$ during the time period $t \in [kh, (k+1)h]$.
    Observe the new state $x_{k+1}$ at time $(k+1)h$.
**end for**
Compute $(\widetilde{A}, \widetilde{B})$ by equation 9, and then $(\bar{A}, \bar{B})$ by equation 10.
Let $(\hat{A}, \hat{B}) = (\bar{A}, \bar{B})$ be the estimates for system dynamics $(A, B)$.

---

With the above analysis, we can set the sampling interval $h \leq \frac{1}{6\|A\|}$ to ensure that $e^{Ah}$ can recover $A$. When Assumption 1 holds, we propose Algorithm 1, which identifies the system parameters from a single trajectory.

**Assumption 1** (Assumptions for Algorithm 1 and Theorem 1). We assume

1. The linear dynamic $A$ is stable, with $\alpha(A) < 0$ (see Definition 1). This is equivalent to assuming the existence of a stable controller $K$ and then set $A \leftarrow A + BK$.

2. $\|A\| \leq \kappa_A, \|B\| \leq \kappa_B$ for some known $\kappa_A, \kappa_B$ ($\kappa_A, \kappa_B$ need not be closed to $\|A\|, \|B\|$).

3. The sample interval $h$ is chosen to be $h = \frac{1}{15\kappa_A}$.

**Description of Algorithm 1** In the $k$-th interval with length $h$, a state $x_k$ is observed at the beginning, and a randomly selected action $u_k$ is uniformly performed during this interval. The state-action set $\{x_k, u_k\}$ is then utilized for estimating discretized $(A', B')$, as in equation 9.

$$(\widetilde{A})^{\mathrm{T}} = \left[\sum_{k=0}^{T_0-1} x_k x_k^{\mathrm{T}}\right]^{\dagger} \sum_{k=0}^{T_0-1} x_k x_{k+1}^{\mathrm{T}}, (\widetilde{B})^{\mathrm{T}} = \left[\sum_{k=0}^{T_0-1} u_k u_k^{\mathrm{T}}\right]^{\dagger} \sum_{k=0}^{T_0-1} u_k \left(x_{k+1} - \widetilde{A}x_k\right)^{\mathrm{T}}. \quad (9)$$

Next, the continuous-time dynamics $(A, B)$ are recovered through the estimates $(\widetilde{A}, \widetilde{B})$. Given that $\|A\|h$ is small, we can use the Taylor expansion to compute the logarithm of $\widetilde{A}$, denoted by $\bar{A}h$, which closely approximates $Ah$. The estimator $(\bar{A}, \bar{B})$ for $(A, B)$ is expressed as follows:

$$\bar{A} = \frac{1}{h} \sum_{k \geq 1} \frac{(-1)^{k-1}}{k} (\widetilde{A} - I)^k, \bar{B} = \left[\int_{t=0}^{h} e^{\bar{A}t} dt\right]^{-1} \widetilde{B}. \quad (10)$$

Algorithm 1 outlines the structured form of this entire procedure, achieving an $\mathcal{O}(T^{-1/2})$ estimation error of $(A, B)$ for a single trajectory with a duration of $T$ (see Theorem 1). An interesting thing is that Algorithm 1 can be generalized to the case where $A$ is not necessarily stable, but a stabilizer $K$ for $(A, B)$ (see Definition 2) is known. This generalization is applied in Algorithm 3 and will be discussed in Section 5. Finally, we note that the number of samples is linear in $T$.

**Summary of Notations** Below, we summarize several notations discussed.

1. Denote the ground truth $(A, B)$ as the **continuous-time** system dynamics.

2. Let $A' = e^{Ah}$ and $B' = \left[\int_{s=0}^{h} e^{A(h-s)} ds\right] B$ be the discretization of the ground truth $A, B$.

3. $(\widetilde{A}, \widetilde{B})$ refers to the estimates for $(A', B')$ from the observations, and are defined in equation 9.

4. Let $(\hat{A}, \hat{B})$ denote the algorithm output of the **continuous-time** system dynamics $(A, B)$. This output notation is used in Algorithm 1, 2, 3 as well as their corresponding theorems.

5. $(\bar{A}, \bar{B})$ refers to the estimates recovered from the discretization $(\widetilde{A}, \widetilde{B})$, defined in equation 10. In Algorithm 1, 2 it is just $(\hat{A}, \hat{B})$. In Algorithm 3 it is the estimates for $(A + BK, B)$.

Now we show that the above algorithm efficiently estimates the system dynamics. We first present the upper bound and postpone its proof sketch to the next subsection.

**Theorem 1** (Upper bound). *In Algorithm 1, there exists a constant $C \in poly\left(|\alpha(A)|^{-1}, \kappa_A, \kappa_B\right)$ such that, $\forall\, 0 < \delta < \frac{1}{2}$, when $T \geq C\left(\|X_0\|_2^2 + \log^2 1/\delta\right)$, with probability at least $1 - \delta$, we have:*

$$\|\hat{A} - A\|, \|\hat{B} - B\| \leq C\sqrt{\frac{\log(1/\delta)}{T}} \,. \tag{11}$$

Furthermore, our subsequent Theorem 2 establishes that this method has already attained the optimal convergence rate for parameter estimation. This theorem primarily asserts that, given a single trajectory lasting for time $T$, any algorithm that estimates system parameters solely based on *an arbitrarily large number of finite observed states* cannot guarantee an estimation error of $o(\sqrt{1/T})$

**Theorem 2** (Lower bound). *Suppose $T \geq 1$ be the running time of a single trajectory of continuous-time linear differential system, represented as in equation 2. Then there exist constants $c_1, c_2$ independent of $d$ such that, for any finite set of observed points $\{t_0 = 0, t_1, t_2, ..., t_N = T\}$, and any (possibly randomized) estimator function $\phi : \{X_{t_0}, X_{t_1}, ..., X_{t_N}\} \to \mathbb{R}^{d \times d}$, there exists bounded $A, B$ satisfying $\mathbb{P}\left[\|\phi(\{X_i\}_{i \leq N}) - A\| \geq \frac{c_1}{\sqrt{T}}\right] \geq c_2$. Here the probability is with respect to system noise.*

In Theorem 2, the mapping $\phi$ can refer to the output of any algorithm that exclusively relies on the finite set of states $X_{t_0}, X_{t_1}, ..., X_{t_N}$. *The interesting observation is that the lower bound does not decrease with a larger observation number $N$.*

The proof sketch of Theorem 2 is as follows: We consider two sets of dynamics, $(A, 0)$ and $(\bar{A}, 0)$, where both $A$ and $\bar{A}$ are stable, and $|A - \bar{A}| = \frac{2c_1}{\sqrt{T}}$. The challenge is ensuring the lower bound when the samples are uneven. Our key observation is that for the two distributions of observed states $S_k = \{X_{t_0}, X_{t_1}, ..., X_{t_k}\}$ and $\bar{S}_k = \{\bar{X}_{t_0}, X_{t_1}, ..., X_{t_k}\}$, where $X$ corresponds to the linear dynamic $A$ and $\bar{X}$ corresponds to $\bar{A}$, the KL divergence between $S_{k+1}$ and $\bar{S}_{k+1}$ increases by at most $\frac{c}{T}(t_{k+1} - t_k)$. Here, $c$ is a universal constant independent of $t_k$ and $t_{k+1}$. Thus, regardless of how the observation times are selected, the KL divergence between the observed states remains bounded.

### 4.3 FINDING AN INITIAL STABLE CONTROLLER

For general $(A, B)$, where a stabilizer is not known in advance, sticking to a single trajectory is not feasible as the state might diverge rapidly before obtaining a stable controller. We first list the assumptions on system parameters below.

**Assumption 2** (Assumptions for Algorithm 2 and Theorem 3). We assume

1. The constants $\kappa_A, \kappa_B, h$ follow the same assumptions as in 1.

2. The running time $T$ for each trajectory is small, say, $T = T_0 h$ where $T_0 \in \mathbb{N}$ and $T_0 \leq 10$.

Then, we employ multiple short trajectories to identify $A$ and $B$ as outlined in Algorithm 2. Similar to what is demonstrated in Dean et al. (2018), this procedure results in an $\mathcal{O}(H^{-1/2})$ estimation error on the trajectory number $H$ (Theorem 3).

**Theorem 3.** *In Algorithm 2, there exists a constant $C \in poly(\kappa_A, \kappa_B)$ such that w.p. at least $1 - \delta$, the estimation error of $(\hat{A}, \hat{B})$ from $H$ trajectories satisfies:*

$$\|\hat{A} - A\|, \|\hat{B} - B\| \leq C\sqrt{\frac{\log(1/\delta)}{H}} \,.$$

The proof is similar to that of Theorem 1, and details are shown in the Appendix. A stable controller can hence be designed from $\hat{A}, \hat{B}$.

### 4.4 ANALYSIS FOR THEOREM 1

In this section, we will primarily discuss the rationale behind the proof of our key theorems. Due to space limitations, detailed proofs of these theorems are provided in the appendix.

---

**Algorithm 2** Multi-trajectory system identification algorithm

---

**Input:** $T, T_0, h$ as in Assumption 2, number of trajectories $H$.
**for** $l = 1, \ldots, H$ **do**
    **for** $k = 0, \ldots, T_0 - 1$ **do**
        Sample the action $u_k^l \overset{\text{i.i.d.}}{\sim} \mathcal{N}(0, I_p)$, use the action $U_t \equiv u_k^l$ during $t \in [kh, (k+1)h]$.
        Observe the new state $x_{k+1}^l$ at time $(k+1)h$.
    **end for**
**end for**
Compute $(\widetilde{A}, \widetilde{B})$ by $(\widetilde{A}, \widetilde{B}) \in \arg\min_{(A,B)} \frac{1}{2} \sum_{l=1}^{H} \left\| x_{T_0}^l - A x_{T_0-1}^l - B u_{T_0-1}^l \right\|_2^2$.
Compute $\bar{A}, \bar{B}$ as in equation 10, let $(\hat{A}, \hat{B}) = (\bar{A}, \bar{B})$ be estimates for system dynamics $(A, B)$.

---

Our initial focus is on examining the error transformation from the discrete system to the original system. In Algorithm 1 and 2, discrete system identification methods are applied for estimating the intermediate dynamics $(A^{'}, B^{'})$. We will prove Lemma 4, which show that the errors of dynamics in the discrete system and the original system only differ by some constant factor, allowing us to only focus on discrete system identification problems:

**Lemma 4.** *In Algorithm 1, 2, suppose we have obtained the relative error $\|\widetilde{A} - A^{'}\|, \|\widetilde{B} - B^{'}\| \leq \epsilon$ for some $\epsilon \leq \frac{1}{15}$ and $\|Ah\| \leq \frac{1}{15}$, then we have the following relative error of the primal system:*

$$\|\hat{A} - A\|, \|\hat{B} - B\| \leq \frac{1}{h}\left(2 + \frac{\kappa_B}{\kappa_A}\right)\epsilon. \tag{12}$$

From this lemma, it becomes clear that if we develop a system identification algorithm for the discrete system that outputs the dynamics estimations $\widetilde{A}$ and $\widetilde{B}$ with minimal error, we can obtain the estimation of the primal system with relatively small error. Then it remains to develop the analysis for the discrete system with transition function $x_{k+1} = Ax_k + Bu_k + w_k$, which has actually been well discussed in previous works such as Simchowitz et al. (2018).

## 5 A Continuous Online Control Algorithm with Improved Regret

In this section, we apply our system identification method to a continuous LQR online control algorithm. Recall the setup introduced in Section 3.2 where we want to minimize the regret $R_T$ defined in equation 6. We will show in this section that with $\mathcal{O}(T)$ samples, our algorithm achieves $\mathcal{O}(\sqrt{T})$ expected regret on a single trajectory, thereby improving upon the previous $\mathcal{O}\left(\sqrt{T}\log(T)\right)$ result. We list the assumption for the online LQR problems below.

**Assumption 3** (Assumptions for Algorithm 3 and Theorem 5)**.** We assume that:

1. A stabilizer $K$ for $(A, B)$ (see Definition 2) with $\alpha(A + BK) < 0$ is known in advance.

2. Sample distance $h$ satisfies $h = \frac{1}{15\kappa}$, where $\kappa \geq \|A\| + \|B\|\|K\| \geq \|A + BK\|$ is known.

3. Denote $P_*$ be the solution in equation 5 and $K_* = -R^{-1}B^{\mathrm{T}}P_*$ be the baseline control dynamic.

4. $Q, R$ are positive-definite symmetric matrices with bounded spectral norms $\|Q\|, \|R\| \leq M$ and for some $\mu > 0$, $\mu I \preceq Q, \mu I \preceq R$.

### 5.1 An $\mathcal{O}(\sqrt{T})$ Regret Algorithm for Continuous Online Control

Our online continuous control algorithm is outlined in Algorithm 3, and we provide a detailed description below. Algorithm 3 is divided into two phases, exploration and exploitation. For the first exploration phase, a previously known stabilizer $K$ is applied to prevent the state from diverging. During the $k$-th interval, by setting $U_t = KX_t + u_k$, the state $X_t$ transits according to

$$dX_t = (A + BK)X_t dt + Bu_k dt + dW_t.$$

---

**Algorithm 3** Continuous online control algorithm

---

**Input:** $K, h$ which follows Assumption 3, running time $T$
**for** $k = 0, \ldots, [\frac{\sqrt{T}}{h}] - 1$ **do**
    Sample the action $u_k \overset{\text{i.i.d.}}{\sim} \mathcal{N}(0, I_p)$.
    For $t \in [kh, (k+1)h]$, set $U_t = KX_t + u_k$.
    Observe the new state $x_{k+1}$ at time $(k+1)h$.
**end for**
**Do system identification and estimate dynamics:**
Compute $(\widetilde{A}, \widetilde{B})$ according to equation 9 by using $\{x_k, u_k\}$.
Compute $\bar{A}, \bar{B}$ by equation 10 with $\widetilde{A}, \widetilde{B}$, and estimators $(\hat{A}, \hat{B})$ by $\hat{A} = \bar{A} - \bar{B}K, \hat{B} = \bar{B}$.
If $\hat{A}$ is stable, compute $P$ by equation 5 with estimated $\hat{A}, \hat{B}$, and set $\bar{K} = -R^{-1}(\hat{B})^{\mathrm{T}}P$.
If $\hat{A}$ is not stable or $P$ computed above satisfies $\|P\| \geq T^{\frac{1}{5}}$, then set $\bar{K} = K$.
**Perform exploitation:**
For $t \in [\sqrt{T}, T]$, set $U_t = \bar{K}X_t$.
**Detect bad policy and prevent the trajectory from diverging:**
If for some $t_0 \geq \sqrt{T}$, $\|X_{t_0}\| \geq T^{\frac{1}{5}}$, then set $U_t = KX_t$ for $t \in [t_0, T]$.

---

Since $A + BK$ is stable, through replacing $A$ in Theorem 1 by $A + BK$ in Algorithm 3, we can obtain a set of estimators $(\hat{A}, \hat{B})$ for $(A, B)$ with small error. This further allows us to accurately estimate $(A, B)$, thereby a controller $\bar{K} = -R^{-1}(\hat{B})^{\mathrm{T}}P$ closed to $K_*$ is obtained.

During exploitation phase, the near-optimal controller is deployed to minimize the cost, resulting in a regret of $\mathcal{O}(\sqrt{T})$ (see Theorem 5). However, as we lack direct feedback on whether $\bar{K}$ is a stabilizer, we need to detect its stability. Our approach involves replacing it with the known stabilizer $K$ whenever the state deviates too far.

**Theorem 5.** *Let $J_T$ be the expected LQR cost introduced in equation 3 that takes the action $U_t$ as in Algorithm 3. Then for some constant $C \in poly\left(\kappa, M, \mu^{-1}, |\alpha(A + BK)|^{-1}, |\alpha(A + BK_*)|^{-1}\right)$, the regret satisfies:*

$$R_T = J_T - J_T^* \leq C\sqrt{T}.$$

**Proof Sketch of Theorem 5** We analyze the two phases of our algorithm. During the exploration phase, the stabilizing controller $K$ effectively bounds the trajectory's radius, ensuring the average cost per unit time is within $\mathcal{O}(1)$, resulting in a total exploration cost of $C\sqrt{T}$.

In the subsequent exploitation phase, we analyze two scenarios separately. The first scenario occurs when the estimators $(\hat{A}, \hat{B})$ have large errors or when $\|X_t\|_2 \geq T^{1/5}$ for some $t \in [\sqrt{T}, T]$. This situation is rare and contributes a limited expected cost that can be bounded by a constant. The second scenario occurs when $(\hat{A}, \hat{B})$ are accurately estimated, and the control $U_t = -R^{-1}(\hat{B})^{\mathrm{T}}PX_t$ is applied throughout the exploitation phase. In this case, the trajectory's performance is straightforward to analyze, and the expected cost is bounded by $\mathcal{O}(\sqrt{T}) + J_T^*$.

By summing the expected costs, the total exploration cost is bounded by $\mathcal{O}(\sqrt{T})$, and the exploitation cost is bounded by $J_T^* + \mathcal{O}(\sqrt{T})$. By the definition of regret, $R_T = J_T - J_T^*$, the total regret is $\mathcal{O}(\sqrt{T})$, leading to the conclusion of Theorem 5.

## 5.2 COMPARISON AGAINST SHIRANI FARADONBEH & SHIRANI FARADONBEH (2023)

Our result is closely related to the result in Shirani Faradonbeh & Shirani Faradonbeh (2023), along with its similar version Faradonbeh (2022). They achieve $\mathcal{O}(\sqrt{T}\log(T))$ regret for both expectation and worst cases. We highlight some comparisons below. Both our and their setups assume a previously known stabilizer, which is also common for online control (Abeille & Lazaric, 2018; Ouyang et al., 2019; Faradonbeh et al., 2020; Shirani Faradonbeh & Shirani Faradonbeh, 2023). However, the work of Shirani Faradonbeh & Shirani Faradonbeh (2023) further assumes a known stabilization set for obtaining a stable controller, which is slightly stronger compared with ours. Such difference exists because our approach detects divergence and avoids sticking to a controller which is not stable.

More importantly, our system identification method is different. In Shirani Faradonbeh & Shirani Faradonbeh (2023), the exploration and exploitation is simultaneous, where a random matrix is added to the near-optimal controller so that both $A$ and $B$ can be identified. This additional noise cannot be too small, to ensure that the system can be well identified. This causes an extra $\log(T)$ factor to the regret. In contrast, our algorithm follows an explore-then-commit structure, which is enabled by the efficient system identification results presented previously.

Finally, we additionally considered the setup of finite observation, which is not discussed in Shirani Faradonbeh & Shirani Faradonbeh (2023).

### 5.3 EXPERIMENTS

In this section, we conduct simulation experiments for the baseline algorithm and our proposed algorithm. The baseline algorithm follows the work of Shirani Faradonbeh & Shirani Faradonbeh (2023). We set $d = p = 3$ for simplicity. Each element of $A$ is sampled uniformly from $[-1, 1]$, making $A$ unstable with high probability. The matrix $B$ is set as the identity matrix $I_3$. $Q$ and $R$ are also set as $I_3$. The sampling interval is set to $h = \frac{1}{30}$.

First, we run Algorithm 1 and Algorithm 2 for system identification. We plot the expected Frobenius norms of the error matrices $\|\hat{A} - A\|_F^2$ and $\|\hat{B} - B\|_F^2$. The results demonstrate that our algorithm can identify $A$ and $B$ within sufficient running time or number of trajectories.

Next, we compare Algorithm 3 with the baseline algorithm. We compute the average regret for different $t \in [600, 10000]$ and plot the results in Figure 1. We also analyze the normalized regret $R(T)/T^{1/2}$. The results show that our online control algorithm with system identification outperforms the baseline algorithm for sufficiently large $T$.

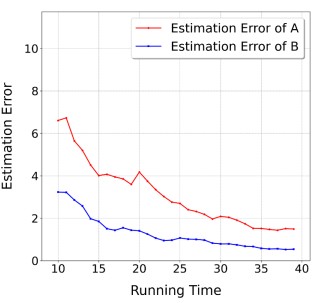 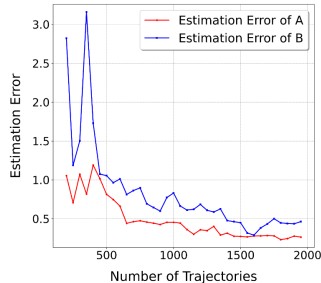 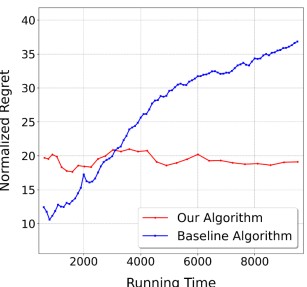

**Figure 1: The empirical validation of our algorithm.** Left: Identification of system dynamics using a single trajectory. Middle: Identification of system dynamics using multiple trajectories. Right: The normalized regret $R(T)/T^{1/2}$ of the baseline algorithm and our algorithm. The results show that our algorithm achieves small identification error within finite time and trajectories and is more efficient than the baseline algorithm.

## 6 CONCLUSIONS, LIMITATIONS AND FUTURE DIRECTIONS

In this work, we establish a novel system identification method for continuous-time linear dynamical systems. This method only uses a finite number of observations instead of requiring the integration of a consequent trajectory, and can be applied to an algorithm for online LQR continuous control which achieves $\mathcal{O}(\sqrt{T})$ regret on a single trajectory. Compared with existed works, our work not only eases the requirement for data collection and computation, but achieves fast convergence rate in identifying the unknown dynamics as well.

Although our method achieves near-optimal results in system identification and LQR online control for continuous systems with stochastic noise, many questions remain unsolved. First, it is unclear whether our system identification approach can be extended to more challenging setups, such as deterministic or adversarial noise. Additionally, many practical models are non-linear, raising the question of under what conditions discretization methods are effective. We believe these questions are crucial for real-world applications.

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

## A SYSTEM IDENTIFICATION FOR CONTINUOUS-TIME LINEAR SYSTEM

We begin by presenting the analysis for our system identification method in Algorithm 1 and Algorithm 2. As a preparation, we establish some properties of matrix exponentials and their inverses.

### A.1 MATRIX EXPONENTIAL

For a matrix exponential $e^{At}$, where the largest real component of $A$'s eigenvalues is denoted by $\alpha(A)$, the spectral norm of $e^{At}$ can be well-bounded (Golub & Van Loan, 2013), as demonstrated in Lemma 6.

**Lemma 6.** *Suppose an $n \times n$ matrix $A$ satisfies that $0 > \alpha(A) = \max\{\Re(\lambda_i)|\lambda_i \in \lambda(A)\}$. Let $Q^H A Q = \text{diag}(\lambda_i) + N$ be the Schur decomposition of $A$, and let $M_S(t) = \sum_{k=0}^{n-1} \frac{\|Nt\|_2^k}{k!}$. Then for $t > 0$, we have:*

$$\|e^{At}\| \le e^{\alpha(A)t M_s(t)}, \tag{13}$$

$$\frac{\|e^{(A+E)t} - e^{At}\|}{\|e^{At}\|} \le t\|E\|_2 (M_s(t))^2 e^{(tM_S(t)\|E\|_2)}. \tag{14}$$

In a special case where $\alpha(A) \le 0$, since $M_s(t) \ge 1$ for all $t$, we obtain

$$\|e^{At}\| \le e^{\alpha(A)t}.$$

We also show some properties of matrix inverse in the following Lemma 7.

**Lemma 7** (Matrix inverse). *For any $A \in \mathbb{R}^{d \times d}$ and $t$ such that $0 < \|At\| \le \frac{1}{10}$, we have the following estimation of $e^{At}$:*

$$\|e^{At} - I_d\| \le e^{\|At\|} - 1,$$

*and if we denote $A_1 = e^{At}$, then $A$ also satisfies that*

$$A = \frac{1}{t} \sum_{k \ge 1} \frac{(-1)^{k+1}}{k} (A_1 - I_d)^k.$$

*Proof.* We expand $e^{At}$ by

$$e^{At} = \sum_{k \ge 0} \frac{1}{k!} (At)^k,$$

which follows that

$$\|e^{At} - I_d\| = \left\| \sum_{k \ge 1} \frac{1}{k!} (At)^k \right\| \le \sum_{k \ge 1} \frac{1}{k!} \|At\|^k = e^{\|At\|} - 1 \le \frac{1}{9}.$$

Since $\|A_1 - I_d\| < 1$, the progression $A_2 = \sum_{k \ge 1} \frac{(-1)^{k+1}}{kt} (A_1 - I_d)^k$ converges, and thus $e^{A_2 t} = e^{At}$. Furthermore, it can be computed that

$$\|A_2 t\| \le \sum_{k \ge 1} \left\| \frac{1}{k} (A_1 - I_d) \right\| \le \sum_{k \ge 1} \frac{1}{k} (\frac{1}{9})^k \le \frac{1}{8}.$$

Now we show that $A_2 = A$. We have already known that $\|At\|$ and $\|A_2 t\|$ are small. We also note that the function $f : X \to e^X$ $(\|X\| \le \frac{1}{8})$ constitutes a one-to-one mapping. This assertion

is supported by the observation that for any $X_1, X_2$ such that $\|X_1\|, \|X_1 + X_2\| \leq \frac{1}{8}$, we have $\|X_2\| \leq \frac{1}{4}$, implying that

$$\left\|e^{X_1+X_2} - e^{X_1} - X_2\right\| = \left\|\sum_{k \geq 2} \frac{1}{k!}(X_1 + X_2)^k - X_1^k\right\| \tag{15}$$

$$\leq \sum_{k \geq 2} \frac{1}{k!} \frac{2^k - 1}{4^{k-1}} \|X_2\| \tag{16}$$

$$\leq \frac{1}{2}\|X_2\|. \tag{17}$$

Then $\left\|e^{X_1+X_2} - e^{X_1}\right\| \geq \frac{1}{2}\|X_2\|$, which means $f$ is one-to-one, and thereby leading that $A_2 = A$.

$\square$

## A.2 PROOF OF LEMMA 4

We restate Lemma 4 and provide the proof here.

**Lemma 4** In Algorithm 1, 2, suppose we have obtained the relative error $\|\widetilde{A} - A^{'}\|, \|\widetilde{B} - B^{'}\| \leq \epsilon$ for some $\epsilon \leq \frac{1}{15}$ and $\|Ah\| \leq \frac{1}{15}$, then we have the following relative error of the primal system:

$$\|\hat{A} - A\|, \|\hat{B} - B\| \leq \frac{C}{h}\epsilon, \tag{18}$$

where $C$ is a constant independent of $h$.

*Proof.* Firstly, according to Lemma 7, the estimated $\widetilde{A}$ is not too far away from $I_d$, as we have:

$$\left\|\widetilde{A} - I_d\right\| \leq \left\|\widetilde{A} - e^{Ah}\right\| + \left\|e^{Ah} - I_d\right\| \leq \epsilon + e^{\|A\|h} - 1 \leq \frac{1}{7},$$

Then, from equation 10 we can bound the matrix norm $\left\|\hat{A}h\right\|$ by

$$\left\|\hat{A}h\right\| = \left\|\sum_{k \geq 1} \frac{(-1)^{k-1}}{k}(\widetilde{A} - I)^k\right\| \leq \sum_{k \geq 1} \frac{1}{k}(\frac{1}{7})^k \leq \frac{1}{6}.$$

Now, let's denote $A_1 = Ah$ and $A_2 = \hat{A}h - A_1$, satisfying the relations $A^{'} = e^{A_1}$ and $\widetilde{A} = e^{A_1 + A_2}$. It is given that $\|A_1\| \leq \frac{1}{15}$ and $\|A_2\| \leq \|A_1\| + \|\hat{A}h\| \leq \frac{1}{4}$, so by equation 15, we obtain that $\|\hat{A} - A\|h = \|A_2\| \leq 2\|\widetilde{A} - A^{'}\|$, which follows that $\|\hat{A} - A\| \leq \frac{2}{h}\|\widetilde{A} - A^{'}\| \leq \frac{2}{h}\epsilon$. $\square$

Next, we will upper bound the estimation error of $B$. Let $A_h = \int_{t=0}^{h} e^{At}dt$ and $\bar{A}_h = \int_{t=0}^{h} e^{\hat{A}t}dt$, satisfying

$$\|A_h - hI\| = \left\|\int_{t=0}^{h} (e^{At} - I)dt\right\| \leq \int_{t=0}^{h} \|e^{At} - I\| dt \leq \int_{t=0}^{h} (e^{\|A\|t} - 1)dt \leq \frac{1}{20}h,$$

$$\|\bar{A}_h - A_h\| = \left\|\int_{t=0}^{h} e^{\hat{A}t} - e^{At}dt\right\| \leq \int_{t=0}^{h} \left\|e^{\hat{A}t} - e^{At}\right\| dt \leq \frac{3}{2}\int_{t=0}^{h} \|\hat{A} - A\|tdt \leq \frac{3}{4}h\epsilon.$$

This follows that

$$\|A_h^{-1}\| = \frac{1}{h}\left\|\left[I + (\frac{A_h}{h} - I)\right]^{-1}\right\| \leq \frac{1}{h}\sum_{k \geq 0}\left\|\frac{A_h}{h} - I\right\|^k \leq \frac{20}{19h},$$

$$\|(\bar{A}_h)^{-1} - A_h^{-1}\|$$

$$= \left\|A_h^{-1}\right\|\left\|\left[I + (\bar{A}_h - A_h)A_h^{-1}\right]^{-1} - I\right\| \leq \left\|A_h^{-1}\right\| \frac{1}{1 - \left\|(\bar{A}_h - A_h)A_h^{-1}\right\|} \leq \frac{1}{h}\epsilon.$$

Since $B$ and its estimator $\hat{B}$ satisfy that

$$B = (A_h)^{-1}B', \hat{B} = (\bar{A}_h)^{-1}\widetilde{B},$$

we can upper bound the estimation error $\left\|\hat{B} - B\right\|$ by

$$\left\|\hat{B} - B\right\| \leq \left\|(\bar{A}_h)^{-1} - A_h^{-1}\right\| \left\|B'\right\| + \left\|(\bar{A}_h)^{-1}\right\| \left\|\widetilde{B} - B'\right\| \leq \frac{\|B'\|}{h}\epsilon + \frac{2}{h}\epsilon \leq (2\|B\| + \frac{2}{h})\epsilon,$$

where the last inequality is because $\|B'\| \leq \|A_h\|\|B\| \leq 2h\|B\|$.

Since $2\|B\| \leq 2\kappa_B \leq \frac{1}{h} \cdot \frac{2\kappa_B}{15\kappa_A} \leq \frac{\kappa_B}{\kappa_A}$, we obtain Lemma 4.

### A.3 ANALYSIS FOR SYSTEM IDENTIFICATION WITH SINGLE TRAJECTORY

In this section, we upper bound the estimation errors of intermediate dynamics $(A', B')$, obtained as in equation 9. We primarily prove Lemma 8 below, providing system identification results on a single trajectory with a stable controller.

**Lemma 8.** *Consider the trajectory* $x_{k+1} = Ax_k + Bu_k + w_k$ *with* $A \in \mathbb{R}^{d \times d}$, $\|A\| < 1$, $B \in \mathbb{R}^{d \times p}$; $u_k \sim \mathcal{N}(0, I_p)$ *and* $w_k \sim \mathcal{N}(0, \Sigma)$ *are i.i.d. random variables. Suppose we compute* $(\hat{A}, \hat{B})$ *by*

$$(\hat{A})^{\mathrm{T}} = \left[\sum_{k=0}^{T_0-1} x_k x_k^{\mathrm{T}}\right]^{\dagger} \sum_{k=0}^{T_0-1} x_k x_{k+1}^{\mathrm{T}}, (\hat{B})^{\mathrm{T}} = \left[\sum_{k=0}^{T_0-1} u_k u_k^{\mathrm{T}}\right]^{\dagger} \sum_{k=0}^{T_0-1} u_k \left(x_{k+1} - \hat{A}x_k\right)^{\mathrm{T}}. \quad (19)$$

*Then there exists a constant* $C$ *(depending only on* $A$, $B$, $d$, $p$ *and* $\Sigma$*) such that for* $T \geq C\left(\|X_0\|_2^2 + \log^2(1/\delta)\right)$, *w.p. at least* $1 - \delta$:

$$\|\hat{A} - A\|, \|\hat{B} - B\| \leq C\sqrt{\frac{\log(1/\delta)}{T}}, \quad (20)$$

We first provide Lemma 9, which is used as the base of Lemma 8.

**Lemma 9.** *Consider* $A \in \mathbb{R}^{d \times d}$ *such that* $\rho(A) < 1$ *and the system* $X_{k+1} = AX_k + w_k$ *with* $w_k \sim \mathcal{N}(0, \Sigma)$ *be i.i.d. random variables. Suppose we estimate* $A$ *as in equation 9. Then there exists a constant* $C$ *depending on* $A$, $\Sigma$ *and* $d$ *such that for* $T \geq C(\|X_0\|_2^2 + \log(1/\delta))$, *w.p. at least* $1 - \delta$, *we have:*

$$\|\hat{A} - A\| \leq C\sqrt{\frac{\log(1/\delta)}{T}}.$$

The work of (Simchowitz et al., 2018) has discussed such systems in their Theorem 2.4, and we list it below:

**Theorem 10.** *Fix* $\epsilon, \delta \in (0, 1)$, $T \in \mathbb{N}$ *and* $0 \prec \Gamma_{sb} \prec \bar{\Gamma}$. *Then if* $(X_t, Y_t)_{t \geq 1} \in (\mathbb{R}^d \times \mathbb{R}^n)^T$ *is a random sequence such that (a)* $Y_t = A_*X_t + \eta_t$, *where* $\eta_t|\mathcal{F}_t$ *is* $\sigma^2$*-sub-Gaussian and mean zero, (b)* $X_1, ..., X_T$ *satisfies the* $(k, \Gamma_{sb}, p)$*-small ball condition, and (c) such that* $\mathbb{P}\left[\sum_{t=1}^{T} X_t X_t^{\mathrm{T}} \not\succeq T\bar{\Gamma}\right] \leq \delta$. *Then if*

$$T \geq \frac{10k}{p^2}\left(\log(1/\delta) + 2d\log(10/p) + \log\det(\bar{\Gamma}\Gamma_{sb}^{-1})\right),$$

*we have*

$$\mathbb{P}\left[\|\hat{A} - A_*\| > \frac{90\sigma}{p}\sqrt{\frac{n + d\log\frac{10}{p} + \log\det(\bar{\Gamma}\Gamma_{sb}^{-1}) + \log(\frac{1}{\delta})}{T\lambda_{\min}(\Gamma_{sb})}}\right] \leq 3\delta.$$

*Here, the* $(k, \Gamma_{sb}, p)$*-small ball condition is defined as follows. Let* $(Z_t)_{t \geq 1}$ *be an* $\mathcal{F}_{t t \geq 1}$*-adapted random process taking values in* $\mathbb{R}$. *We say* $(Z_t)_{t \geq 1}$ *satisfies the* $(k, \nu, p)$*-block martingale small-ball (BMSB) condition if, for any* $j \geq 0$, *one has* $\frac{1}{k}\sum_{i=1}^{k}\mathbb{P}(|Z_{j+i}| \geq \nu|\mathcal{F}_j) \geq p$ *almost surely. Given a process* $(X_t)_{t \geq 1}$ *taking values in* $\mathbb{R}^d$, *we say that it satisfies the* $(k, \Gamma_{sb}, p)$*-BMSB condition for* $\Gamma_{sb} \succ 0$ *if, for any fixed* $w \in \mathcal{S}^{d-1}$, *the process* $Z_t := \langle w, X_t \rangle$ *satisfies* $(k, \sqrt{w^{\mathrm{T}}\Gamma_{sb}w}, p)$*-BMSB.*

In the work of (Simchowitz et al., 2018), they have discussed the case when $X_0 = 0$, and now we modify it to a general starting state $X_0$. From equation 9, we derive the estimation error of $A$ as

$$\hat{A}^{\mathrm{T}} - A^{\mathrm{T}} = \left[\sum_{k=0}^{T-1} X_k X_k^{\mathrm{T}}\right]^{\dagger} \sum_{k=0}^{T-1} X_k X_{k+1}^{\mathrm{T}} - A^{\mathrm{T}}$$

$$= \left[\sum_{k=0}^{T-1} X_k X_k^{\mathrm{T}}\right]^{\dagger} \sum_{k=0}^{T-1} X_k (A X_k + w_k)^{\mathrm{T}} - A^{\mathrm{T}}$$

$$= \left[\sum_{k=0}^{T-1} X_k X_k^{\mathrm{T}}\right]^{\dagger} \sum_{k=0}^{T-1} X_k w_k^{\mathrm{T}} .$$

For the first term, consider any $v \in \mathcal{S}^{d-1}$, we lower bound $v^{\mathrm{T}} \left(\sum_{k=0}^{T-1} X_k X_k^{\mathrm{T}}\right) v$. Let $a_k = v^{\mathrm{T}} X_k$, then $a_k = v^{\mathrm{T}} A X_{k-1} + v^{\mathrm{T}} w_k$. We claim that for any $k \geq 1$, $\mathbb{P}\left[|a_k| \geq \frac{1}{2}|X_{k-1}\right] \geq \frac{1}{2}$. Let $b_k = v^{\mathrm{T}} w_k$, which is independent of $X_{k-1}$. It suffices to show that for any $c \in \mathbb{R}$, $\mathbb{P}\left[b_k \in [c, c+1]\right] \leq \frac{1}{2}$. Since $\|v\|_2 = 1$ and $w_k \sim \mathcal{N}(0, I_d)$, we have $b_k \sim \mathcal{N}(0, 1)$, from which we estimate the probability as

$$\mathbb{P}\left[b_k \in [c, c+1]\right] = \int_{x=c}^{c+1} \frac{1}{\sqrt{2\pi}} e^{-\frac{1}{2}x^2} dx \leq \frac{1}{\sqrt{2\pi}} \leq \frac{1}{2} . \tag{21}$$

Based on equation 21, we can simply choose $k = 1$, $\Gamma_{sb} = \frac{1}{4} I_d$ and $p = \frac{1}{2}$, then the random sequence $(X_i)_{i \geq 0}$ satisfies the $(k, \Gamma_{sb}, p)$-BMSB condition. It remains to choose a proper $\bar{\Gamma}$ that meets the condition (c) in Theorem 10.

Since $X_k = A^k X_0 + \sum_{i=1}^{k} A^{k-i} w_i$, we have:

$$\mathbb{E}\left[\sum_{k=0}^{T-1} X_k X_k^{\mathrm{T}}\right] = \mathbb{E}\left[\sum_{k=0}^{T-1} \left(A^k X_0 + \sum_{i=1}^{k} A^{k-i} w_i\right)\left(A^k X_0 + \sum_{i=1}^{k} A^{k-i} w_i\right)^{\mathrm{T}}\right]$$

$$= \sum_{k=0}^{T-1} A^k X_0 X_0^{\mathrm{T}} (A^k)^{\mathrm{T}} + \mathbb{E}\left[\sum_{k=0}^{T-1} \sum_{i=0}^{k} A^k X_0 w_i^{\mathrm{T}} (A^{k-i})^{\mathrm{T}}\right]$$

$$+ \mathbb{E}\left[\sum_{k=0}^{T-1} \sum_{i=0}^{k} A^{k-i} w_i X_0^{\mathrm{T}} (A^k)^{\mathrm{T}}\right] + \mathbb{E}\left[\sum_{k=0}^{T-1} \sum_{i,j=0}^{k} A^{k-i} w_i w_j^{\mathrm{T}} (A^{k-j})^{\mathrm{T}}\right]$$

$$= \sum_{k=0}^{T-1} A^k X_0 X_0^{\mathrm{T}} (A^k)^{\mathrm{T}} + \sum_{k=0}^{T-1} \sum_{i=0}^{k} A^{k-i} \Sigma (A^{k-i})^{\mathrm{T}} .$$

Let $\Gamma_{\infty} = \sum_{k \geq 0} A^k \Sigma (A^k)^{\mathrm{T}}$ which is bounded and $C_1$ be a constant such that $C_1 \geq \sum_{k \geq 0} \|A^k\|^2$. We then show that for $\bar{\Gamma} = \left(\frac{C_1 \|X_0\|_2^2}{T} d I_d + d\|\Gamma_{\infty}\| I_d\right)/\delta$, the condition (c) in Theorem 10 is satisfied. This is because $\mathbb{E}\left[\mathrm{tr}\left(\sum_{k=0}^{T-1} X_k X_k^{\mathrm{T}}\right)\right] = \mathrm{tr}\left(\mathbb{E}\left[\sum_{k=0}^{T-1} X_k X_k^{\mathrm{T}}\right]\right) \leq \frac{T\delta}{d} \mathrm{tr}(\bar{\Gamma})$ so that $\mathbb{P}\left[\mathrm{tr}(\sum_{k=0}^{T-1} X_k X_k^{\mathrm{T}}) \geq \frac{1}{d} T \mathrm{tr}(\bar{\Gamma})\right] \leq \delta$. Furthermore, a necessary condition for $\sum_{k=0}^{T-1} X_k X^{\mathrm{T}} \nprec T\bar{\Gamma}$ is $\mathrm{tr}(\sum_{k=0}^{T-1} X_k X^{\mathrm{T}}) \geq \frac{1}{d} T \mathrm{tr}(\bar{\Gamma})$.

Now, we apply such $\bar{\Gamma}$ to Theorem 10. It can be computed that
$$\log \det(\bar{\Gamma} \Gamma_{sb}^{-1}) = d \log \left(4d(C_1 \|X_0\|_2^2/T + \|\Gamma_{\infty}\|)\right) + d \log(1/\delta) .$$
Then when $T \geq C_1 \|X_0\|^2$ as well as $T \geq 40 \left(2d \log(20) + d \log(4d(1 + \|\Gamma_{\infty}\|)) + 2d \log(1/\delta)\right)$, we have:

$$\mathbb{P}\left[\|\hat{A} - A\| > 360\sqrt{\frac{d + d \log(20) + d \log(4d(1 + \|\Gamma_{\infty}\|)) + 2d \log(\frac{1}{\delta})}{T}}\right] \leq 3\delta .$$

This implies our Lemma 9.

**Proof of Lemma 8** As for the estimation error $\|\hat{A} - A\|$, let $w_k^{'} = Bu_k + w_k \sim \mathcal{N}(0, \Sigma + BB^{\mathrm{T}})$, which form a sequence of $i.i.d$ random variables. With the results in Lemma 9, there exist some constants $C_1, C_2$ such that, as long as $T \geq C_1 \left(\|X_0\|_2^2 + \log(1/\delta)\right)$ we have:

$$\|\hat{A} - A\| \leq C_2 \sqrt{\frac{\log(1/\delta)}{T}} \,.$$

Now we upper bound the estimation error $\|\hat{B} - B\|$. With the expression in equation 9, we obtain:

$$\|\hat{B} - B\| = \left\| \left[ \sum_{k=0}^{T-1} u_k u_k^{\mathrm{T}} \right]^{\dagger} \sum_{k=0}^{T-1} u_k \left[ (A - \hat{A})X_k + w_k \right]^{\mathrm{T}} \right\|$$

$$\leq \lambda_{\min}^{-1}(\sum_{k=0}^{T-1} u_k u_k^{\mathrm{T}}) \left[ \left\| \sum_{k=0}^{T-1} u_k X_k^{\mathrm{T}} \right\| \left\| \hat{A} - A \right\| + \left\| \sum_{k=0}^{T-1} u_k w_k^{\mathrm{T}} \right\| \right] \,.$$

For the quantities $\lambda_{\min}^{-1}(\sum_{k=0}^{T-1} u_k u_k^{\mathrm{T}})$ and $\|\sum_{k=0}^{T-1} u_k w_k^{\mathrm{T}}\|$, we apply Lemma 2.1. and Lemma 2.2. in the work of (Dean et al., 2018), where they present the following results.

**Lemma 11.** *Let $N \geq 2 \log(1/\delta)$. Suppose $f_k \in \mathbb{R}^m$, $g_k \in \mathbb{R}^n$ are independent vectors such that $f_k \sim \mathcal{N}(0, \Sigma_f)$ and $g_k \sim \mathcal{N}(0, \Sigma_g)$ for $1 \leq k \leq N$. With probability at least $1 - \delta$,*

$$\left\| \sum_{k=1}^{N} f_k g_k^{\mathrm{T}} \right\| \leq 4 \|\Sigma_f\|_2^{1/2} \|\Sigma_g\|_2^{1/2} \sqrt{N(m+n) \log(9/\delta)} \,.$$

**Lemma 12.** *Let $X \in \mathbb{R}^{N \times n}$ have i.i.d. $\mathcal{N}(0, 1)$ entries. With probability at least $1 - \delta$,*

$$\sqrt{\lambda_{\min}(X^{\mathrm{T}}X)} \geq \sqrt{N} - \sqrt{n} - \sqrt{2 \log(1/\delta)} \,.$$

With these two lemmas, we can conclude that if $T \geq 32(d + p) \log(4/\delta)$, then both $\lambda_{\min}(u_k u_k^{\mathrm{T}}) \geq \frac{1}{2}T$ and $\left\| \sum_{k=0}^{T-1} u_k w_k^{\mathrm{T}} \right\| \leq 4 \|\Sigma\|_2^{1/2} \sqrt{T(d+p) \log(18/\delta)}$, w.p. at least $1 - \delta$.

Now we concentrate on the term $\left\| \sum_{k=0}^{T-1} u_k X_k^{\mathrm{T}} \right\|$. Since $w_i^{'} = Bu_i + w_i \sim \mathcal{N}(0, \Sigma + BB^{\mathrm{T}})$, it can be directly computed that, w.p. at least $1 - \delta/T$, $\left\| w_i^{'} \right\|_2 \leq 2 \left\| d(\Sigma + BB^{\mathrm{T}}) \right\|_2^{1/2} \sqrt{\log(T/\delta)}$. Then by union bound we get $\mathbb{P} \left[ \sup_{0 \leq i \leq T-1} \left\| w_i^{'} \right\|_2 \leq 2 \|\Sigma + BB^{\mathrm{T}}\|_2^{1/2} \sqrt{d \log(T/\delta)} \right] \leq \delta$. Furthermore, when $\sup_{0 \leq i \leq T-1} \left\| w_i^{'} \right\|_2 \leq 2 \left\| \Sigma + BB^{\mathrm{T}} \right\|_2^{1/2} \sqrt{d \log(T/\delta)}$, we must have

$$\|X_k\|_2 = \left\| A^k X_0 + \sum_{i=0}^{k-1} A^{k-1-i} w_i \right\| \leq \|A\|^k \|X_0\|_2 + \frac{2}{1 - \|A\|} \left\| \Sigma + BB^{\mathrm{T}} \right\|_2^{1/2} \sqrt{d \log(T/\delta)} \,. \tag{22}$$

For any $u \in \mathcal{S}^{p-1}$ and $v \in \mathcal{S}^{d-1}$, let $x_i = u^{\mathrm{T}} u_i (0 \leq i \leq T - 1)$. Then, $x_i$ follows a normal distribution $x_i \sim \mathcal{N}(0, 1)$ and $\{x_i\}$ is a sequence of independent random variables. Furthermore, $x_k$ is also independent of $(X_i)_{0 \leq i \leq k}$. On the other hand, denote $y_k = X_k^{\mathrm{T}} v$, equation 22 implies that w.p. at least $1 - \delta$, for all $k$ we have $|y_k| \leq \|X_0\|_2 + \frac{2}{1 - \|A\|} \left\| \Sigma + BB^{\mathrm{T}} \right\|_2^{1/2} \sqrt{d \log(T/\delta)} := Y$. Let

$$Z_k := \sum_{i=0}^{k} u^{\mathrm{T}} \left( u_k X_k^{\mathrm{T}} \right) v \cdot 1_{\|X_k\|_2 \leq Y} = \sum_{i=0}^{k} x_k y_k \cdot 1_{\|X_k\|_2 \leq Y} \,,$$

and let $\mathcal{F}_0, \mathcal{F}_1, ..., \mathcal{F}_T$ be the filtration of $X_0, X_1, ..., X_T$, then for any $\alpha \geq 0$,

$$\mathbb{E} \left[ e^{\frac{\alpha Z_{k+1}}{Y}} | \mathcal{F}_k \right] = e^{\frac{\alpha Z_k}{Y}} \mathbb{E}_{X_{k+1}} \left[ \mathbb{E}_{x \sim \mathcal{N}(0,1)} \left[ \exp \left( \frac{\alpha x y_{k+1} \cdot 1_{\|X_{k+1}\|_2 \leq Y}}{Y} \right) \right] \right] \leq e^{\frac{1}{2}\alpha^2} e^{\frac{\alpha Z_k}{Y}} \,,$$

implying that $\mathbb{E}\left[e^{\frac{\alpha Z_{k+1}}{Y}}\right] \leq e^{\frac{1}{2}\alpha^2}\mathbb{E}\left[e^{\frac{\alpha Z_{k+1}}{Y}}\right]$ So we have: $\mathbb{E}\left[e^{\frac{\alpha Z_{T-1}}{Y}}\right] \leq e^{\frac{1}{2}\alpha^2 T}$. By choosing $\alpha = \pm\sqrt{\frac{1}{T}}$, we obtain that

$$\mathbb{P}\left[|Z_{T-1}| \geq 2Y\sqrt{T\log(4/\delta)}\right] \leq \delta$$

For $\mathcal{T}_d$ be a $\frac{1}{4}$-net of $\mathcal{S}^{d-1}$ and $\mathcal{T}_p$ be a $\frac{1}{4}$-net of $\mathcal{S}^{p-1}$, we use union bound on them and obtain that, w.p. at least $1 - \delta$

$$|Z_{T-1}| \leq 2Y\sqrt{T\log(4|\mathcal{T}_p||\mathcal{T}_d|/\delta)} \leq 2Y\sqrt{T[4(d+p) + \log(4/\delta)]}.$$

Where the last inequality is because $|\mathcal{T}_p| \leq 9^p$ and $|\mathcal{T}_d| \leq 9^d$

Next we upper bound $\left\|\sum_{k=0}^{T-1} u_k X_k^{\mathrm{T}}\right\|$. For any $u_* \in \mathcal{S}^{p-1}$ and $v_* \in \mathcal{S}^{p-1}$, with some $u \in \mathcal{T}_p$ and $v \in \mathcal{T}_d$ s.t. $\|u - u_*\|_2, \|v - v_*\|_2 \leq \frac{1}{2}$, we have:

$$\left| u_*^{\mathrm{T}}(\sum_{k=0}^{T-1} u_k X_k^{\mathrm{T}})v_* \right|$$

$$\leq \left| u^{\mathrm{T}}(\sum_{k=0}^{T-1} u_k X_k^{\mathrm{T}})v \right| + \left| (u_* - u)^{\mathrm{T}}(\sum_{k=0}^{T-1} u_k X_k^{\mathrm{T}})v_* \right| + \left| u^{\mathrm{T}}(\sum_{k=0}^{T-1} u_k X_k^{\mathrm{T}})(v - v_*) \right|$$

$$\leq \sup_{u \in \mathcal{T}_p, v \in \mathcal{T}_d} \left| u^{\mathrm{T}}(\sum_{k=0}^{T-1} u_k X_k^{\mathrm{T}})v \right| + \frac{1}{2}\left\|\sum_{k=0}^{T-1} u_k X_k^{\mathrm{T}}\right\|.$$

This leads $\left\|\sum_{k=0}^{T-1} u_k X_k^{\mathrm{T}}\right\| \leq 2\sup_{u \in \mathcal{T}_p, v \in \mathcal{T}_d}\left| u^{\mathrm{T}}(\sum_{k=0}^{T-1} u_k X_k^{\mathrm{T}})v \right|$. Therefore, for any $\delta \in (0, \frac{1}{2})$, we have:

$$\mathbb{P}\left[\left\|\sum_{k=0}^{T-1} u_k X_k^{\mathrm{T}}\right\|_2 \geq 4Y\sqrt{T[4(d+p) + \log(4/\delta)]}\right]$$

$$\leq \mathbb{P}\left[\sup_{u \in \mathcal{T}_d, v \in \mathcal{T}_p}\left| u^{\mathrm{T}}\left(\sum_{k=0}^{T-1} u_k X_k^{\mathrm{T}}\mathbf{1}_{\|X_k\|_2 \leq Y}\right)v \right| \geq 2Y\sqrt{T[4(d+p) + \log(4/\delta)]}\right]$$

$$+ \mathbb{P}\left[\exists\, 0 \leq k \leq T - 1, \|X_k\|_2 \geq Y\right]$$

$$\leq 2\delta.$$

We choose constant $C$ depending on $A, B, d, p$ such that for all $T \geq C\left(\|X_0\|_2^2 + \log^2(1/\delta)\right)$,

$$4Y\sqrt{T[4(d+p) + \log(4/\delta)]} \leq T,$$

and we further have: whenever $T \geq C\left(\|X_0\|_2^2 + \log^2(1/\delta)\right)$, w.p. at least $1 - 3\delta$,

$$\left\|\sum_{k=0}^{T-1} u_k X_k^{\mathrm{T}}\right\|\|\hat{A} - A\| \leq C_2\sqrt{\log(1/\delta)T}.$$

Finally, when $T \geq \max\left(C\left(\|X_0\|_2^2 + \log^2(1/\delta)\right), 32(d+p)\log(4/\delta)\right)$, we combine this upper bound with $\mathbb{P}\left(\lambda_{\min}(\sum_{k=0}^{T-1} u_k u_k^{\mathrm{T}}) \leq \frac{1}{2}T\right) \leq \delta$, and obtain Lemma 8.

## A.4 SYSTEM IDENTIFICATION WITH MULTIPLE TRAJECTORIES

Now, we aim to establish Theorem 3. The analysis of system identification for discrete-time linear dynamical systems with multiple trajectories has been thoroughly investigated by (Dean et al., 2018). We hereby cite their findings, denoting the relevant result as Lemma 13.

**Lemma 13.** *Suppose we have $N$ i.i.d. trajectories $X_k^i$, each is defined by $X_{(k+1)h}^i = AX_k^i + Bu_k^i + w_k^i$, where $T_0$ is any integer, $u_k^i \sim \mathcal{N}(0, I_p)$ and $w_k^i \sim \mathcal{N}(0, \Sigma)$ are two sets of i.i.d. random variables. Then, for the estimator $(\hat{A}, \hat{B})$ of*

$$(\hat{A}, \hat{B}) \in \arg \min_{(A,B)} \frac{1}{2} \sum_{i=1}^{N} \left\| X_{T_0}^i - AX_{T_0-1}^i - Bu_{T_0-1}^i \right\|_2^2 \tag{23}$$

*with probability at least $1 - \delta$, we have:*

$$\|\hat{A} - A\|, \|\hat{B} - B\| \leq \mathcal{O}\left(\sqrt{\frac{\log(1/\delta)}{N}}\right).$$

Combining Lemma 13 with Lemma 4, we directly obtain Theorem 3.

### A.5 LOWER BOUND OF SYSTEM IDENTIFICATION WITH FINITE OBSERVATION

We restate and provide the proof of Theorem 2.

**Theorem 2** Suppose $T \geq 1$ be the running time of a single trajectory of continuous-time linear differential system, represented as in equation 2. Then there exist constants $c_1, c_2$ independent of $d$ such that, for any finite set of observed points $\{t_0 = 0, t_1, t_2, ..., t_N = T\}$, and any (possibly randomized) estimator function $\phi : \{X_{t_0}, X_{t_1}, ..., X_{t_N}\} \to \mathbb{R}^{d \times d}$, there exists bounded $A, B$ satisfying $\mathbb{P}\left[\|\phi(\{X_i\}_{i \leq N}) - A\| \geq \frac{c_1}{\sqrt{T}}\right] \geq c_2$. Here the probability corresponds to the dynamical system dominated by $(A, B)$.

*Proof.* Firstly, we consider a special case where $d = 1$, and let $A = [-1]$ and $\bar{A} = [-1 - \delta]$. We show that when $\delta = \frac{1}{5\sqrt{T}}$, for the two dynamical systems $\psi_\theta : dX_t = AX_tdt + dW_t$ and $\psi_{\bar{\theta}} : dX_t = \bar{A}X_tdt + dW_t$, any algorithm $\mathcal{A}$ that outputs according only to $\{X_{t_0}, X_{t_1}, ..., X_{t_N}\}$ satisfies:

$$\max\left\{\mathbb{P}\left[\|\mathcal{A}(X_{t_0}, X_{t_1}, ..., X_{t_N}) - A\| \geq \frac{1}{10\sqrt{T}}\right], \mathbb{P}\left[\|\mathcal{A}(X_{t_0}, X_{t_1}, ..., X_{t_N}) - \bar{A}\| \geq \frac{1}{10\sqrt{T}}\right]\right\}$$

$$\geq \frac{1}{4e^3}.$$

We note that this special case can be easily generalized to any dimension $d$, since we can consider $A = -I_d$ and $\bar{A}$ satisfies $\bar{A}_{1,1} = A_{1,1} - \delta$, and for any $(i, j) \neq (1, 1)$, $\bar{A}_{i,j} = A_{i,j}$. In this case the last $d - 1$ dimension is independent of the first dimension, so it is essentially the same as the simplest one-dimensional case.

Denote $X = \{X_{t_0}, X_{t_1}, ..., X_{t_N}\}$ and $g(X), \bar{g}(X)$ be the probability density of $\psi_\theta$ and $\psi_{\bar{\theta}}$, respectively. For these two probability densities we have:

$$g(X) = \prod_{i=1}^{N} \frac{1}{\sqrt{2\pi\Gamma(t_i - t_{i-1})}} exp\left(-\frac{1}{2\Gamma(t_i - t_{i-1})}(X_{t_i} - e^{-(t_i - t_{i-1})}X_{t_{i-1}})^2\right),$$

and

$$\bar{g}(X) = \prod_{i=1}^{N} \frac{1}{\sqrt{2\pi\bar{\Gamma}(t_i - t_{i-1})}} exp\left(-\frac{1}{2\bar{\Gamma}(t_i - t_{i-1})}(X_{t_i} - e^{-(1+\delta)(t_i - t_{i-1})}X_{t_{i-1}})^2\right).$$

Where

$$\Gamma(t) = \int_{s=0}^{t} e^{-2s}ds = \frac{1}{2}(1 - e^{-2t}) \quad \bar{\Gamma}(t) = \int_{s=0}^{t} e^{(-2-2\delta)s}ds = \frac{1}{2 + 2\delta}(1 - e^{-(2+2\delta)t}).$$

Denote $\alpha_i = \sqrt{\frac{1}{\Gamma(t_i - t_{i-1})}}(X_{t_i} - e^{-(t_i - t_{i-1})}X_{t_{i-1}})$,

$\beta_i = \sqrt{\frac{1}{\Gamma(t_i - t_{i-1})}}(e^{-(t_i - t_{i-1})} - e^{-(1+\delta)(t_i - t_{i-1})})X_{t_{i-1}}$ and $\gamma_i = \sqrt{\frac{\Gamma(t_i - t_{i-1})}{\bar{\Gamma}(t_i - t_{i-1})}}$. Then

$$\ln\left(\frac{g(X)}{\bar{g}(X)}\right) = \sum_{i=1}^{N} -\ln(\gamma_i) + \frac{1}{2}\gamma_i^2(\alpha_i + \beta_i)^2 - \frac{1}{2}\alpha_i^2.$$

Next we show that $\left|\ln\left(\frac{g(X)}{\bar{g}(X)}\right)\right|$ is not large with high probability when $X$ follows the probability density of $g$. Consider the following subsets of $X$: $\mathcal{E}_1 = \left\{X\big| \left|\sum_{i=1}^{N} -\ln(\gamma_i) + \frac{1}{2}(\gamma_i^2 - 1)\alpha_i^2\right| \leq 1\right\}$.

$\mathcal{E}_2 = \left\{X\big| |\sum_{i=1}^{N} \gamma_i^2 \alpha_i \beta_i| \leq 1\right\}$ and $\mathcal{E}_3 = \left\{X\big| \frac{1}{2}\sum_{i=1}^{N} \gamma_i^2 \beta_i^2 \leq 1\right\}$. When $X$ lies in the intersection of these three sets, $\left|\ln\left(\frac{g(X)}{\bar{g}(X)}\right)\right|$ is guaranteed to be not very large.

Let $\mathbb{P}$ be the probability with respect to density $g$. We will explicitly show that $\mathbb{P}[X \in \mathcal{E}_k] \geq \frac{5}{6}(k = 1, 2, 3)$.

**Lower bound $\mathbb{P}[X \in \mathcal{E}_1]$** Firstly, we estimate $\sum_{i=1}^{N} \frac{1}{2}(\gamma_i^2 - 1) - \ln(\gamma_i)$. We first prove the following inequality:

$$0 \leq \gamma_i^2 - 1 \leq 2\delta \min\{1, t_i - t_{i-1}\}. \tag{24}$$

Let $t = t_i - t_{i-1}$. Then $\gamma_i^2 = (1+\delta)\frac{1-e^{-2t}}{1-e^{-(2+2\delta)t}}$.

The left hand side of this inequality is because $\Gamma_t \geq \bar{\Gamma}_t$, due to the reason that $e^{-2s} \geq e^{-(2+2\delta)s}$ for all $s \geq 0$ and when $f(x) \geq g(x)$ for any $x \in I$ we have: $\int_{x \in I} f(x)dx \geq \int_{x \in I} g(x)dx$. Now we consider the right hand side of the inequality.

**Case 1:** When $t \geq 1$, we directly use the fact that $1 - e^{-2t} \leq 1 - e^{-(2+2\delta)t}$ and obtain $\gamma_i \leq 1 + \delta$.

**Case 2:** When $t \in (0, 1]$, it suffices to show that

$$(1+\delta)(1 - e^{-2t}) \leq (1 + 2\delta t)(1 - e^{-(2+2\delta)t}).$$

Let $h(t) = (1+\delta)(1 - e^{-2t}) - (1 + 2\delta t)(1 - e^{-(2+2\delta)t})$, then

$$\begin{aligned}
h(t) &= \delta(1 - 2t) - e^{-2t}[1 + \delta - (1 + 2\delta t)e^{-2\delta t}] \\
&\leq \delta(1 - 2t - e^{-2t}) \\
&\leq 0.
\end{aligned}$$

Where for the first inequality we use the relation that $e^{-2\delta t} \leq \frac{1}{1+2\delta t}$. The second inequality is obtained by the relation that $e^{-2t} \geq 1 - 2t$.

Now we bound $\frac{1}{2}(\gamma_i^2 - 1) - \ln(\gamma_i)$. We first show that

$$0 \leq \frac{1}{2}(\gamma_i^2 - 1) - \ln(\gamma_i) \leq \frac{1}{4}(\gamma_i^2 - 1)^2.$$

Let $x = \gamma_i^2 - 1$ and we obtain $\frac{1}{2}(\gamma_i^2 - 1) - \ln(\gamma_i) = \frac{1}{2}[x - \ln(1 + x)]$, and the inequality is obtained directly since we have $x \geq \ln(1 + x) \geq x - \frac{1}{2}x^2 (x \geq 0)$.

Then we can bound $\sum_{i=1}^{N} \frac{1}{2}(\gamma_i^2 - 1) - \ln(\gamma_i)$ as

$$0 \leq \sum_{i=1}^{N} \frac{1}{2}(\gamma_i^2 - 1) - \ln(\gamma_i) \leq \sum_{i=1}^{N} \frac{1}{4}(\gamma_i^2 - 1)^2$$

$$\leq \sum_{i=1}^{N} \delta^2 \min(1, (t_i - t_{i-1}))^2$$

$$\leq \sum_{i=1}^{N} \delta^2 (t_i - t_{i-1})$$

$$\leq \delta^2 T$$

$$\leq \frac{1}{25} \,.$$

Now we bound $\sum_{i=1}^{N} \frac{1}{2}(\gamma_i^2 - 1)(\alpha_i^2 - 1)$. Notice that this variable has zero mean, so we can bound its variance and then apply Markov inequality to obtain a high probability bound.

At first, consider the variance of $\alpha_i^2 - 1$, denoted as $Var(\alpha_i^2 - 1)$. By noticing that $\alpha_i \sim \mathcal{N}(0,1)$, we can directly calculate that

$$Var(\alpha_i^2 - 1) = \int_{x \in \mathbb{R}} \frac{1}{\sqrt{2\pi}} e^{-\frac{1}{2}x^2} (x^2 - 1)^2 dx = 2 \,.$$

Since all the $\alpha_i$'s are independent, we have:

$$Var\left( \sum_{i=1}^{N} \frac{1}{2}(\gamma_i^2 - 1)(\alpha_i^2 - 1) \right) = \sum_{i=1}^{N} \frac{1}{4}(\gamma_i^2 - 1)^2 Var(\alpha_i^2 - 1)$$

$$\leq \frac{1}{2} \sum_{i=1}^{N} (\gamma_i^2 - 1)^2$$

$$\leq 2\delta^2 \sum_{i=1}^{N} \min(1, t_i - t_{i-1})^2$$

$$\leq 2\delta^2 T$$

$$\leq \frac{2}{25} \,.$$

By Markov inequality, we have:

$$\mathbb{P}\left[ \left| \sum_{i=1}^{N} \frac{1}{2}(\gamma_i^2 - 1)(\alpha_i^2 - 1) \right| \geq \frac{4}{5} \right] \leq Var\left( \sum_{i=1}^{N} \frac{1}{2}(\gamma_i^2 - 1)(\alpha_i^2 - 1) \right) / \left( \frac{4}{5} \right)^2 \leq \frac{1}{8} \,.$$

Finally, for the subset $\mathcal{E}_1 = \left\{ X \middle| \left| \sum_{i=1}^{N} - \ln(\gamma_i) + \frac{1}{2}(\gamma_i^2 - 1)\alpha_i^2 \right| \leq 1 \right\}$, we have:

$$\mathbb{P}\left[ x \in \mathcal{E}_1 \right] \geq 1 - \mathbb{P}\left[ \left| \sum_{i=1}^{N} \frac{1}{2}(\gamma_i^2 - 1)(\alpha_i^2 - 1) \right| \geq \frac{4}{5} \right] \geq \frac{7}{8} \,.$$

**Lower bound $\mathbb{P}[X \in \mathcal{E}_2]$** Since all the $\alpha_i$'s are independent, and $\alpha_i$ is independent of $\{\beta_1, ..., \beta_i\}$ and $\{\gamma_1, ..., \gamma_N\}$, we obtain that

$$\mathbb{E}\left[\left(\sum_{i=1}^{N}\gamma_i^2\alpha_i\beta_i\right)^2\right] = \mathbb{E}\left[\sum_{i=1}^{N}(\gamma_i^2\alpha_i\beta_i)^2\right]$$

$$= \mathbb{E}\left[\sum_{i=1}^{N}(\gamma_i^2\beta_i)^2\right]$$

$$= \sum_{i=1}^{N}\mathbb{E}\left[(\gamma_i^2\beta_i)^2\right].$$

We have shown that $\gamma_i^2 \leq 1 + 2\delta$. Then for $T \geq 1$ we have: $\gamma_i^4 \leq (1 + \frac{2}{5})^2 \leq 2$. Therefore, we obtain:

$$\mathbb{E}\left[\left(\sum_{i=1}^{N}\gamma_i^2\alpha_i\beta_i\right)^2\right] \leq 2\sum_{i=1}^{N}\mathbb{E}\left[\beta_i^2\right].$$

Now we upper bound $\mathbb{E}\left[\beta_i^2\right]$, where $\beta_i = \sqrt{\frac{1}{\Gamma(t_i-t_{i-1})}}(e^{-(t_i-t_{i-1})} - e^{-(1+\delta)(t_i-t_{i-1})})X_{t_{i-1}}$

Firstly, we show that

$$\sqrt{\frac{1}{\Gamma(t_i-t_{i-1})}}(e^{-(t_i-t_{i-1})} - e^{-(1+\delta)(t_i-t_{i-1})}) \leq \delta\sqrt{t_i - t_{i-1}}. \qquad (25)$$

Again denote $t = t_i - t_{i-1}$. By using $\Gamma_t = \frac{1}{2}(1 - e^{-2t})$, it suffices to show that

$$e^{-t} - e^{-(1+\delta)t} \leq \delta\sqrt{\frac{1}{2}t(1 - e^{-2t})}.$$

By multiplying $e^t$ on both sides, the inequality is equivalent to

$$1 - e^{-\delta t} \leq \delta\sqrt{\frac{1}{2}t(e^{2t} - 1)}.$$

This is true since $e^{-\delta t} \geq 1 - \delta t$, and $e^{2t} \geq 1 + 2t$, implying that

$$1 - e^{-\delta t} \leq \delta t \leq \delta\sqrt{\frac{1}{2}t(e^{2t} - 1)}.$$

With this result, we can upper bound $2\sum_{i=1}^{N}\mathbb{E}\left[\beta_i^2\right]$ by

$$2\sum_{i=1}^{N}\mathbb{E}\left[\beta_i^2\right] \leq \sum_{i=1}^{N}2\delta^2(t_i - t_{i-1})\mathbb{E}\left[X_{t_{i-1}}^2\right].$$

Finally, since $X_t \sim \mathcal{N}(0, \Gamma(t))$, for all $t \geq 0$,

$$\mathbb{E}\left[X_t^2\right] = \Gamma_t = \frac{1}{2}(1 - e^{-2t}) \leq 1.$$

Therefore, we obtain

$$\mathbb{E}\left[\left(\sum_{i=1}^{N}\gamma_i^2\alpha_i\beta_i\right)^2\right] \leq 2\sum_{i=1}^{N}\mathbb{E}\left[\beta_i^2\right] \leq \sum_{i=1}^{N}2\delta^2(t_i-t_{i-1})\mathbb{E}\left[X_{t_{i-1}}^2\right] \leq \sum_{i=1}^{N}2\delta^2(t_i-t_{i-1}) = 2T\delta^2 = \frac{2}{25}$$

Again by using Markov inequality, we obtain:

$$\mathbb{P}\left[|\sum_{i=1}^{N}\gamma_i^2\alpha_i\beta_i| > 1\right] \leq \frac{2}{25}.$$

Which follows that

$$\mathbb{P}\left[X \in \mathcal{E}_2\right] = 1 - \mathbb{P}\left[|\sum_{i=1}^{N}\gamma_i^2\alpha_i\beta_i| \geq 1\right] \geq \frac{23}{25}.$$

**Lower bound** $\mathbb{P}[X \in \mathcal{E}_3]$ We have shown that $\gamma_i^2 \leq 2, \forall i$ and $\sum_{i=1}^{N}\mathbb{E}\left[\beta_i^2\right] \leq \delta^2 T$. Therefore,

$$\mathbb{E}\left[\frac{1}{2}\sum_{i=1}^{N}\gamma_i^2\beta_i^2\right] \leq \delta^2 T \leq \frac{2}{25}.$$

And we also have

$$\mathbb{P}\left[X \in \mathcal{E}_3\right] = 1 - \mathbb{P}\left[\frac{1}{2}\sum_{i=1}^{N}\gamma_i^2\beta_i^2 > 1\right] \geq \frac{23}{25}.$$

Now we come back to prove the theorem. With lower bounds of $\mathbb{P}[X \in \mathcal{E}_1], \mathbb{P}[X \in \mathcal{E}_2], \mathbb{P}[X \in \mathcal{E}_3]$, we have

$$\mathbb{P}\left[X \in \mathcal{E}_1 \cap \mathcal{E}_2 \cap \mathcal{E}_3\right] \geq 1 - (1 - \mathbb{P}[X \in \mathcal{E}_1]) - (1 - \mathbb{P}[X \in \mathcal{E}_2]) - (1 - \mathbb{P}[X \in \mathcal{E}_3]) \geq \frac{1}{2}.$$

With this bound, we have:

$$\mathbb{E}_{X \sim g}\left[1\left(|\phi(X) - A| \geq \frac{1}{10\sqrt{T}}\right)\right] + \mathbb{E}_{X \sim \bar{g}}\left[1\left(|\phi(X) - \bar{A}| \geq \frac{1}{10\sqrt{T}}\right)\right]$$

$$\geq \int_{X \in \mathcal{E}_1 \cap \mathcal{E}_2 \cap \mathcal{E}_3} g(X)\mathbb{E}\left[1\left(\|\phi(X) - A\| \geq \frac{1}{10\sqrt{T}}\right)|X\right] + \bar{g}(X)\mathbb{E}\left[1\left(\|\phi(X) - \bar{A}\| \geq \frac{1}{10\sqrt{T}}\right)|X\right]dX$$

$$\geq \int_{X \in \mathcal{E}_1 \cap \mathcal{E}_2 \cap \mathcal{E}_3} \min\{g(X), \bar{g}(X)\}dX$$

$$\geq \int_{X \in \mathcal{E}_1 \cap \mathcal{E}_2 \cap \mathcal{E}_3} \frac{1}{e^3}g(X)dX$$

$$\geq \frac{1}{2e^3}.$$

Where the second inequality is because $\|\phi(X) - A\| + \|\phi(X) - \bar{A}\| \geq \|A - \bar{A}\| = \frac{1}{5\sqrt{T}}$ so we cannot have both $\|\phi(X) - A\| \leq \frac{1}{10\sqrt{T}}$ and $\|\phi(X) - \bar{A}\| \leq \frac{1}{10\sqrt{T}}$. The third inequality is because for any $X \in \mathcal{E}_1 \cap \mathcal{E}_2 \cap \mathcal{E}_3$, we have

$$\left| \ln \frac{g(X)}{\bar{g}(X)} \right| = \left| \sum_{i=1}^{N} -\ln(\gamma_i) + \frac{1}{2}\gamma_i^2(\alpha_i + \beta_i)^2 - \frac{1}{2}\alpha_i^2 \right|$$

$$\leq \left| \sum_{i=1}^{N} -\ln(\gamma_i) + \frac{1}{2}(\gamma_i^2 - 1)\alpha_i^2 \right|$$

$$+ \left| \sum_{i=1}^{N} \gamma_i^2 \alpha_i \beta_i \right|$$

$$+ \frac{1}{2}\sum_{i=1}^{N} \gamma_i^2 \beta_i^2$$

$$\leq 3 \,,$$

implying that $\bar{g}(X) \geq \frac{1}{e^3}g(X)$.

Therefore, we have:

$$\max\left\{ \mathbb{P}_{X \sim g}\left[ |\phi(X) - A| \geq \frac{1}{10\sqrt{T}} \right], \mathbb{P}_{X \sim \bar{g}}\left[ |\phi(X) - \bar{A}| \geq \frac{1}{10\sqrt{T}} \right] \right\} \geq \frac{1}{4e^3}\,.$$

This means that for any algorithm, it cannot achieve $\frac{1}{10\sqrt{T}}$ estimation error with success probability $1 - \frac{1}{4e^3}$ for at least one of the systems controlled by $(A, 0)$ and $(\bar{A}, 0)$.

$\square$

## B  REGRET ANALYSIS

Having demonstrated the results of system identification for continuous-time linear systems, we leverage these findings to establish upper bounds on the regret for Algorithm 3. Elaborations on the details will be presented in the subsequent sections.

### B.1  CONVERGENCE OF $P$ AND THE ESTIMATION ERROR OF $K$

In this section we provide the following Lemma 14, along with its proof, which shows that $\|P - P_*\|$ converges at the same speed as $\|\hat{A} - A\| + \|\hat{B} - B\|$.

**Lemma 14.** *There exist constants $\epsilon_0 > 0$ and $C_2 > 0$ such that as long as $\|\hat{A} - A\|, \|\hat{B} - B\| \leq \epsilon$ for some $0 < \epsilon < \epsilon_0$, with $P$ obtained from equation 5 we have:*

$$\|P - P_*\| \leq C_2\epsilon\,. \tag{26}$$

Recall that the optimal dynamic is $K_* = -R^{-1}B^{\mathrm{T}}P_*$ with $P_*$ obtained from equation equation 5. Now we consider the distance between it and the sub-optimal dynamic $\bar{K} = -R^{-1}B^{\mathrm{T}}P$ with $P$ obtained from equation 5 with $(\hat{A}, \hat{B})$. Denote $\Delta A = \hat{A} - A$ and $\Delta B = \hat{B} - B$, along with $\|\Delta A\|, \|\Delta B\| \leq \epsilon$ where $\epsilon \in [0, \epsilon_0]$ with some $\epsilon_0$ determined later. We establish the proof by constructing a sequence of matrices $(P_k)_{k \geq 0}$, and we will prove that such sequence converges to the unique symmetric solution $P$ satisfying

$$P\hat{B}R^{-1}\hat{B}^{\mathrm{T}}P - \hat{A}^{\mathrm{T}}P - P\hat{A} - Q = 0\,.$$

At first we introduce a solution of a particular kind of matrix equation (Kleinman, 1968).

**Lemma 15.** *Suppose $A$ satisfies $\alpha(A) = \max\{\Re(\lambda_i)|\lambda_i \in \lambda(A)\} < 0$. $Q$ is a symmetric matrix. Consider such a function*

$$A^{\mathrm{T}}X + XA + Q = 0\,. \tag{27}$$

*Then, the unique symmetric solution $X$ of this equation can be expressed as:*

$$X = \int_{t \geq 0} e^{A^{\mathrm{T}}t} Q e^{At} dt \,. \tag{28}$$

Now we consider the relation between $P$ and $P_*$. The core is iteratively constructing a sequence of matrices $P_k$ such that $P_0 = P_*$ and $\lim_{k \to +\infty} P_k = P$. Such matrices follows the relation $P_{k+1} = P_k + \Delta P_k$ where $\Delta P_k$ converges rapidly. As for the starting case, consider the expansion

$$(P_* + \Delta P)(B + \Delta B)R^{-1}(B + \Delta B)^{\mathrm{T}}(P_* + \Delta P)$$
$$- (A + \Delta A)^{\mathrm{T}}(P_* + \Delta P) - (P_* + \Delta P)(A + \Delta A) - Q$$
$$= \left[(B + \Delta B)R^{-1}(B + \Delta B)^{\mathrm{T}}P_* - A - \Delta A\right]^{\mathrm{T}} \Delta P$$
$$+ \Delta P \left[(B + \Delta B)R^{-1}(B + \Delta B)^{\mathrm{T}}P_* - A - \Delta A\right]$$
$$+ \left[P_* BR^{-1}B^{\mathrm{T}}P_* - A^{\mathrm{T}}P_* - P_* A - Q\right] + P_* \left[\Delta B \left(R^{-1}(B + \Delta B)^{\mathrm{T}}\right) + BR^{-1}\Delta B\right] P_*$$
$$+ \Delta P(B + \Delta B)R^{-1}(B + \Delta B)^{\mathrm{T}}\Delta P \,.$$

Define

$$A_0 = A + \Delta A - (B + \Delta B)R^{-1}(B + \Delta B)^{\mathrm{T}}P_* \,,$$
$$F_0 = -P_* \left[\Delta B \left(R^{-1}(B + \Delta B)^{\mathrm{T}}\right) + BR^{-1}\Delta B\right] P_* \,.$$

We set $\Delta P_0$ be a solution of

$$A_0^{\mathrm{T}}\Delta P_0 + \Delta P_0 A_0 + F_0 = 0 \,.$$

which satisfies that (see Lemma 15)

$$\Delta P_0 = \int_{t \geq 0} e^{A_0^{\mathrm{T}}t} F_0 e^{A_0 t} dt \,,$$

$$\|\Delta P_0\| \leq \int_{t \geq 0} e^{2\alpha(A_0)t}\|F_0\|dt = \frac{1}{-2\alpha(A_0)}\|F_0\| \leq \frac{1}{-\alpha(A_0)}\|P_*\|^2(\|BR^{-1}\|\epsilon + \|R^{-1}\|\epsilon^2) \,.$$

This $\Delta P_0$ also satisfies

$$(P_* + \Delta P_0)(B + \Delta B)R^{-1}(B + \Delta B)^{\mathrm{T}}(P_* + \Delta P_0)$$
$$- (A + \Delta A)^{\mathrm{T}}(P_* + \Delta P) - (P_* + \Delta P)(A + \Delta A) - Q$$
$$= \Delta P_0(B + \Delta B)R^{-1}(B + \Delta B)^{\mathrm{T}}\Delta P_0 \,.$$

An important thing is to guarantee that $A_0$ is stable, and $|\alpha(A_0)|$ can not be too closed to zero. For any $\epsilon_1 \in (0, 1)$ and $C_1 = \|R^{-1}\|\|P_*\| + 1 + 2\|BR^{-1}\|\|P_*\|$, as long as $\epsilon \leq \epsilon_1$, $\|A_0 - (A - BR^{-1}B^{\mathrm{T}}P_*)\| \leq C_1\epsilon$. Furthermore, there exists $\epsilon_2 > 0$ such that if $\|X - (A - R^{-1}B^{\mathrm{T}}P_*)\| \leq \epsilon_2$, then $\alpha(X) \leq \frac{1}{2}\alpha(A - R^{-1}B^{\mathrm{T}}P_*)$(the work of (Shirani Faradonbeh & Shirani Faradonbeh, 2023) shows this result). We can further let this $\epsilon_2$ satisfies that, as long as $\|\Delta A\|, \|\Delta B\|, \|\Delta P\| \leq \epsilon_2$, we always have:

$$\alpha\left(A + \Delta(A) - (B + \Delta B)R^{-1}(B + \Delta B)^{\mathrm{T}}(P_* + \Delta P)\right) \leq \frac{1}{2}\alpha(A - BR^{-1}B^{\mathrm{T}}P_*) \,. \tag{29}$$

Now we additionally set $\epsilon_1$ satisfying $\epsilon_1 \leq \frac{1}{2C_1}\epsilon_2$ and $\|R^{-1}\|\epsilon_1 \leq 1$, then for all $\epsilon \leq \epsilon_1$,

$$\|\Delta P_0\| \leq \frac{2}{-\alpha(A - BR^{-1}B^{\mathrm{T}}P_*)}\|P_*\|^2(1 + \|BR^{-1}\|)\epsilon \,.$$

Denote $P_1 = P_0 + \Delta P_0$, $C_2 = \frac{2}{-\alpha(A - BR^{-1}B^{\mathrm{T}}P_*)}\|P_*\|^2(1 + \|BR^{-1}\|)$, and set some constant $C_3$ satisfying $C_3 \geq \|BR^{-1}B^{\mathrm{T}}\| + 2\|BR^{-1}\| + \|R^{-1}\|$. We then inductively define $P_{k+1}$ and $\Delta P_k$ ($k \geq 1$). For defined $\Delta P_{k-1}$, we set $P_k = P_{k-1} + \Delta P_{k-1}$, which satisfies

$$P_k(B + \Delta B)R^{-1}(B + \Delta B)^{\mathrm{T}}P_k - (A + \Delta A)^{\mathrm{T}}P_k - P_k(A + \Delta(A)) - Q$$
$$= \Delta P_{k-1}(B + \Delta B)R^{-1}(B + \Delta B)^{\mathrm{T}}\Delta P_{k-1} \,.$$

Then we denote $A_k = A + \Delta A - (B + \Delta B)R^{-1}(B + \Delta B)^{\mathrm{T}}P_k$, and set $\Delta P_k$ satisfying:

$$A_k^{\mathrm{T}}\Delta P_k + \Delta P_k A_k = \Delta P_{k-1}(B + \Delta B)R^{-1}(B + \Delta B)^{\mathrm{T}}\Delta P_{k-1}\,.$$

By the hypothesis of $\epsilon_2$, as long as $\|P_k - P_*\| \leq \epsilon_2$, we have $\alpha(A_k) \geq \frac{1}{2}\alpha(A - BR^{-1}B^{\mathrm{T}}P_*)$. By using equation 28 we obtain that $\|\Delta P_k\| \leq C_4\|\Delta P_{k-1}\|^2$, where $C_4 = \frac{2}{-\alpha(A - BR^{-1}B^{\mathrm{T}}P_*)}C_3$. Now if we define $P_{k+1} = P_k + \Delta P_k$, $P_{k+1}$ also satisfies:

$$P_{k+1}(B + \Delta B)R^{-1}(B + \Delta B)^{\mathrm{T}}P_{k+1} - (A + \Delta A)^{\mathrm{T}}P_{k+1} - P_{k+1}(A + \Delta(A)) - Q$$
$$= \Delta P_k(B + \Delta B)R^{-1}(B + \Delta B)^{\mathrm{T}}\Delta P_k\,,$$

Then these sequences $\Delta P_k$ and $P_k$ are well defined, along with the relation that $P_{k+1} = P_k + \Delta P_k$. Furthermore, when $\|P_k - P_*\| \leq \epsilon_2$, we have $\|\Delta P_{k+1}\| \leq C_4\|\Delta P_k\|^2$. Note that for the base case we have $\|\Delta P_0\| \leq C_2\epsilon$.

Finally, it remains to constrain $\|P_k - P_*\|$. By choosing $\epsilon \leq \min(\frac{1}{2C_2C_4}, \frac{1}{2C_2}\epsilon_2, 1)$, we obtain $\|\Delta P_0\| \leq C_2\epsilon$. We can also see that if for all $0 \leq k \leq m$, $\|\Delta P_k\| \leq 2^{-k}C_2\epsilon$, then $\|P_m - P_*\| \leq 2(1 - 2^{-m+1})C_2\epsilon \leq \epsilon_2$ so that $\|\Delta P_{m+1}\| \leq C_4\|\Delta P_m\|^2 \leq 2^{-m-1}C_2\epsilon$. So by induction we see that $\|\Delta P_k\| \leq 2^{-k}C_2\epsilon$ for any $k$.

On the other hand, since $\|\Delta P_k\| \leq 2^{-k}\|\Delta P_0\|$, $\lim_{k \to +\infty} P_k = P_\infty$ exists, and such $P_\infty$ is the unique symmetric solution of

$$P(B + \Delta B)R^{-1}(B + \Delta B)^{\mathrm{T}}P - (A + \Delta A)^{\mathrm{T}}P - P(A + \Delta(A)) - Q = 0\,,$$

such that $(A + \Delta A) - (B + \Delta B)R^{-1}(B + \Delta B)^{\mathrm{T}}P$ is stable (recall the stable margin in equation 29, which implies that $(A + \Delta A) - (B + \Delta B)R^{-1}(B + \Delta B)^{\mathrm{T}}P_\infty$ is stable).

So $P_\infty$ is exactly $P$, satisfying $\|P - P_*\| \leq 2C_2\epsilon$.

Therefore, we conclude that there exists some $\epsilon_0 > 0$ and constant $C$, both depending on $A, B, K, d, p$ such that for any $\epsilon \in [0, \epsilon_0]$, $\|P - P_*\| \leq C\epsilon$ as long as $\|\hat{A} - A\|, \|\hat{B} - B\| \leq \epsilon$.

Then we apply our results for system identification to establish an upper bound for $\|\bar{K} - K_*\|$.

Based on Lemma 14, fix constant $\epsilon_1 > 0$ and constant $C_1 \geq 0$ so that we have $\|P - P_*\| \leq C_1\left(\|\hat{A} - A\| + \|\hat{B} - B\|\right)$ whenever $\|\hat{A} - A\| + \|\hat{B} - B\| \leq \epsilon_1$

We set $C_2 \geq 1$ be two times the constant $C$ in Lemma 8, and obtain that, when $\log^2(1/\delta) \leq \frac{T^{1/2}}{C_2}$ and $T^{1/2} \geq C_2\|X_0\|_2^2$, we have:

$$\mathbb{P}\left[\|\hat{A} - A\| + \|\hat{B} - B\| \leq 2C_2\sqrt{\frac{\log(1/\delta)}{T^{1/2}}}\right] \geq 1 - \delta\,.$$

Then, for $\log(1/\delta) \leq \min\left\{\frac{T\epsilon_1^2}{4C_2^2}, \frac{T^{1/4}}{C_2^{1/2}}\right\} \leq \frac{T^{1/4}\epsilon_1^2}{4C_2^2}$, we have:

$$\mathbb{P}\left[\|P - P_*\| \leq 2C_1C_2\sqrt{\frac{\log(1/\delta)}{T^{1/2}}}\right] \geq 1 - \delta\,. \tag{30}$$

Finally, since $\bar{K} = -R^{-1}(\hat{B})^{\mathrm{T}}P$, $K_* = -R^{-1}B^{\mathrm{T}}P_*$, we have:

$$\|\bar{K} - K_*\| \leq \|R^{-1}\|\left[\|\hat{B} - B\|\|P\| + \|B\|\|P - P_*\|\right]\,.$$

We can reset $C_1$ such that $\|\bar{K} - K_*\| \leq C_1\left(\|\hat{A} - A\| + \|\hat{B} - B\|\right)$ whenever $\|\hat{A} - A\| + \|\hat{B} - B\| \leq \epsilon_1$, and combine this with equation 30, we have: for any $\log(1/\delta) \leq \frac{T^{1/4}\epsilon_1^2}{4C_2^2}$

$$\mathbb{P}\left[\|\bar{K} - K_*\| \leq 2C_1C_2\sqrt{\frac{\log(1/\delta)}{T^{1/2}}}\right] \geq 1 - \delta\,. \tag{31}$$

With this probability bound on $\|\bar{K} - K_*\|$, we can further upper bound the regret, shown in the following part.

## B.2 Key Lemmas

We first upper bound the radius of a single trajectory with stable controller, for which we introduce and provide a proof for the following lemma:

**Lemma 16.** *Consider the continuous system $dX_t = AX_t dt + dW_t$ such that $\alpha(A) < 0$ where $\alpha(A)$ is the largest real component of $A$ and $W$ is a standard Brownian noise. Then, w.p. at least $1 - \delta$:*

$$\sup_{0 \leq t \leq T} \left( \|X_t\|_2 - e^{\alpha(A)t} \|X_0\|_2 \right) \leq C\sqrt{d \log((1+T)/\delta)} \,.$$

Then we concentrate on how the error $\|P - P_*\|$ will influence the regret during the exploitation phase. For a dynamic $U$ with $\alpha(A + BU) < 0$, we define a cost function:

$$cost(U) = \mathrm{tr}\left( \int_{t \geq 0} (e^{(A+BU)t})^{\mathrm{T}} (Q + U^{\mathrm{T}} R U) e^{(A+BU)t} dt \right) \,.$$

The convergence rate of this cost function is stated in the following lemma:

**Lemma 17.** *Let $U_*$ minimize $cost(U)$. Then, there exists $\epsilon_0 \geq 0$ such that for any $\|\Delta U\| = 1$ and $\epsilon \in [0, \epsilon_0]$, we have:*

$$cost(U_* + \epsilon \Delta U) - cost(U_*) \leq C_1 \epsilon^2 \,.$$

The above result shows the average cost per unit time when applying fixed controller for infinite time.

Then we further consider the case when the running time is finite. We derive the following lemma:

**Lemma 18.** *Let $U_*$ follows the same definition as in Lemma 17. Then, for some $\epsilon > 0$, there exist constants $C_2$ and $C_3$ (independent of $U$) such that for all $T > 0$ and any $U$ such that $\|U - U_*\| \leq \epsilon$,*

$$|J_T - cost(U)T| \leq C_2 \|x\|_2^2 + C_3 \,.$$

*Here $J_T$ is the expected cost of the policy that takes action by $U_t = U X_t$ ($t \in [0, T]$), with initial state $X_0 = x$.*

With this lemma, by definition of $U_*$, we actually have $U_* = K_*$, where $K_* = -R^{-1} B^{\mathrm{T}} P_*$ and $P_*$ is the solution of equation 4. Since such $C_2, C_3$ also satisfy:

$$|J_T^* - cost(U_*)T| \leq C_2 \|x\|_2^2 + C_3 \,,$$

so it follows that

$$R_T = J_T - J_T^* \leq 2C_2 \|x\|_2^2 + 2C_3 \,. \tag{32}$$

## B.3 Proof of Lemma 16

We first upper bound the radius of a single trajectory with stable controller, for which we introduce and provide a proof for the following lemma:

**Lemma 16.** *Consider the continuous system $dX_t = AX_t dt + dW_t$ such that $\alpha(A) < 0$ where $\alpha(A)$ is the largest real component of $A$ and $W$ is a standard Brownian noise. Then, w.p. at least $1 - \delta$:*

$$\sup_{0 \leq t \leq T} \left( \|X_t\|_2 - e^{\alpha(A)t} \|X_0\|_2 \right) \leq C\sqrt{d \log((1+T)/\delta)} \,.$$

*Proof.* The trajectory $X_t$ with differential equation $dX_t = AX_t + dW_t$ can be derived as

$$X_t = e^{At} X_0 + \int_{s=0}^{t} e^{A(t-s)} dW_t \,.$$

Lemma 6 tells that when $A$ is stable, $\left\| e^{At} X_0 \right\|_2 \leq e^{\alpha(A)t} \|X_0\|_2$. So it suffices to show that

$$\mathbb{P}\left[ \sup_{0 \leq t \leq T} \left\| \int_{s=0}^{t} e^{A(t-s)} dW_t \right\|_2 \geq C\sqrt{d \log(1+T)/\delta} \right] \leq \delta \,.$$

Let $T = T_0 h$ with $T_0$ be an integer. We first consider the set of points $\{X_{kh}\}$. Denote $w_k := \int_{t=0}^{kh} e^{A(kh-t)} dW_t$, then $w_k \sim \mathcal{N}(0, \Sigma_k)$ with $\Sigma_k = \int_{t=0}^{kh} e^{At} e^{A^\mathrm{T} t} dt$. This $\Sigma_h$ also satisfies

$$\|\Sigma_k\| \leq \int_{t=0}^{h} \left\| e^{At} \right\|^2 dt \leq \int_{t=0}^{kh} e^{2\alpha(A)t} dt \leq \frac{1}{2|\alpha(A)|}.$$

Which follows that $\sup_{0 \leq k \leq T_0} \|w_k\|_2 \leq 2\sqrt{\frac{d}{|\alpha(A)|} \log((1+T_0)/\delta)}$, w.p. at least $1 - \delta$.

Next we consider any $X_{kh+t}$ with $t \in [0, h]$. Bounding such terms requires the Doob's martingale inequality (Durrett), stated as in Lemma 19. We denote $x_t^k = \int_{s=0}^{t} e^{A(t-s)} dW_{kh+s} ds$ with corresponding filtration $\mathcal{F}_t$. We also define $Z_t^k := e^{\lambda \| e^{-At} x_t^k \|_2^2}$ with $\lambda \geq 0$. Then $Z_t^k$ is a submartingale under the filtration $\mathcal{F}_t$, since for any $t \geq s$,

$$\mathbb{E}\left[Z_t^k | \mathcal{F}_s\right] = \mathbb{E}\left[\exp\left(\lambda \left\| e^{-As} x_s^k + \int_{t_1=s}^{t} e^{-At_1} dW_{kh+t_1} \right\|_2^2\right) \Big| x_s^k\right] \geq e^{\lambda \| e^{-As} x_s^k \|_2^2} = Z_s^k.$$

Where we notice that $\mathbb{E}\left[\left\| e^{-As} x_s^k + \int_{t_1=s}^{t} e^{-At_1} dW_{kh+t_1} \right\|_2^2 | x_s^k\right] \geq \left\| e^{-As} x_s^k \right\|_2^2$, and apply Jensen's inequality on the non-decreasing convex function $f(x) = e^{\lambda x}$ to obtain the above inequality. Now we apply Lemma 19 and get

$$\mathbb{P}\left[\sup_{t \in [0,h]} \left\| e^{-At} x_t^k \right\|_2 \geq C\right] \leq e^{-\lambda C^2} \mathbb{E}[Z_h^k]. \tag{33}$$

We next estimate $\mathbb{E}(Z_h^k)$. Since $e^{-Ah} x_h^k = \int_{t=0}^{h} e^{-At} dW_{kh+t}$, we obtain that $e^{-Ah} x_h^k \sim \mathcal{N}(0, \bar{\Sigma})$, where

$$\bar{\Sigma} = \int_{t=0}^{h} e^{-At} e^{-A^\mathrm{T} t} dt.$$

By setting $\lambda = \frac{1}{4\|\bar{\Sigma}\|}$, it can be computed that

$$\mathbb{E}\left[e^{\lambda \| e^{-Ah} x_h^k \|_2^2}\right] = \int_{x \in \mathbb{R}^d} \frac{1}{(2\pi)^{d/2} \sqrt{\det(\bar{\Sigma})}} e^{-\frac{1}{2} x^\mathrm{T} \Sigma_1^{-1} x} e^{\lambda x^\mathrm{T} I_d x} dx$$

$$= \sqrt{\frac{1}{\det(\bar{\Sigma}) \det(\Sigma_1^{-1} - 2\lambda I_d)}}$$

$$= \sqrt{\frac{1}{\det(I_d - 2\lambda \bar{\Sigma})}}$$

$$\leq 2^{d/2},$$

where the last inequality is because $I_d - 2\lambda \bar{\Sigma} \succeq \frac{1}{2} I_d$.

We combine this result with equation 33 and obtain:

$$\mathbb{P}\left[\sup_{0 \leq k \leq T_0-1, 0 \leq t \leq h} \left\| x_t^k \right\|_2 \geq 2 e^{\|A\| h} \left\| \bar{\Sigma} \right\|^{1/2} \sqrt{\log(2^{d/2} T_0/\delta)}\right]$$

$$\leq \sum_{k=0}^{T_0-1} \mathbb{P}\left[\sup_{t \in [0,h]} Z_t^k \geq 2^{d/2} \frac{T_0}{\delta}\right]$$

$$\leq \sum_{k=0}^{T_0-1} \mathbb{P}\left[\sup_{t \in [0,h]} Z_t^k \geq \frac{T_0}{\delta} \mathbb{E}(Z_h^k)\right]$$

$$\leq \delta.$$

Finally, since $X_{kh+t} = e^{A(kh+t)}X_0 + e^{At}w_k + x_t^k$, it follows that

$$\|X_{kh+t}\|_2 \leq \left\|e^{A(kh+t)}X_0\right\|_2 + \left\|e^{At}w_k\right\|_2 + \left\|x_t^k\right\|_2$$
$$\leq e^{\alpha(A)(kh+t)}\|X_0\|_2 + \|w_k\|_2 + \left\|x_t^k\right\|_2 .$$

By applying union bound on $\|w_k\|_2$ and $\left\|x_t^k\right\|_2$ we finally obtain Lemma 16. $\qquad\square$

**Lemma 19** (Doob's martingale inequality). *Let $X_1, \ldots, X_n$ be a discrete-time submartingale relative to a filtration $\mathcal{F}_1, \ldots, \mathcal{F}_n$ of the underlying probability space, which is to say:*

$$X_i \leq \mathbb{E}\left[X_{i+1} \mid \mathcal{F}_i\right].$$

*The submartingale inequality says that*

$$\mathbb{P}\left[\max_{1 \leq i \leq n} X_i \geq C\right] \leq \frac{\mathbb{E}\left[\max\left(X_n, 0\right)\right]}{C}$$

*for any positive number $C$.*

*Moreover, let $X_t$ be a submartingale indexed by an interval $[0, \text{T}]$ of real numbers, relative to a filtration $F_t$ of the underlying probability space, which is to say:*

$$X_s \leq \text{E}\left[X_t \mid \mathcal{F}_s\right]$$

*for all $s < t$. The submartingale inequality says that if the sample paths of the martingale are almost-surely right-continuous, then*

$$\mathbb{P}\left[\sup_{0 \leq t \leq T} X_t \geq C\right] \leq \frac{\mathbb{E}\left[\max\left(X_T, 0\right)\right]}{C}$$

*for any positive number $C$.*

### B.4 PROOF OF LEMMA 17

In this section, we proof Lemma 17 which refers to the convergence rate of the cost function:

**Lemma 17.** *Let $U_*$ minimize $cost(U)$. Then, there exists $\epsilon_0 \geq 0$ such that for any $\|\Delta U\| = 1$ and $\epsilon \in [0, \epsilon_0]$, we have:*

$$cost(U_* + \epsilon\Delta U) - cost(U_*) \leq C_1\epsilon^2 .$$

*Proof.* For any $\|\Delta U\| = 1$ and $\epsilon > 0$, consider $U = U_* + \epsilon\Delta U$, we show that as $\epsilon \to 0$, there exists $V \in \mathbb{R}^d$ such that $\text{tr}(V) = 0$, and

$$\int_{t \geq 0} e^{(A+BU)^{\text{T}}t}(Q + U^{\text{T}}RU)e^{(A+BU)t}dt - \int_{t \geq 0} e^{(A+BU_*)^{\text{T}}t}(Q + U_*^{\text{T}}RU_*)e^{(A+BU_*)t}dt$$
$$= \epsilon V + \mathcal{O}(\epsilon^2) .$$

Let $D(\epsilon, t) = e^{(A+B(U_*+\epsilon\Delta U))t} - e^{(A+BU_*)t}$. The most important intuition is that $D(\epsilon, t)$ can be represented by the form of $D(\epsilon, t) = \epsilon D_1(t) + \epsilon^2 D_2(\epsilon, t)$, where $D_1(t)$ does not depend on $\epsilon$, and the residual $D_2(\epsilon, t)$ can be well bounded. Now we find such $D_1(t)$ and upper bound $\|D_2(\epsilon, t)\|$. For $t \leq t_0 = \frac{1}{\max\{\|A+BU_*\|, \|B\|\}}$ and $\epsilon < 1$, the Taylor expansion of $e^{(A+B(U_*+\epsilon\Delta U))t}$ can be represented as follows:

$$D(\epsilon, t) = \sum_{k \geq 1} \frac{1}{k!}\left[(A + BU_* + \epsilon B\Delta U)^k t^k - (A + BU_*)^k t^k\right]$$
$$= \sum_{k \geq 1} \frac{1}{k!}\left[\left(\sum_{i=0}^{k-1}(A + BU_*)^i(B\Delta U_*)(A + BU_*)^{k-1-i}\right)\epsilon + D_1(\epsilon, k)\epsilon^2\right]t^k,$$

where $D_1(\epsilon, k)$ is the residual of $(A + BU + \epsilon B\Delta U)^k - (A + BU)^k$ with order at least $\epsilon^2$. This sequence of matrices are expressed and bounded as follows.

$$D_1(k, \epsilon) = \sum_{i=2}^{k} \epsilon^i \sum_{j_1 + \ldots + j_{i+1} = k-i} (A + BU_*)^{j_1} (B\Delta U)(A + BU_*)^{j_2}(B\Delta U)\ldots(A + BU_*)^{j_{i+1}},$$

$$\|D_1(k, \epsilon)\| \le \sum_{i=2}^{k} \frac{k!}{i!(k-i)!} \|A + BU_*\|^{k-i} \|B\|^i \epsilon^{i-2}.$$

Thus we have:

$$\left\| \sum_{k \ge 1} \frac{t^k}{k!} D_1(k, \epsilon) \right\| \le \sum_{k \ge 2} \sum_{i \ge 2} \frac{1}{i!(k-i)!} \le 4.$$

Define $E(t)$ and $E_1(\epsilon, t)$ as follows: for $0 \le t \le t_0$, let

$$E(t) = \sum_{k \ge 1} \frac{t^k}{k!} \sum_{i=0}^{k-1} (A + BU_*)^i (B\Delta U_*)(A + BU_*)^{k-1-i}, E_1(\epsilon, t) = \sum_{k \ge 1} \frac{t^k}{k!} D_1(k, \epsilon),$$

and for $t \in [\frac{1}{2}t_0, t_0]$, $l \ge 1$, we inductively define $E(2^l t)$ and $E_1(2^l t)$ as follows:

$$E(2^l t) = e^{(A+BU_*)2^{l-1}t} E(2^{l-1}t) + E(2^{l-1}t)e^{(A+BU_*)2^{l-1}t},$$

$$E_1(\epsilon, 2^l t) = e^{(A+BU_*)2^{l-1}t} E_1(\epsilon, 2^{l-1}t) + E_1(\epsilon, 2^{l-1}t)e^{(A+BU_*)2^{l-1}t}$$
$$+ \left( E(2^{l-1}t) + \epsilon E_1(\epsilon, 2^{l-1}t) \right)^2.$$

Then we have the relation that $e^{(A+BU_*+B\Delta U)t} - e^{(A+BU_*)t} = \epsilon E(t) + \epsilon^2 E_1(\epsilon, t)$.

Now we upper bound $\|E(t)\|$ and $\|E_1(\epsilon, t)\|$. When $t \le t_0$:

$$\|E(t)\| \le \sum_{k \ge 1} \frac{t^k}{k!} \sum_{i=0}^{k-1} \left\| (A + BU_*)^i (B\Delta U_*)(A + BU_*)^{k-1-i} \right\| \le \sum_{k \ge 1} \frac{1}{(k-1)!} = e$$

For $t \ge t_0$, let $t = 2^{l_1} t_1$, with $l_1$ be an integer and $t_1 \in (\frac{1}{2}t_0, t_0]$, then

$$\|E(2^{l_1} t_1)\| = \left\| e^{(A+BU_*)2^{l_1-1}t_1} E(t) + E(t)e^{(A+BU_*)2^{l_1-1}t_1} \right\|$$
$$\le 2e^{\alpha(A+BU_*)2^{l_1-1}t_1} \left\| E(2^{l_1-1}t_1) \right\|$$
$$\le 2^{l_1} e^{1 + \alpha(A+BU_*)2^{l_1-2}t_0}$$
$$\le \frac{4}{-\alpha(A+BU_*)t_0},$$

where the last inequality is because for any $x, a > 0$, $xe^{-ax} \le \frac{1}{ae}$, and thus for any $t \ge 0$, $\|E(t)\| \le C = \frac{4}{-\alpha(A+BU_*)t_0}$.

When $t \ge \frac{2}{-\alpha(A+BU_*)}$, we additionally have

$$\|E(t)\| \le 2e^{\frac{1}{2}\alpha(A+BU_*)t} \left\| E(\frac{t}{2}) \right\| \le \frac{4t}{t_0} e^{\frac{1}{2}\alpha(A+BU_*)t} \le \frac{8}{-\alpha(A+BU_*)t_0} e^{\frac{1}{4}\alpha(A+BU_*)t}.$$

Now we consider $E_1(\epsilon, t)$. When $t \le t_0$,

$$\|E_1(\epsilon, t)\| \le \sum_{k \ge 1} \left\| \frac{t^k}{k!} D_1(k, \epsilon) \right\| \le 4.$$

When $t > t_0$, with $t = 2^l t_1$ and $t_1 \in (\frac{1}{2} t_0, t_0]$, we obtain:

$$\left\| E_1(\epsilon, 2^l t_1) \right\| =$$

$$\left\| e^{(A+BU_*)2^{l-1}t_1} E_1(\epsilon, 2^{l-1}t_1) + E_1(\epsilon, 2^{l-1}t_1) e^{(A+BU_*)2^{l-1}t_1} + \left( E(2^{l-1}t_1) + \epsilon E_1(\epsilon, 2^{l-1}t_1) \right)^2 \right\|$$

$$\leq 2 e^{\alpha(A+BU_*)2^{l-1}t_1} \left\| E_1(\epsilon, 2^{l-1}t_1) \right\| + \left\| E(2^{l-1}t_1) + \epsilon E_1(\epsilon, 2^{l-1}t_1) \right\|^2$$

$$\leq 2 e^{\alpha(A+BU_*)2^{l-1}t_1} \left\| E_1(\epsilon, 2^{l-1}t_1) \right\| + 2 \left\| E(2^{l-1}t_1) \right\|^2 + 2\epsilon^2 \left\| E_1(\epsilon, 2^{l-1}t_1) \right\|^2 .$$

Now, we show that $\left\| E_1(\epsilon, 2^l t_1) \right\|$ converges exponentially eventually. The proof consists of two parts: first, for $t$ which is not too large, $\|E_1(\epsilon, t)\|$ can be bounded uniformly over all possible $\Delta U$ and any constrained $\epsilon$. Then, for larger $t$ we can utilize the construction of $\|E_1(\epsilon, t)\|$ to estimate its convergence speed.

Let $\epsilon \leq \frac{-\alpha(A+BU_*)t_0}{(64C)^2}$, $l_0 = 1 + \lfloor \log_2 \frac{4}{-\alpha(A+BU_*)t_0} \rfloor$. We first inductively show that for any $l \leq l_0$, $\left\| E_1(\epsilon, 2^l t_1) \right\| \leq (2^{l+3} - 4)C^2$. The base case where $l = 0$ is certainly true. Suppose we already have $\left\| E_1(\epsilon, 2^{l-1}t_1) \right\| \leq (2^{l+2} - 4)C^2$. Then for the case of $l$, we obtain:

$$\left\| E_1(\epsilon, 2^l t_1) \right\| \leq 2 \left\| E_1(\epsilon, 2^{l-1}t_1) \right\| + 4C^2 \leq (2^{l+3} - 4)C^2 ,$$

where for the first inequality we use the inductive hypothesis that

$$\epsilon \left\| E_1(\epsilon, 2^{l-1}t_1) \right\| \leq 2^{l_0+3} C^2 \epsilon \leq \frac{64}{-\alpha(A+BU_*)t_0} C^2 \epsilon \leq C ,$$

along with facts that $\left\| E(2^{l-1}t_1) \right\| \leq C$ and $2 e^{\alpha(A+BU_*)2^{l-1}t_1} \leq 2$. Specifically, we have $\left\| E_1(\epsilon, 2^{l_0}t_1) \right\| \leq \frac{64C^2}{-\alpha(A+BU_*)t_0}$.

Now, we consider $l > l_0$. We first show that for all such $l$, $\left\| E_1(\epsilon, 2^l t_1) \right\| \leq \frac{64C^2}{-\alpha(A+BU_*)t_0}$. Since $2^{l-1}t_1 \geq 2^{l_0-1}t_0 \geq \frac{2}{-\alpha(A+BU_*)}$, we have $2 e^{\alpha(A+BU_*)2^{l-1}t_1} \leq 2e^{-2}$, and thus

$$\left\| E_1(\epsilon, 2^l t_1) \right\| \leq 2 e^{\alpha(A+BU_*)2^{l-1}t_1} \left\| E_1(\epsilon, 2^{l-1}t_1) \right\| + 2 \left\| E(2^{l-1}t_1) \right\|^2 + 2\epsilon^2 \left\| E_1(\epsilon, 2^{l-1}t_1) \right\|^2$$

$$\leq 2e^{-2} \left\| E_1(\epsilon, 2^{l-1}t) \right\| + 4C^2$$

$$\leq \frac{64C^2}{-\alpha(A+BU_*)t_0} ,$$

which holds for all $l \geq l_0$ with induction on $l$. Now we reuse the above expression and obtain that

$$\left\| E_1(\epsilon, 2^l t_1) \right\|$$

$$\leq 2 e^{\alpha(A+BU_*)2^{l-1}t_1} \left\| E_1(\epsilon, 2^{l-1}t_1) \right\| + 2 \left\| E(2^{l-1}t_1) \right\|^2 + 2\epsilon^2 \left\| E_1(\epsilon, 2^{l-1}t_1) \right\|^2$$

$$\leq 2e^{-2^{l-l_0}} \frac{64C^2}{-\alpha(A+BU_*)t_0} + \frac{128}{\alpha^2(A+BU_*)t_0^2} e^{-2^{l-l_0-1}} + 2\epsilon^2 \left\| E_1(\epsilon, 2^{l-1}t_1) \right\|^2 .$$

Let $l_*$ be the smaller integer greater than $l_0 + 1$ which satisfies:

$$2e^{-2^{l_*-l_0}} \frac{64C^2}{-\alpha(A+BU_*)t_0} + \frac{128}{\alpha^2(A+BU_*)t_0^2} e^{-2^{l_*-l_0-1}} \leq \frac{1}{4} .$$

Then by using the relation that $2\epsilon^2 \left\| E_1(\epsilon, 2^{l-1}t_1) \right\|^2 \leq 2\epsilon^2 \left( \frac{64C^2}{-\alpha(A+BU_*)t_0} \right)^2 \leq \frac{1}{4}$, we have:

$$\left\| E_1(\epsilon, 2^{l_*}t_1) \right\| \leq \frac{1}{2} .$$

Now we inductively show that for all $k \geq 0$,

$$\left\| E_1(\epsilon, 2^{l_*+k}t_1) \right\| \leq 2^{-2^k} .$$

By using the hypothesis for $k$ and $2\epsilon^2 \leq \frac{1}{4}$, we obtain:

$$\left\| E_1(\epsilon, 2^{l_*+k+1}t_1) \right\| \leq 2\epsilon^2 \left\| E_1(\epsilon, 2^{l_*+k}t_1) \right\|^2 + \frac{1}{4}e^{-2^{k+l_*-l_0}+2^{l_*-l_0}}$$

$$\leq \frac{1}{4}2^{-2^{k+1}} + \frac{1}{4}e^{-2^{k+2}+2^2}$$

$$\leq 2^{-2^{k+1}},$$

leading to the claim. This means there exist some constants $C_1, c_1 > 0$ depending on $\alpha(A + BU_*)$ such that for all $t \geq 0$, $\|E_1(\epsilon, t)\| \leq C_1 e^{-c_1 t}$.

Finally, we consider $\int_{t\geq 0} e^{(A+BU)^{\mathrm{T}}t}(Q + U^{\mathrm{T}}RU)e^{(A+BU)t}dt$. Since $e^{(A+BU_*+\epsilon\Delta U)t} = e^{(A+BU_*)t} + \epsilon E(t) + \epsilon^2 E_1(\epsilon, t)$, with $\|E(t)\| \leq \frac{8}{-\alpha(A+BU_*)t_0}e^{\frac{1}{4}\alpha(A+BU_*)t}$ and bounded $E_1(\epsilon, t)$, we obtain:

$$\int_{t\geq 0} e^{(A+BU)^{\mathrm{T}}t}(Q + U^{\mathrm{T}}RU)e^{(A+BU)t}dt$$

$$= \int_{t\geq 0} (e^{(A+BU_*)^{\mathrm{T}}t} + \epsilon E^{\mathrm{T}}(t) + \epsilon^2 E_1^{\mathrm{T}}(\epsilon, t))(Q + U^{\mathrm{T}}RU)(e^{(A+BU_*)t} + \epsilon E(t) + \epsilon^2 E_1(\epsilon, t))dt$$

$$= \int_{t\geq 0} e^{(A+BU_*)^{\mathrm{T}}t}(Q + U_*^{\mathrm{T}}RU_*)e^{(A+BU_*)t}dt$$

$$+ \epsilon \int_{t\geq 0} E^{\mathrm{T}}(t)(Q + U_*^{\mathrm{T}}RU_*)e^{(A+BU_*)t} + e^{(A+BU_*)^{\mathrm{T}}t}(Q + U_*^{\mathrm{T}}RU_*)E(t)dt$$

$$+ \epsilon \int_{t\geq 0} e^{(A+BU_*)^{\mathrm{T}}t} \left( \Delta U^{\mathrm{T}}RU_* + U_*^{\mathrm{T}}R\Delta U \right) e^{(A+BU_*)t}dt$$

$$+ \mathcal{O}(\epsilon^2).$$

Where the last term $\mathcal{O}(\epsilon^2)$ contains any terms with order at least $\epsilon^2$, whose norm is at most $C_2\epsilon^2$ for any $\epsilon \in [0, \epsilon_0)$ and $\|\Delta U\| = 1$, where the constant $C_2$ depends on $A, B, \alpha(A + BU_*)$ and $\epsilon_0$ is some small constant.

For any $\|\Delta U\| = 1$, define $V$ by

$$V = \int_{t\geq 0} E^{\mathrm{T}}(t)(Q + U_*^{\mathrm{T}}RU_*)e^{(A+BU_*)t} + e^{(A+BU_*)^{\mathrm{T}}t}(Q + U_*^{\mathrm{T}}RU)E(t)dt$$

$$+ \int_{t\geq 0} e^{(A+BU_*)^{\mathrm{T}}t} \left( \Delta U^{\mathrm{T}}RU_* + U^{\mathrm{T}}R\Delta U \right) e^{(A+BU_*)t}dt,$$

then $cost(U) = cost(U_*) + \epsilon \mathrm{tr}(V) + O(\epsilon^2)$.

Since $U_*$ minimizes $cost(U)$, $\mathrm{tr}(V) = \lim_{\epsilon\to 0} \epsilon^{-1}(cost(U_* + \epsilon\Delta U) - cost(U_*)) = 0$. Therefore, we obtain that $cost(U) = cost(U_*) + \mathcal{O}(\epsilon^2)$.

$\square$

### B.5 PROOF OF LEMMA 18

In this section, we proof Lemma 18.

**Lemma 18.** *Let $U_*$ follows the same definition as in Lemma 17. Then, for some $\epsilon > 0$, there exist constants $C_2$ and $C_3$ (independent of $U$) such that for all $T > 0$ and any $U$ such that $\|U - U_*\| \leq \epsilon$,*

$$|J_T - cost(U)T| \leq C_2\|x\|_2^2 + C_3.$$

*Here $J_T$ is the expected cost of the policy that takes action by $U_t = UX_t$ ($t \in [0, T]$), with initial state $X_0 = x$.*

*Proof.* By definition of $J_T$, we have:

$$J_T = \mathbb{E}\left[\int_{t=0}^T \left(X_t^{\mathrm{T}} Q X_t + U_t^{\mathrm{T}} R U_t\right) dt\right] = \mathbb{E}\left[\int_{t=0}^T X_t^{\mathrm{T}}(Q + U^{\mathrm{T}} R U) X_t dt\right].$$

Since the state transits according to $dX_t = AX_t dt + BUX_t dt + dW_t$, we can derive the expression of $X_t$ by $X_t = e^{(A+BU)t} X_0 + \int_{s=0}^t e^{(A+BU)(t-s)} dW_s$. Then by utilizing this expression we obtain:

$$\mathbb{E}\left[X_t^{\mathrm{T}}(Q + U^{\mathrm{T}} R U) X_t\right]$$
$$= (e^{(A+BU)t} X_0)^{\mathrm{T}}(Q + U^{\mathrm{T}} R U) e^{(A+BU)t} X_0$$
$$+ 2\mathbb{E}\left[(e^{(A+BU)t} X_0)^{\mathrm{T}}(Q + U^{\mathrm{T}} R U)\left(\int_{s=0}^t e^{(A+BU)(t-s)} dW_s\right)\right]$$
$$+ \mathbb{E}\left[\left(\int_{s=0}^t e^{(A+BU)(t-s)} dW_s\right)^{\mathrm{T}}(Q + U^{\mathrm{T}} R U)\left(\int_{s=0}^t e^{(A+BU)(t-s)} dW_s\right)\right]$$
$$= X_0^{\mathrm{T}} e^{(A+BU)^{\mathrm{T}} t}(Q + U^{\mathrm{T}} R U) e^{(A+BU)t} X_0$$
$$+ tr\left(\int_{s=0}^t e^{(A+BU)^{\mathrm{T}} s}(Q + U^{\mathrm{T}} R U) e^{(A+BU)s} ds\right)$$
$$= X_0^{\mathrm{T}} e^{(A+BU)^{\mathrm{T}} t}(Q + U^{\mathrm{T}} R U) e^{(A+BU)t} X_0$$
$$+ \int_{s=0}^t tr\left(e^{(A+BU)^{\mathrm{T}} s}(Q + U^{\mathrm{T}} R U) e^{(A+BU)s}\right) ds.$$

Then, the expected cost on a trajectory lasting for time $T$ can be computed as:

$$\mathbb{E}\left[\int_{t=0}^T X_t^{\mathrm{T}}(Q + U^{\mathrm{T}} R U) X_t dt\right]$$
$$= \int_{t=0}^T \mathbb{E}\left[X_t^{\mathrm{T}}(Q + U^{\mathrm{T}} R U) X_t\right] dt$$
$$= \int_{t=0}^T X_0^{\mathrm{T}} e^{(A+BU)^{\mathrm{T}} t}(Q + U^{\mathrm{T}} R U) e^{(A+BU)t} X_0 dt$$
$$+ \int_{t=0}^T (T - t) tr\left(e^{(A+BU)^{\mathrm{T}} t}(Q + U^{\mathrm{T}} R U) e^{(A+BU)t}\right) dt$$
$$= \int_{t=0}^T X_0^{\mathrm{T}} e^{(A+BU)^{\mathrm{T}} t}(Q + U^{\mathrm{T}} R U) e^{(A+BU)t} X_0 dt + cost(U)T$$
$$- \int_{t=0}^T tr\left(e^{(A+BU)^{\mathrm{T}} t}(Q + U^{\mathrm{T}} R U) e^{(A+BU)t}\right) t dt$$
$$- T \int_{t=T}^{+\infty} tr\left(e^{(A+BU)^{\mathrm{T}} t}(Q + U^{\mathrm{T}} R U) e^{(A+BU)t}\right) dt.$$

Here the first term satisfies

$$\left|\int_{t=0}^T X_0^{\mathrm{T}} e^{(A+BU)^{\mathrm{T}} t}(Q + U^{\mathrm{T}} R U) e^{(A+BU)t} X_0 dt\right| \leq \int_{t\geq 0} e^{2\alpha(A+BU)t} \|X_0\|_2^2 dt$$
$$\leq \frac{1}{-2\alpha(A+BU)} \|X_0\|_2^2,$$

and the latter two integral terms can be bounded as follows.

$$\left| \int_{t=0}^{T} tr\left( e^{(A+BU)^{\mathrm{T}}t}(Q+U^{\mathrm{T}}RU)e^{(A+BU)t} \right) tdt \right|$$

$$\leq \int_{t\geq 0} d \cdot e^{2\alpha(A+BU)t} \left\| Q+U^{\mathrm{T}}RU \right\| tdt$$

$$\leq \frac{d\left\| Q+U^{\mathrm{T}}RU \right\|}{4\alpha^2(A+BU)},$$

$$\left| T \int_{t=T}^{+\infty} tr\left( e^{(A+BU)^{\mathrm{T}}t}(Q+U^{\mathrm{T}}RU)e^{(A+BU)t} \right) dt \right|$$

$$\leq T \int_{t\geq T} d \cdot e^{2\alpha(A+BU)t} \left\| Q+U^{\mathrm{T}}RU \right\| dt$$

$$\leq \frac{Td\left\| Q+U^{\mathrm{T}}RU \right\|}{-2\alpha(A+BU)} e^{2\alpha(A+BU)T}$$

$$\leq \frac{d\left\| Q+U^{\mathrm{T}}RU \right\|}{4\alpha^2(A+BU)}.$$

Therefore, for $C_2 \geq -\frac{1}{2\alpha(A+BU)}$ and $C_3 \geq \frac{d\|Q+U^{\mathrm{T}}RU\|}{2\alpha^2(A+BU)}$, we have

$$|J_T - cost(U)T| \leq C_2\|x\|_2^2 + C_3.$$

□

### B.6    Proof of Lemma 20

Finally, we prove Lemma 20. In this part we suppose $T \geq T_0$, where $T_0 \geq 1$ is a constant depending on some hidden constants and $\|X_0\|_2^2$.

**Lemma 20.** *regret Let $U_t$ be the action applied as in Algorithm 3. Then there exists a constant $C \in poly(\kappa, M, \mu^{-1}, |\alpha(A+BK)|^{-1}, |\alpha(A+BK_*)|^{-1})$ such that for sufficiently large $T$:*

$$\mathbb{E}\left[ \int_{t=0}^{\sqrt{T}} \left( X_t^{\mathrm{T}}QX_t + U_t^{\mathrm{T}}RU_t \right) dt \right] \leq C \cdot \sqrt{T},$$

$$\mathbb{E}\left[ \int_{t=\sqrt{T}}^{T} \left( X_t^{\mathrm{T}}QX_t + U_t^{\mathrm{T}}RU_t \right) dt \right] \leq C \cdot \sqrt{T} + J_T^*.$$

Define the following events where the stabilizing controller $K$ might ever be applied during the exploitation phase. Let $\mathcal{E}_1 = \left\{ \|X_{\sqrt{T}}\|_2 \geq \frac{1}{2}T^{1/5} \right\}$, $\mathcal{E}_2 = \left\{ \|X_t\|_2 \geq T^{1/5} \text{ for some } t \in [\sqrt{T}, T] \right\}$, and $\mathcal{E}_3 = \left\{ \|\bar{K} - K_*\| \leq \epsilon_3 \right\}$, where $\epsilon_3 > 0$ depends on the constant $\epsilon_0$ in Lemma 17, which will be determined later. In this part, we again let $C_1, C_2$ be the same as in equation 31, and denote $C_3$ be the constant $C_1$ in Lemma 17. We firstly analyze these three events.

**Upper bound** $\mathbb{P}[\mathcal{E}_1]$   By Lemma 16, we can find some constant $C_0$ depending on $\|A\|, \|B\|, \|K\|, d, p, h$ such that

$$\mathbb{P}\left[ \|X_{\sqrt{T}}\|_2 \geq C_0\sqrt{\log(2T/\delta)} \right] \leq \delta.$$

This is because we have the recursive function of $\{X_{kh}\}$ that

$$X_{(k+1)h} = e^{(A+BK)h}X_{kh} + \int_{t=0}^{h} e^{(A+BK)(h-t)}dW_{kh+t} + \int_{t=0}^{h} e^{(A+BK)(h-t)}u_k dt,$$

from which we can derive that

$$X_{kh}$$

$$= e^{(A+BK)kh}X_0 + \int_{t=0}^{kh} e^{(A+BK)(kh-t)}dW_t + \sum_{i=0}^{k-1} e^{(A+BK)(k-i-1)h} \left( \int_{t=0}^{h} e^{(A+BK)t}dt \right) u_i \,.$$

Then, for sufficiently large $T$, $\left\| e^{(A+BK)\sqrt{T}}X_0 \right\|_2$ can be bounded by 1, and from the proof in Lemma 16 we can apply similar idea to upper bound the norm of the last two terms. So we can obtain the probability bound on $\|X_{\sqrt{T}}\|_2$.

By setting $\delta = 2T \cdot e^{-\frac{T^{1/5}}{4C_0^2}}$, we obtain that $\mathbb{P}[\mathcal{E}_1] \leq 2T \cdot e^{-\frac{T^{1/5}}{4C_0^2}}$.

**Upper bound $\mathbb{P}[\mathcal{E}_3^C]$**   By equation 30, we obtain that, for $\epsilon_3 \leq \frac{C_1\epsilon_1}{T^{1/8}4C_2^2}$, we have:

$$\mathbb{P}\left[ \|\bar{K} - K_*\| \geq x \right] \leq e^{-\frac{T^{1/2}x^2}{4C_1^2C_2^2}} \; \forall x \leq \epsilon_3 \,,$$

and we also have: $\mathbb{P}[\mathcal{E}_3^C] \leq e^{-\frac{T^{1/2}\epsilon_3^2}{4C_1^2C_2^2}}$.

By setting $\epsilon_3 = \frac{C_1\epsilon_1}{T^{1/8}4C_2^2}$, we have: $\mathbb{P}[\mathcal{E}_3^C] \leq e^{-\frac{T^{1/4}\epsilon_1^2}{64C_2^2}}$.

**Upper bound $\mathbb{P}[\mathcal{E}_2]$**   Consider any $\|X_{\sqrt{T}}\|_2 \leq \frac{1}{2}T^{1/5}$ and any $\|\bar{K} - K_*\| \leq \epsilon_3$, we claim that $\mathbb{P}\left[ \mathcal{E}_2 | X_{\sqrt{T}}, \bar{K} \right] \leq e^{-\Omega(T^{1/5})}$.

As what have discussed in Lemma 14 (see the discussion about stable margin near equation 29), such $\bar{K}$ satisfies $\alpha(A + B\bar{K}) \leq \frac{1}{2}\alpha(A + BK_*)$.

Then by Lemma 16 we can derive that, for some constant $C$,

$$\mathbb{P}\left[ \sup_{t\in[\sqrt{T},T]} \|X_t\|_2 - \|X_{\sqrt{T}}\|_2 \leq \frac{1}{2}T^{1/5} \right] \leq CTe^{-\frac{T^{1/5}}{C}} \leq e^{-\Omega(T^{1/5})} \,.$$

Therefore,

$$\mathbb{P}\left[ \mathcal{E}_2 \right] \leq 1 - \mathbb{P}\left[ \mathcal{E}_1^C \cap \mathcal{E}_3 \right] + e^{-\Omega(T^{1/5})}\mathbb{P}\left[ \mathcal{E}_1^C \cap \mathcal{E}_3 \right]$$

$$\leq \mathbb{P}\left[ \mathcal{E}_1 \right] + \mathbb{P}\left[ \mathcal{E}_3^C \right] + e^{-\Omega(T^{1/5})}$$

$$\leq e^{-\Omega(T^{1/5})} \,.$$

Now we come to estimate the expected cost of Algorithm 3, as well as bound the regret. We separately calculate the cost during the two phases.

**Cost During Exploration Phase**   For $(k+1)h \leq \sqrt{T}$ and $t \in [0, h]$, we have:

$$X_{kh+t} = e^{(A+BK)t}X_{kh} + \int_{s=kh}^{kh+t} e^{(A+BK)(kh+t-s)}dW_s + \left( \int_{s=0}^{t} e^{(A+BK)s}ds \right) u_k \,.$$

Then

$$\mathbb{E}\left[ X_{kh+t}^{\mathrm{T}}QX_{kh+t} + U_{kh+t}^{\mathrm{T}}RU_{kh+t} \right]$$

$$= \mathbb{E}\left[ X_{kh+t}^{\mathrm{T}}(Q + K^{\mathrm{T}}RK)X_{kh+t} + u_k^{\mathrm{T}}Ru_k \right] + 2\mathbb{E}\left[ X_{kh+t}^{\mathrm{T}}K^{\mathrm{T}}Ru_k \right]$$

$$\leq \mathbb{E}\left[ X_{kh+t}^{\mathrm{T}}(Q + K^{\mathrm{T}}RK)X_{kh+t} + u_k^{\mathrm{T}}Ru_k \right]$$

$$+ 2\mathbb{E}\left[ u_k^{\mathrm{T}} \left( \int_{s=0}^{t} e^{(A+BK)s}ds \right)^{\mathrm{T}} K^{\mathrm{T}}Ru_k \right] \,,$$

where the inequality is because $u_k$ is independent of $X_{kh}$ and $W_s(s \in [kh, kh + t])$.

For the first term, we first upper bound $\mathbb{E}\left[\|X_{kh+t}\|_2^2\right]$.

Denote $w_{k,t} = \int_{s=kh}^{kh+t} e^{(A+BK)(kh+t-s)}dW_s + \left(\int_{s=0}^{t} e^{(A+BK)s}ds\right) u_k$, which is a Gaussian variable with zero mean and is independent of $X_{kh}$. Then

$$\mathbb{E}\left[\|X_{kh+t}\|_2^2\right] = \mathbb{E}\left[\left\|e^{(A+BK)t}X_{kh} + w_{k,t}\right\|_2^2\right]$$

$$= \mathbb{E}\left[\left\|e^{(A+BK)t}X_{kh}\right\|_2^2\right] + \mathbb{E}\left[\|w_{k,t}\|_2^2\right]$$

$$\leq \mathbb{E}\left[\|X_{kh}\|_2^2\right] + \mathbb{E}\left[\|w_{k,t}\|_2^2\right] .$$

For $\mathbb{E}\left[\|X_{kh}\|_2^2\right]$, since

$$X_{kh}$$
$$= e^{(A+BK)h}X_0 + \int_{t=0}^{kh} e^{(A+BK)(kh-t)}dW_t + \sum_{i=0}^{k-1} e^{(A+BK)(k-i-1)h}\left(\int_{t=0}^{h} e^{(A+BK)t}dt\right) u_i .$$

We have:

$$\mathbb{E}\left[\|X_{kh}\|_2^2\right]$$
$$= \left\|e^{(A+BK)kh}X_0\right\|_2^2$$
$$+ \mathbb{E}\left[\left\|\int_{t=0}^{kh} e^{(A+BK)(kh-t)}dW_t\right\|_2^2\right]$$
$$+ \sum_{i=0}^{k-1} \mathbb{E}\left[\left\|e^{(A+BK)(k-i-1)h}\left(\int_{t=0}^{h} e^{(A+BK)t}dt\right) u_i\right\|_2^2\right]$$
$$\leq e^{2\alpha(A+BK)\cdot kh}\|X_0\|_2^2$$
$$+ tr\left(\int_{t=0}^{kh} e^{(A+BK)t}e^{(A+BK)^{\mathrm{T}}t}dt\right)$$
$$+ \sum_{i=0}^{k-1} tr\left(\left[e^{(A+BK)ih}\left(\int_{t=0}^{h} e^{(A+BK)t}dt\right)\right]\left[e^{(A+BK)ih}\left(\int_{t=0}^{h} e^{(A+BK)t}dt\right)\right]^{\mathrm{T}}\right)$$
$$\leq e^{2\alpha(A+BK)\cdot kh}\|X_0\|_2^2 + \int_{t=0}^{kh} d \cdot e^{2\alpha(A+BK)t}dt + \sum_{i=0}^{k-1} d \cdot e^{2\alpha(A+BK)ih} \cdot h^2$$
$$\leq e^{2\alpha(A+BK)\cdot kh}\|X_0\|_2^2 + \frac{d}{-2\alpha(A+BK)} + \frac{dh^2}{1-e^{2\alpha(A+BK)h}}$$
$$\leq C_3 + e^{2\alpha(A+BK)\cdot kh}\|X_0\|_2^2 .$$

Where $C_3$ is a constant depending on $\alpha(A+BK)$ and $d$.

For the second term $\mathbb{E}\left[\|w_{k,t}\|_2^2\right]$, can follow the same process of the above bound and obtain $\mathbb{E}\left[\|w_{k,t}\|_2^2\right] \leq C_3$. Therefore, $\mathbb{E}\left[\|X_{kh+t}\|_2^2\right] \leq 2C_3$.

Now we can upper bound $\mathbb{E}\left[X_{kh+t}^{\mathrm{T}}QX_{kh+t} + U_{kh+t}^{\mathrm{T}}RU_{kh+t}\right]$. Since

$$\mathbb{E}\left[X_{kh+t}^{\mathrm{T}}(Q+K^{\mathrm{T}}RK)X_{kh+t}\right] \leq \mathbb{E}\left[\|Q+K^{\mathrm{T}}RK\|\|X_{kh+t}\|_2^2\right]$$
$$\leq \|Q+K^{\mathrm{T}}RK\|\mathbb{E}\left[\|X_{kh+t}\|_2^2\right] ,$$

$$\mathbb{E}\left[u_k^{\mathrm{T}} R u_k\right] = tr(R) \,,$$

$$\mathbb{E}\left[u_k^{\mathrm{T}} \left(\int_{s=0}^{t} e^{(A+BK)s} ds\right)^{\mathrm{T}} K^{\mathrm{T}} R u_k\right]$$

$$\leq (d+p) \cdot \left\|\left(\int_{s=0}^{t} e^{(A+BK)s} ds\right)^{\mathrm{T}} K^{\mathrm{T}} R\right\|$$

$$\leq (d+p)h\|KR\| \,.$$

We can conclude that there exists constant $C_4$ depending on $A, B, K, Q, R, d, p, h$ such that

$$\mathbb{E}\left[X_{kh+t}^{\mathrm{T}} Q X_{kh+t} + U_{kh+t}^{\mathrm{T}} R U_{kh+t}\right] \leq C_4 \left(1 + e^{2\alpha(A+BK)\cdot(kh+t)}\|X_0\|_2^2\right) , \forall k, t$$

Then the cost during exploration phase can be bounded as

$$\mathbb{E}\left[\int_{t=0}^{\sqrt{T}} \left(X_{kh+t}^{\mathrm{T}} Q X_{kh+t} + U_{kh+t}^{\mathrm{T}} R U_{kh+t}\right) dt\right] \leq C_4 \left(\sqrt{T} + \frac{\|X_0\|_2^2}{-2\alpha(A+BK)}\right) . \tag{34}$$

**Cost During Exploitation Phase**

**Upper Bound of the Cost when $\mathcal{E}_2$ happens**   We first concentrate on $\mathcal{E}_2$, which is the hardest event for the analysis of the cost. Consider the following two cases:

**Case 1:** $\|X_{\sqrt{T}}\|_2 \geq T^{1/5}$. In this case, the action is applied by $U_t = KX_t, t \in [\sqrt{T}, T]$.

**Case 2:** $\|X_{\sqrt{T}}\|_2 < T^{1/5}$. In this case, the trajectory is unfortunately controlled by a bad controller, and suffers from large risk of diverging.

We first consider **Case 1**. By equation 7 we can derive that

$$X_t = e^{(A+BK)(t-\sqrt{T})} X_{\sqrt{T}} + \int_{s=\sqrt{T}}^{t} e^{(A+BK)(t-s)} dW_s \,.$$

Then, we have:

$$\mathbb{E}\left[X_t^{\mathrm{T}} Q X_t + U_t^{\mathrm{T}} R U_t\right]$$
$$= \mathbb{E}\left[X_t^{\mathrm{T}} (Q + K^{\mathrm{T}} R K) X_t\right]$$
$$\leq \|Q + K^{\mathrm{T}} R K\| \mathbb{E}\left[\|X_t\|_2^2\right]$$
$$\leq \|Q + K^{\mathrm{T}} R K\| \left[\|X_{\sqrt{T}}\|_2^2 + \int_{s=\sqrt{T}}^{t} tr\left(e^{(A+BK)(t-s)} e^{(A+BK)^{\mathrm{T}}(t-s)}\right) dt\right]$$
$$\leq \|Q + K^{\mathrm{T}} R K\| \left[\|X_{\sqrt{T}}\|_2^2 + \int_{s=\sqrt{T}}^{t} d \cdot e^{2\alpha(A+BK)(t-s)} dt\right] \,.$$

Therefore, for some constants $C_5, C_6$, we have:

$$\mathbb{E}\left[X_t^{\mathrm{T}} Q X_t + U_t^{\mathrm{T}} R U_t\right] \leq C_5 \|X_{\sqrt{T}}\|_2^2 + C_6 \,.$$

Now we consider **Case 2**. Let $t_0 = \inf_t\{\|X_t\|_2 \geq T^{1/5}, t \geq \sqrt{T}\}$, then $\|X_{t_0}\|_2 = T^{1/5}$ almost surely.

For $t \in [\sqrt{T}, t_0]$, since we always have

$$\|U_t\|_2 \leq \max\left\{\|K\|, \|R^{-1} B^{\mathrm{T}} P\|\right\} \|X_t\|_2 \leq \left(\|K\| + \|R^{-1} B^{\mathrm{T}}\| T^{1/5}\right) T^{1/5} \,,$$

the cost satisfies:

$$X_t^{\mathrm{T}} Q X_t + U_t^{\mathrm{T}} R U_t \leq C_7 T^{4/5} \,.$$

Where $C_7$ is a constant depending on $B, R, K, P$.

For $t \in [t_0, T]$, the trajectory $X_t$ satisfies

$$X_t = e^{(A+BK)(t-t_0)} X_{t_0} + \int_{s=t_0}^{t} e^{(A+BK)(t-s)} dW_s \,.$$

Similar to the analysis for **Case 1**, we have:

$$\mathbb{E}\left[ X_t^{\mathrm{T}} Q X_t + U_t^{\mathrm{T}} R U_t \right] \leq C_5 T^{2/5} + C_6$$

Combining them, we can conclude that for some constant $C_8$, no matter whether $\mathcal{E}_2$ happens, we always have:

$$\mathbb{E}\left[ X_t^{\mathrm{T}} Q X_t + U_t^{\mathrm{T}} R U_t \right] \leq C_8 \left[ T^{4/5} + \|X_{\sqrt{T}}\|_2^2 \right] \; \forall t \in [\sqrt{T}, T] \,.$$

Now we establish the upper bound for the regret. Since

$$1 = 1_{\mathcal{E}_1^C \cap \mathcal{E}_3} + 1_{\mathcal{E}_1} + 1_{\mathcal{E}_1^C \cap \mathcal{E}_3^C}$$

Then we can rewrite $\mathbb{E}\left[ \int_{t=\sqrt{T}}^{T} \left( X_t^{\mathrm{T}} Q X_t + U_t^{\mathrm{T}} R U_t \right) dt \right]$ as

$$\mathbb{E}\left[ \int_{t=\sqrt{T}}^{T} \left( X_t^{\mathrm{T}} Q X_t + U_t^{\mathrm{T}} R U_t \right) dt \right]$$

$$= \mathbb{E}\left[ \int_{t=\sqrt{T}}^{T} \left( X_t^{\mathrm{T}} Q X_t + U_t^{\mathrm{T}} R U_t \right) dt \cdot 1_{\mathcal{E}_1^C \cap \mathcal{E}_3} \right]$$

$$+ \mathbb{E}\left[ \int_{t=\sqrt{T}}^{T} \left( X_t^{\mathrm{T}} Q X_t + U_t^{\mathrm{T}} R U_t \right) dt \cdot 1_{\mathcal{E}_1} \right]$$

$$+ \mathbb{E}\left[ \int_{t=\sqrt{T}}^{T} \left( X_t^{\mathrm{T}} Q X_t + U_t^{\mathrm{T}} R U_t \right) dt \cdot 1_{\mathcal{E}_1^C \cap \mathcal{E}_3^C} \right] \,.$$

For the first term, we can upper bound it by

$$\mathbb{E}\left[ \int_{t=\sqrt{T}}^{T} \left( X_t^{\mathrm{T}} Q X_t + U_t^{\mathrm{T}} R U_t \right) dt \cdot 1_{\mathcal{E}_1^C \cap \mathcal{E}_3} \right]$$

$$\leq \mathbb{E}\left[ \int_{t=\sqrt{T}}^{T} \left( X_t^{\mathrm{T}} Q X_t + U_t^{\mathrm{T}} R U_t \right) dt \cdot 1_{\mathcal{E}_1^C \cap \mathcal{E}_2^C \cap \mathcal{E}_3} \right]$$

$$+ \mathbb{E}\left[ \int_{t=\sqrt{T}}^{T} \left( X_t^{\mathrm{T}} Q X_t + U_t^{\mathrm{T}} R U_t \right) dt \cdot 1_{\mathcal{E}_1^C \cap \mathcal{E}_2} \right]$$

$$\leq \mathbb{E}\left[ \left( cost \left( R^{-1} B^{\mathrm{T}} P \right) T + C_9 \|X_{\sqrt{T}}\|_2^2 \right) \cdot 1_{\mathcal{E}_1^C \cap \mathcal{E}_3} \right] + \mathbb{E}\left[ C_8 \left( T^{4/5} + \|X_{\sqrt{T}}\|_2^2 \right) \cdot 1_{\mathcal{E}_1^C \cap \mathcal{E}_2} \right]$$

$$\leq C_9 T^{2/5} + cost(R^{-1} B^{\mathrm{T}} P_*) T + C_{10} T \mathbb{E}\left[ \|\bar{K} - K_*\|^2 \cdot 1_{\mathcal{E}_3} \right] + 2 C_8 T^{4/5} \cdot \mathbb{E}\left[ 1_{\mathcal{E}_1^C \cap \mathcal{E}_2} \right] \,.$$

Here the first inequality is because $1_{\mathcal{E}_1^C \cap \mathcal{E}_3} = 1_{\mathcal{E}_1^C \cap \mathcal{E}_2^C \cap \mathcal{E}_3} + 1_{\mathcal{E}_1^C \cap \mathcal{E}_2 \cap \mathcal{E}_3}$ and $1_{\mathcal{E}_1^C \cap \mathcal{E}_2 \cap \mathcal{E}_3} \leq 1_{\mathcal{E}_1^C \cap \mathcal{E}_3}$. For the second inequality, the first term is because we can assume a situation that we do not change the dynamic when $\mathcal{E}_2$ happens, and that will not make the expectation smaller. By applying the results of Lemma 17 and Lemma 18 we can get this term, where the constant $C_9$ is related to constants in these two lemmas. The last inequality is obtained from these two lemmas and the definitions of $\mathcal{E}_1, \mathcal{E}_2, \mathcal{E}_3$.

As for $\mathbb{E}\left[ \|\bar{K} - K_*\|^2 \cdot 1_{\mathcal{E}_3} \right]$, we use the bound that

$$\mathbb{P}\left[ \|\bar{K} - K_*\| \geq x \right] \leq e^{-\frac{T^{1/2} x^2}{4 C_1^2 C_2^2}} \; \forall x \leq \epsilon_3 \,,$$

and compute that

$$\mathbb{E}\left[\|\bar{K} - K_*\|^2 \cdot 1_{\mathcal{E}_3}\right]$$

$$\leq \int_{x=0}^{\epsilon_3^2} \mathbb{P}\left[\|\bar{K} - K_*\|^2 \geq x\right] \cdot dx$$

$$\leq \int_{x\geq 0} e^{-\frac{T^{1/2}x}{4C_1^2 C_2^2}} dx$$

$$= \frac{4C_1^2 C_2^2}{T^{1/2}}.$$

For $\mathbb{E}\left[1_{\mathcal{E}_1^C \cap \mathcal{E}_2}\right]$, we directly have $\mathbb{E}\left[1_{\mathcal{E}_1^C \cap \mathcal{E}_2}\right] \leq \mathbb{P}\left[\mathcal{E}_2\right] \leq e^{-\Omega(T^{1/5})}$. Combining these results and Lemma 18 we obtain that for some constant $C$,

$$\mathbb{E}\left[\int_{t=\sqrt{T}}^{T} \left(X_t^{\mathrm{T}} Q X_t + U_t^{\mathrm{T}} R U_t\right) dt \cdot 1_{\mathcal{E}_1^C \cap \mathcal{E}_3}\right] \leq J_{\theta_*,T} + C\sqrt{T}.$$

For the second term $\mathbb{E}\left[\int_{t=\sqrt{T}}^{T} \left(X_t^{\mathrm{T}} Q X_t + U_t^{\mathrm{T}} R U_t\right) dt \cdot 1_{\mathcal{E}_1}\right]$, given any $X_{\sqrt{T}}$, we always have

$$\mathbb{E}\left[X_t^{\mathrm{T}} Q X_t + U_t^{\mathrm{T}} R U_t\right] \leq C_8 \left[T^{4/5} + \|X_{\sqrt{T}}\|_2^2\right] \ \forall t \in [\sqrt{T}, T].$$

So we can upper bound $\mathbb{E}\left[\int_{t=\sqrt{T}}^{T} \left(X_t^{\mathrm{T}} Q X_t + U_t^{\mathrm{T}} R U_t\right) dt \cdot 1_{\mathcal{E}_1}\right]$ by

$$\mathbb{E}\left[\int_{t=\sqrt{T}}^{T} \left(X_t^{\mathrm{T}} Q X_t + U_t^{\mathrm{T}} R U_t\right) dt \cdot 1_{\mathcal{E}_1}\right]$$

$$\leq C_8 T^{9/5} \mathbb{P}[\mathcal{E}_1] + C_8 T \mathbb{E}\left[\|X_{\sqrt{T}}\|_2^2 \cdot 1_{\mathcal{E}_1}\right]$$

$$\leq O(1) + C_8 T \mathbb{E}\left[\|X_{\sqrt{T}}\|_2^2 \cdot 1_{\mathcal{E}_1}\right],$$

where for the last inequality we apply the upper bound of $\mathbb{P}[\mathcal{E}_1]$ shown before.

For $\mathbb{E}\left[\|X_{\sqrt{T}}\|_2^2 \cdot 1_{\mathcal{E}_1}\right]$, we can apply Lemma 16 and obtain that for some constant $c > 0$, for any $x \geq \frac{1}{2} T^{1/5}$, we have

$$\mathbb{P}\left[\|X_{\sqrt{T}}\|_2 \geq x\right] \leq e^{-cx^2}.$$

Thus we have:

$$T\mathbb{E}\left[\|X_{\sqrt{T}}\|_2^2 \cdot 1_{\mathcal{E}_1}\right]$$

$$\leq \frac{1}{4} T^{7/5} \mathbb{P}\left[\|X_{\sqrt{T}}\|_2 \geq \frac{1}{2} T^{1/5}\right] + T \int_{x \geq \frac{1}{4} T^{2/5}} \mathbb{P}\left[\|X_{\sqrt{T}}\|_2^2 \geq x\right] dx$$

$$\leq \mathcal{O}(1).$$

Therefore, we have $\mathbb{E}\left[\int_{t=\sqrt{T}}^{T} \left(X_t^{\mathrm{T}} Q X_t + U_t^{\mathrm{T}} R U_t\right) dt \cdot 1_{\mathcal{E}_1}\right] \leq \mathcal{O}(1)$

Finally, for the last term $\mathbb{E}\left[\int_{t=\sqrt{T}}^{T} \left(X_t^{\mathrm{T}} Q X_t + U_t^{\mathrm{T}} R U_t\right) dt \cdot 1_{\mathcal{E}_1^C \cap \mathcal{E}_3^C}\right]$, when condition on any $\|X_{\sqrt{T}}\|_2 \leq \frac{1}{2} T^{1/5}$, estimator $(\hat{A}, \hat{B})$ and $X_{t_0}$, where $t_0 = \inf_{t \geq \sqrt{T}}(\|X_t\|_2 \geq T^{1/5})$, we still have:

$$\mathbb{E}\left[X_t^{\mathrm{T}} Q X_t + U_t^{\mathrm{T}} R U_t\right] \leq C_8 \left[T^{4/5} + \|X_{\sqrt{T}}\|_2^2\right] \leq 2C_8 T^{4/5}, \forall t \in [\sqrt{T}, T].$$

So we can upper bound it by

$$\mathbb{E}\left[\int_{t=\sqrt{T}}^{T} \left(X_t^{\mathrm{T}} Q X_t + U_t^{\mathrm{T}} R U_t\right) dt \cdot 1_{\mathcal{E}_1^C \cap \mathcal{E}_3^C}\right]$$

$$\leq 2C_8 T^{9/5} \mathbb{P}\left[\mathcal{E}_1^C \cap \mathcal{E}_3^C\right]$$

$$\leq 2C_8 T^{9/5} \mathbb{P}[\mathcal{E}_3^C]$$

$$\leq \mathcal{O}(1).$$

Combining them we finally obtain Lemma 20.

