# OpenReview forum: "Finite Sample Analyses for Continuous-time Linear Systems: System Identification and Online Control"
_ICLR.cc/2025/Conference — Submitted to ICLR 2025_

### Official Review · Reviewer_vfoN · 2024-10-30

**Soundness:** 3
**Presentation:** 2
**Contribution:** 3
**Rating:** 6
**Confidence:** 3

**Summary:**

This paper studies the system identification and control of continuous-time linear systems.  The three main contributions are (i) a system ID method that achieves optimal performance (up to constant factors) when the system is stabilized by a known controller, (ii) an approach for identification via N trajectories in the case when a stable controller is not available, and (iii) an approach that learns to control an unknown system assuming a stabilizing controller is known in advance.   This final contribution is the most exciting result to me, and eliminates an annoying log factor in the current state of the art.

**Strengths:**

Learning to control an unknown system has been a rich and exciting line of research in recent years.  This paper contributes to it by pushing the theory in the continuous-time setting, whereas most work has considered discrete systems.  This is an important direction for the field and the paper achieves optimal (up to constant factor) results in the context where a stabilizing controller is known.  This is a clear contribution to the literature.

The key ideas underlying the analysis include a novel discretization of the continuous-time system which allows adaptation of recent tools developed for the analysis of discrete time systems.  I expect that this discretization will allow others to build on the work in the paper, studying more general settings.

**Weaknesses:**

The paper is closely related to a long line of recent papers and does not differentiate itself clearly to show the size of the technical contribution in the work.  This leads to some questions detailed in the next section.

Given the discrete time results in the literature, one may hope that the assumption of a known stabilizing controller would not be needed for the online control algorithm.  This assumption has been removed in the discrete-time setting (e.g. in Kargin, Lale, et al Thompson Sampling Achieves $O(\sqrt{T})$ Regret in Linear Quadratic Control and the papers that follow that work.

Section 5.2 provides a detailed comparison to the line of work of Faradonbeh et al.  There is clearly a difference, however the delta does not seem to be large.  They previous work uses a slightly stronger assumption and the algorithm is different.  Removing the annoying log factor is a clear contribution, but the technical and algorithmic contribution is not as clear.

The results in the case where a stabilizing controller is not known are weaker than the other two main contributions.  They do not seem to provide a practical algorithm or novel technical tool.

Empirical evaluation of the proposed algorithms is not provided. Such results would strengthen the impact of the theory.

**Questions:**

What are the major technical challenges in continuous time (or in your reduction) when transitioning results on control of discrete time systems without an initial stabilizing controller to continuous time?

Beyond the novel discretization, what technical challenges (and novel technical ideas) where needed when adapting the discrete time tools to continuous time?

---

> ### Author Response · Authors · 2024-11-21
> **Comments by Authors**
>
> We extend our gratitude to the reviewer for the constructive suggestions and questions. We **have added relevant experiments and clarified the main novelty and contributions of the paper**. We address the reviewer's concerns belows:
>
> >Q1: Given the discrete time results in the literature, one may hope that the assumption of a known stabilizing controller would not be needed for the online control algorithm. This assumption has been removed in the discrete-time setting.
>
> **A1:** Thank you for the insightful comment. While existing methods like Thompson sampling[1] have removed this assumption, they face another issue: **the state is not well-constrained during the early exploration phase before a stabilizing controller is identified**. Instead of relying on a single trajectory without a stabilizing controller, we prefer to first find a stabilizing controller using multiple trajectories, as shown in Algorithm 2, and then proceed with single-trajectory online control.
>
> >Q2: Section 5.2 provides a detailed comparison to the line of work of Faradonbeh et al. There is clearly a difference, however the delta does not seem to be large. The previous work uses a slightly stronger assumption and the algorithm is different. Removing the annoying log factor is a clear contribution, but the technical and algorithmic contribution is not as clear.
>
> **A2:** Thank you for pointing out this question. We have updated Section 5 to emphasize the differences and contributions more clearly. The modifications have been highlighted and marked in blue for clarity in the updated version.
>
>
> >Q3: The results in the case where a stabilizing controller is not known are weaker than the other two main contributions. They do not seem to provide a practical algorithm or novel technical tool.
>
> **A3:** We acknowledge that this contribution is relatively weaker compared to the other two. This result is primarily a straightforward application of the system identification method. Our main goal in proposing this algorithm is to provide an efficient approach for obtaining a stabilizer.
>
>
> >Q4: Empirical evaluation of the proposed algorithms is not provided. Such results would strengthen the impact of the theory.
>
> **A4:** We appreciate the reviewer's suggestions. In response, **we have added an experimental section in Section 5.3 of the paper**, comparing our algorithm with the baseline algorithm. Please refer to the latest PDF version for details.
>
>
> >Q5: What are the major technical challenges in continuous time (or in your reduction) when transitioning results on control of discrete time systems without an initial stabilizing controller to continuous time?
>
> **A5:** Thank you for the question. The primary technical challenge lies in providing a sharp bound on the early exploration phase rather than in transitioning the results. While methods like Thompson sampling [1] can work with sufficient exploration, the state **may experience rapid divergence during early exploration**, making these methods impractical. To address this, we adopt a simpler and more efficient approach: using multiple trajectories to first find a stabilizer and then applying control.
>
>
> >Q6: Beyond the novel discretization, what technical challenges (and novel technical ideas) where needed when adapting the discrete time tools to continuous time?
>
> **A6:** Thank you for pointing out this question. The technical novelty lies in **finding the exact reverse $Ah$ of the matrix exponential $e^{Ah}$**. Two key novel ideas address this challenge. First, we identify a domain $\|X\|\leq \frac{1}{4}\$ where $f(X)=e^{X}$ is one-to-one, and the distance between $f(X_1)$ and $f(X_2)$ is lower bounded by $\frac{1}{2}\|X_1-X_2\|$, as shown in Lemma 7 in our paper. This property enables the analysis of estimation errors. Second, we apply Taylor expansion to compute the estimator. Detailed descriptions are highlighted in blue in Section 4.
>
>
> Finally, we thank the reviewer once again for the effort in providing us with valuable suggestions. We will continue to provide clarifications if the reviewer has any further questions.
>
> **References**
>
> [1] Shirani Faradonbeh, M. K., Shirani Faradonbeh, M. S., & Bayati, M. (2022). Thompson sampling efficiently learns to control diffusion processes. Advances in Neural Information Processing Systems, 35, 3871-3884.

---

> > ### Comment · Reviewer_vfoN · 2024-11-22
> >
> > Thank you for the response and the added experimental section.  I will maintain my positive score.

---

> > > ### Author Response · Authors · 2024-11-23
> > > **Thank you!**
> > >
> > > We thank the reviewer for acknowledging our work!

---

### Official Review · Reviewer_NsJr · 2024-11-03

**Soundness:** 3
**Presentation:** 2
**Contribution:** 3
**Rating:** 6
**Confidence:** 3

**Summary:**

This paper presents the first $\mathcal{O}(\sqrt{T})$ regret LQ continuous control algorithm. Besides, it shows its lower bound ($\sqrt{T}$) matching the upper bound, which means this algorithm nearly optimal (the regret is tight).  The algorithms seems novel.

**Strengths:**

1. The first continuous LQ algorithm achieving $\mathcal{O}(\sqrt{T})$ regret.
2. This algorithm is nearly optimal (the upper bound matches lower bound).

**Weaknesses:**

1. There are no empirical results presented in paper. Could you perform the experiment to compare your algorithms with (Thompson sampling efficiently learns to control diffusion processes)?
2. I thought in Algorithm 3 line 435 (for $k=0,....\frac{\sqrt{T}}{h}-1$), the $\\frac{\sqrt{T}}{h}-1$ is not an integer in most cases, this should be fixed.
3. In page 6 $\textbf{Summary of Notations}$, there are some typos $\textbf{??}$ in 4 and 5.
4. In Abstract 'Our system identification method allows us explore more efficiently' to 'Our system identification method allows us to explore more efficiently'
5. Can you check the $\textbf{Lemma 12}$ in page 17, you should make your subscript more clear, don not always use $k$ as subscript.

**Questions:**

Also I confess that I have not checked each mathematical conduction presented in the paper in details since this paper is mathematical heavily.

This algorithm can not be applied in nonstochastic settings or adversarial settings since in exploration stage the action is perturbed by gaussian noises, right?

---

> ### Author Response · Authors · 2024-11-21
> **Comments by Authors**
>
> We express our gratitude to the reviewer for the insightful comments. We address the reviewer's questions in more detail as follows:
>
>
> >Q1: There are no empirical results presented in paper.
>
> **A1:** We appreciate the reviewer's suggestions. In response, **we have added an experimental section in Section 5.3 of the paper**, comparing our algorithm with the baseline algorithm. Please refer to the latest PDF version for details.
>
>
>
> >Q2: There are typos in Algorithm 3, line 435, on page 6 (in steps 4 and 5), and in the Abstract.
>
>
> **A2:** We sincerely thank the reviewer for pointing out the typos. These have been corrected in the updated version, with changes highlighted in blue.
>
>
> >Q3: Also I confess that I have not checked each mathematical conduction presented in the paper in details since this paper is mathematical heavily.
>
> **A3:** Thank you for regarding the mathematical details. To address this, we have made some modifications in the appendix to improve clarity and ensure the mathematical derivations are easier to follow.
>
> >Q4: This algorithm can not be applied in nonstochastic settings or adversarial settings since in exploration stage the action is perturbed by gaussian noises.
>
> **A4:** We thank the reviewer for the insightful comment. Our primary focus is on system identification and regret analysis for continuous-time linear systems with stochastic noise. Addressing nonstochastic noise is indeed a more challenging problem. Current research of nonstochastic noise **has only achieved sublinear regret for discrete-time systems[1][2], and not for continuous-time systems**. We consider this a promising direction for future work.
>
>
>
> We thank the reviewer once again for the valuable and helpful suggestions. We would be happy to provide further clarifications if the reviewer has any additional questions.
>
>
> **References**
>
> [1] Agarwal, Naman, et al. "Online control with adversarial disturbances." International Conference on Machine Learning. PMLR, 2019.
>
> [2] Hazan, Elad, Sham Kakade, and Karan Singh. "The nonstochastic control problem." Algorithmic Learning Theory. PMLR, 2020.

---

> > ### Comment · Reviewer_NsJr · 2024-11-25
> > **Reply to authors**
> >
> > I thank the authors detailed explainations. After double checking the proofs of the paper and look at the details of the proposed method. I have more confidence about the novelty of the method as well as the technical maturity presented in the paper. I prefer to raise my score up to 6. However, as recently I am not active in the area of learning to control, I am not sure the progress of the whole community, I keep my confidence 3 still.

---

> > > ### Author Response · Authors · 2024-11-25
> > > **Thank you!**
> > >
> > > We thank the reviewer for acknowledging our work!

---

### Official Review · Reviewer_isKP · 2024-11-04

**Soundness:** 2
**Presentation:** 2
**Contribution:** 3
**Rating:** 5
**Confidence:** 4

**Summary:**

The paper studies the problem of control in a continuous time setting, where the state/environment varies by a stochastic process. The authors suggest a method of discrete sampling that involves a mixture of system identification and discarding non-stable controllers, allowing a regret of O(sqrt(T)), matching the best-known bounds in the literature.

**Strengths:**

1) Addresses a weakness in the literature of approximating $X_{t+\epsilon}$, (could add references for readers, line 227), but then proposes a solution that involves preserving bijections between matrix exponentials.
2) Nice ideas in section 4.2 involving system identification,
3) Makes progress towards a challenging problem in the literature

**Weaknesses:**

1) Firstly, it is hard to trust the regret bounds presented in the paper without any numerical simulations, and it is also hard to accept the paper in good faith without a limitations section.
2) The assumption of having H>1 could be justified further in line 349. For instance, it is a significant weakening to run the system of multiple trajectories, instead of a single, purely online, trajectory.
3) The paper only studies the model of the LQR cost function (the main focus is more on stability, granted); but how does this extend to more general costs?
4) The exact technical novelty of the paper seems to be unclear. For instance, optimizing while estimating system parameters is not new (The Nonstochastic Control Problem, Hazan et. al., Online Policy Optimization in Unknown Nonlinear Systems, et. al.). Maybe the authors can comment on why this result doesn't follow from the literature?
5) Why aren’t the time intervals $h$ chosen in a dynamic fashion? Surely, this would lead to better bounds? Right now, it seems $h=\frac{1}{15\kappa}$, where $\kappa$ is assumed to be known in advance (this is potentially a very significant assumption since the entire point of the system identification is to estimate A and B).
6) Line 162, would be good to clarify somewhere that $\mathbb{S}_+$ represents the set of symmetric PSD matrices,
7) Line 169, how do we know that the optimal mapping is unique? If it is not unique, then the convergence described in Line 176 is not guaranteed, right?
8) Line 1456 - why is the determinant never 0?
9) Line 283. How small should h be for the estimation to hold?
10) Line 190. There are many notions of regret. It would be good for the authors to clarify that this is the expected regret of the system. In this sense, it seems that the paper overclaims a little at the start.
11) Grammatical issues: line 483. $U^*$ should “minimize” cost, not minimizes. Line 502 - “analyze”, not “analysis”. Line 241. Missing an “in” between “relationship” and “equation"
12) Missing references: Line 100 seems to be missing a reference for Simchowitz et. al, Line 293, 304, 306. Missing reference for Algorithm (hard to tell which algorithm the paper is referring to)

**Questions:**

See weaknesses section.

---

> ### Author Response · Authors · 2024-11-21
> **Comments by Authors**
>
> We thank the reviewer for the comments and constructive suggestions. We **have added relevant experiments and clarified the main novelty and contributions of the paper**. We address the reviewer's questions in more detail as follows:
>
> >Q1: It is hard to trust the regret bounds presented in the paper without any numerical simulations, and it is also hard to accept the paper in good faith without a limitations section.
>
> **A1:** We appreciate the reviewer's suggestions. In response, **we have added an experimental section in Section 5.3 of the paper**, comparing our algorithm with the baseline algorithm. Please refer to the latest PDF version for details.
>
>
> >Q2: The assumption of having $H>1$ could be justified further in line 349. For instance, it is a significant weakening to run the system of multiple trajectories, instead of a single, purely online, trajectory.
>
> **A2:** Our primary focus is on the system identification problem when a stabilizer $K$ is known beforehand. Running multiple trajectories is not our main concern; it serves merely as a tool to estimate the stabilizer $K$ in cases where no prior information about it is available.
>
> >Q3: The paper only studies the model of the LQR cost function, but how does this extend to more general costs?
>
> **A3:** We thank the reviewer's comment. Our primary focus is on system identification. The LQR cost function serves as an example to demonstrate that our system identification method is applicable to online control and performs well under the finite observation assumption. For general costs, a explore-then-exploit strategy can still be applied, and we consider this a promising direction for future work.
>
>
> >Q4: The exact technical novelty of the paper seems to be unclear.
>
> **A4:** We appreciate the reviewer's feedback and provide a clearer explanation of the contributions. The technical novelty lies in **finding the exact reverse $Ah$ of the matrix exponential $e^{Ah}$**. Two key novel ideas address this challenge. First, we identify a domain $\|X\|\leq \frac{1}{4}\$ where $f(X)=e^{X}$ is one-to-one, and the distance between $f(X_1)$ and $f(X_2)$ is lower bounded by $\frac{1}{2}\|X_1-X_2\|$, as shown in Lemma 7 in our paper. This property enables the analysis of estimation errors. Second, we apply Taylor expansion to compute the estimator. Detailed descriptions are highlighted in blue in Section 4.
>
>
> >Q5: Why aren’t the time intervals chosen in a dynamic fashion? Surely, this would lead to better bounds?
>
> **A5:** We appreciate the reviewer’s question. We also considered using dynamic time intervals during our analysis. We find that changing the time interval gap does not significantly improve performance. As stated in Theorem 2 regarding the lower bound, the estimation error **is determined by the total running time $T$, rather than the interval gap**.
>
> >Q6: Line 162, would be good to clarify somewhere that $\mathcal{S}_+$ represents the set of symmetric PSD matrices,
>
> **A6:** Thank you for pointing this out. We have updated the text to clarify that $\mathcal{S}_+$ represents the set of symmetric positive semi-definite matrices.
>
> >Q7: How do we know that the optimal mapping is unique in line 169? If it is not unique, then the convergence described in Line 176 is not guaranteed?
>
> **A7:** Thank you for the question. The uniqueness of the optimal mapping in line 169 is guaranteed by the assumptions on $A,B,Q,R$, as detailed in [1]. The convergence described in line 176, however, **relies on the properties of the Riccati function** rather than the uniqueness of the optimal mapping.
>
> >Q8: Why is the determinant in line 1456 never becomes 0?
>
> **A8:** Thank you for raising this question. We ensure that $\lambda\leq\frac{1}{4\|\Sigma\|}$, which guarantees that for any nonzero $x$, $$\|(I-2\lambda\Sigma)x\|\geq \|x\|-\|2\lambda\Sigma x\|\geq \|x\|-\frac{1}{2}\|x\|>0.$$
>
> Thus, the determinant of $I -\lambda \Sigma$ never becomes zero.
>
> >Q9: How small should $h$ in line 283 be for the estimation to hold?
>
> **A9:** Thank you for the question. The value of $h$ can follow $\frac{1}{15\kappa_{A}}$ as specified in Assumption 1.
>
> >Q10: There are many notions of regret in line 190. It would be good for the authors to clarify that this is the expected regret of the system.
>
> **A10:** Thank you for pointing this out. We have clarified the statement in the updated version to specify that this refers to the expected regret of the system.
>
> >Q11: Grammatical issues and missing reference.
>
> **A11:** We thank the reviewer for pointing out the mistakes. These have been corrected in the updated version, with changes highlighted in blue.
>
> We thank the reviewer once again for the valuable and helpful suggestions. We would be happy to provide further clarifications if the reviewer has any additional questions.
>
>
> **References**
>
> [1] Yong, Jiongmin, and Xun Yu Zhou. Stochastic controls: Hamiltonian systems and HJB equations. Vol. 43. Springer Science & Business Media, 2012.

---

> ### Comment · Reviewer_isKP · 2024-11-21
> **Response to the authors**
>
> I thank the authors for their detailed reply.
>
> Regarding the experiments, I have a couple of questions:
> 1) Why does the estimation error increase so rapidly in Figure 2?
> 2) Since this is a 3x3 matrix, each element between [-1,1], the Frobenius norm of this matrix is always bounded (from above) by sqrt(9*1^2) = 3, but this is the same (worst case) error the model gets even after seeing 500 trajectories (Figure 2). It seems that the model requires around 2000 trajectories to get an estimation error below 1.0. Can the authors comment on why this is the case? In general, some discussion about these results would be great.
> 3) 2000 trajectories is a lot. As the authors mention in their rebuttals, they are not concerned with the amount of trajectories required... however, I fear that this large number makes the algorithm intractable for actual applications.
>
> Also, as I asked in my initial review:
>  $\kappa$ is assumed to be known in advance (this is potentially a very significant assumption since the entire point of the system identification is to estimate A and B). I would certainly appreciate it if the authors could comment on this.
>
> I wonder if "the technical innovation" the authors describe is novel. Upon looking at the proof of Lemma 7, the proof seems straightforward. Based on point 3 in the author's reply, this seems to be a result focused more on stability, rather than optimization. This makes me wonder if the results are as novel as the authors claim, since many works in the control literature work in continuous-time settings. I also feel that this result should be compared with the nonstochastic optimization and policy optimization results.
>
> Based on the above, I tentatively lower my score to a 3.

---

> > ### Author Response · Authors · 2024-11-22
> >
> > We sincerely appreciate the reviewer for the feedback to our paper. We would like to clarify several points:
> >
> > First, regarding the experiments, we evaluated all three algorithms presented in the paper. However, the main contributions of our work lie in **Algorithm 1 and Algorithm 3**. We acknowledge that Algorithm 2’s contribution is relatively weaker compared to the other two. This result primarily represents a straightforward application of the system identification method. The primary goal of proposing this algorithm is to **provide an efficient approach for obtaining a stabilizer**, rather than providing an optimal practical method for system identification.
> >
> > The increase in the estimation error during the initial stage occurs because the algorithm has not yet converged. The reported values represent **the square of the Frobenius norm** rather than the norm itself, which can exceed 3. This is explicitly explained in the paper. Additionally, the parameter $h$ in our experiments was chosen conservatively ($h = \frac{1}{30}$) and is not optimal, requiring multiple trajectories for convergence. A larger $h$ would lead to faster convergence.
> >
> > In fact, we do not rely heavily on a precise value of $\kappa$. It serves merely as an upper bound for system identification and does not need to be particularly close to $\|A\|$, as clarified in Assumption 1 of the paper. By selecting $h$ sufficiently small, we can conservatively ensure it remains less than $\frac{1}{15\kappa}$. We acknowledge that an imprecise choice of $h$ can lead to slower algorithm convergence, as noted by the reviewer. However, this limitation primarily affects the optimality of Algorithm 2 and does not impact Algorithm 1 or Algorithm 3 seriously.
> >
> > The novelty of our work does not lie in the analysis of the matrix inverse itself but in leveraging this perspective to transform the identification problem of continuous systems into one for discrete systems, followed by detailed theoretical analysis. Under the single-trajectory setup, we identify the upper and lower bounds of the problem, demonstrating optimization rather than simply achieving stabilization. While constant bounds were less emphasized (e.g., $h$ was chosen conservatively for identification), the main term with respect to time $T$ achieves optimality.
> >
> > Finally, the nonstochastic optimization of online control we consider is a problem with weaker assumptions on system noise. Current research of nonstochastic noise **has only achieved sublinear regret for discrete-time systems[1][2], and not for continuous-time systems**. We consider this a promising direction for future work.
> >
> >
> > We thank the reviewer once again for the suggestions. We would be happy to provide further clarifications if the reviewer has any additional questions.
> >
> >
> >
> > **References**
> >
> > [1] Agarwal, Naman, et al. "Online control with adversarial disturbances." International Conference on Machine Learning. PMLR, 2019.
> >
> > [2] Hazan, Elad, Sham Kakade, and Karan Singh. "The nonstochastic control problem." Algorithmic Learning Theory. PMLR, 2020.

---

> > > ### Comment · Reviewer_isKP · 2024-11-23
> > > **Reply to the Authors**
> > >
> > > I thank the authors for their detailed reply about the experimental figure and recommend adding this explanation about the initial increase in the Frobenius error to the manuscript.
> > >
> > > However, if the goal is to perform system identification and obtain a stabilizer, I still wonder if having any knowledge of $\kappa$ is a reasonable assumption... If I knew nothing about the system, a priori, $\kappa$ would be very large, which means that the sampling intervals $h$ are small. But then, from Lemma 4, this implies that the error is still substantial, suggesting that the algorithm intuitively is not doing much learning and has an over-reliance on $\kappa$ - after all, the final error bounds are written (basically) only in terms of $\kappa$...
> > >
> > > I wonder if I am not understanding something here, and would greatly appreciate a clarification from the authors. Thanks!

---

> > > > ### Author Response · Authors · 2024-11-24
> > > >
> > > > We appreciate the reviewer’s thoughtful feedback. We would like to clarify two points:
> > > >
> > > > 1. In practical system identification tasks, it is generally possible to estimate $\kappa$ based on physical properties or prior knowledge of the system. This is a standard assumption in many identification scenarios. Previous studies on continuous-time system identification have also made assumptions regarding system parameters. For instance, [1] includes assumption H.1, and [2] assumes that the initial estimation of system parameters $(A,B)$ satisfies specific error bounds with respect to the true parameters $(A^*, B^*)$, as stated in Equation 7.
> > > > 2. Without any knowledge of $\kappa$, it is fundamentally impossible to identify certain systems with finite samples. For example, consider the rotation matrix $$ e^{Ah} = \begin{pmatrix} \cos(ah) & \sin(ah) \\ -\sin(ah) & \cos(ah) \end{pmatrix}. $$ It can be seen from Equation 8 in our paper, with any finite observation set, it is impossible to distinguish between a system with a very small $a$ and one with a sufficiently large $a$, as the mapping is not injective.
> > > >
> > > > We hope this addresses the concern and clarifies the rationale behind the assumption. Thank you!
> > > >
> > > > **References**
> > > >
> > > > [1] Basei, Matteo, et al. "Logarithmic regret for episodic continuous-time linear-quadratic reinforcement learning over a finite-time horizon." Journal of Machine Learning Research 23.178 (2022): 1-34.
> > > >
> > > > [2] Faradonbeh, Mohamad Kazem Shirani, and Mohamad Sadegh Shirani Faradonbeh. "Online Reinforcement Learning in Stochastic Continuous-Time Systems." The Thirty Sixth Annual Conference on Learning Theory. PMLR, 2023.

---

> > > > > ### Comment · Reviewer_isKP · 2024-11-24
> > > > > **Reply to the Authors**
> > > > >
> > > > > I thank the authors for their reply and acknowledge now that $\kappa$ must be known to identify certain systems. I think it would be very illuminating for future readers if this example with the rotation matrix (or any other example that conveyed the necessity of this assumption) were to be included in the manuscript.
> > > > >
> > > > > Based on the authors' reply, I happily return my score to a 5; where I hesitate to increase the score further as I am uncertain about the novelty of the analysis/algorithm concerning what has been done previously in the control theory literature.

---

> > > > > > ### Author Response · Authors · 2024-11-24
> > > > > >
> > > > > > We sincerely thank the reviewer for their patience and insightful discussions throughout the review process! These discussions have been extremely valuable, and we will incorporate the suggested example and other improvements into the next version of the manuscript to enhance clarity and accessibility.
> > > > > >
> > > > > > Regarding the novelty of our analysis and algorithm compared to prior work in the control theory literature, we emphasize the following key distinctions:
> > > > > >
> > > > > > In previous approaches to continuous-time system identification[1][2], **discretization has often been handled through approximations**(see Section 5.1 of [1]):
> > > > > > $$X_{t+h} \approx (I + hA)X_t + hBU_t + (W_{t+h} - W_t). $$ However, this approximation introduces a discretization error between the approximated and true dynamics. The discretization error term, characterized by $(e^{hA} - I)/h - A$, is of order $O(h)$. Consequently, the sampling interval $h$ must be chosen as $O(\frac{1}{\sqrt{T}})$ to ensure that **the discretization error does not dominate**, leading to a super-linear sampling complexity, with the sampling number $m = T/h = \Omega(T^{3/2})$, which significantly **increases computational demands**. This problem is discussed in Section 4.1 of our manuscript.
> > > > > >
> > > > > > In contrast, our proposed method **eliminates the need for such approximations by leveraging an identity transformation rather than an approximate one**(see Eq. 8 in our manuscript). As a result, our approach avoids the discretization error entirely, allowing the sampling interval $h = \frac{1}{15\kappa_A}$ to **depend solely on system parameters rather than the total sampling time $T$**. This innovation reduces the sampling complexity to grow linearly with $T$, offering significant computational advantages.
> > > > > >
> > > > > > We hope this clarification addresses the reviewer’s concerns and highlights the novelty and contribution of our work. Thank you again for the valuable feedback.
> > > > > >
> > > > > > **References**
> > > > > >
> > > > > > [1] Faradonbeh, Mohamad Kazem Shirani, and Mohamad Sadegh Shirani Faradonbeh. "Online Reinforcement Learning in Stochastic Continuous-Time Systems." The Thirty Sixth Annual Conference on Learning Theory. PMLR, 2023.
> > > > > >
> > > > > > [2] Shirani Faradonbeh, Mohamad Kazem, Mohamad Sadegh Shirani Faradonbeh, and Mohsen Bayati. "Thompson sampling efficiently learns to control diffusion processes." Advances in Neural Information Processing Systems 35 (2022): 3871-3884.

---

### Official Review · Reviewer_TTf1 · 2024-11-06

**Soundness:** 3
**Presentation:** 3
**Contribution:** 3
**Rating:** 5
**Confidence:** 4

**Summary:**

This paper considers a continuous time linear control system and proposes a system identification method with a convergence rate guarantee. A lower bound is also developed to demonstrate the optimality of the upper bound. The system identification method is then applied to an online control method to obtain a regret bound.

**Strengths:**

1. The identification of continuous time linear system is not fully studied in the statistical learning setting, especially the convergence rate analysis.
2. The paper uses two approximation of A, B to construct the estimator, which is quite novel.
3. The paper is well written.

**Weaknesses:**

1. My major concern is the novelty and significance. The system ID of linear systems is very well studied for discrete-time systems. The extension to continuous time systems is interesting, but I don't see significant technical difficulty. Besides, most classical control literature considers continuous time linear systems already in the past, e.g. [W1]. The authors should provide a better summary of the technical novelty for their convergence rate analysis.

2. The lower bound and the regret analysis also seem to be straightforward extensions of the existing results for the discrete time systems in [W2]. The authors should provide a better summary of the technical novelty for their lower bound analysis.


[W1] Kopp, R.E. and Orford, R.J., 1963. Linear regression applied to system identification for adaptive control systems. Aiaa Journal, 1(10), pp.2300-2306.

[W2] Simchowitz, M., Mania, H., Tu, S., Jordan, M.I. and Recht, B., 2018, July. Learning without mixing: Towards a sharp analysis of linear system identification. In Conference On Learning Theory (pp. 439-473). PMLR.

**Questions:**

1. This paper mentions that the one major motivation is practicability: most linear systems in practice are continuous time. But does this paper consider other practical issues, such as uneven time sampling? Does the algorithm still work if the time discretization is also noisy and uneven?

2. Another practical issue is the imperfect state observation, i.e. the observation is $x_t+\epsilon_t$. This is easier than output feedback but this is also a common issue in real-world applications. How can the proposed algorithm be generalized to handle imperfect state observations?

---

> ### Author Response · Authors · 2024-11-21
> **Comments by Authors**
>
> We sincerely appreciate the reviewer's insightful comments and suggestions. Below, we provide responses to the raised questions.
>
> >Q1: Concern of the novelty and significance. The system ID of linear systems is very well studied for discrete-time systems. The extension to continuous time systems is interesting, but I don't see significant technical difficulty. Besides, most classical control literature [1] considers continuous time linear systems already in the past. The authors should provide a better summary of the technical novelty for their convergence rate analysis.
>
> **A1:** We appreciate the reviewer's feedback and provide a clearer explanation of the contributions. The technical novelty lies in **finding the exact reverse $Ah$ of the matrix exponential $e^{Ah}$**. Two key ideas address this challenge. First, we identify a domain $\|X\|\leq \frac{1}{4}\$ where $f(X)=e^{X}$ is one-to-one, and the distance between $f(X_1)$ and $f(X_2)$ is lower bounded by $\frac{1}{2}\|X_1-X_2\|$, as shown in Lemma 7 in our paper. This property enables the analysis of estimation errors. Second, we apply Taylor expansion to compute the estimator. Detailed descriptions are highlighted in blue in Section 4.
>
>
> >Q2: The lower bound and the regret analysis also seem to be straightforward extensions of the existing results for the discrete time systems in [2]. The authors should provide a better summary of the technical novelty for their lower bound analysis.
>
> **A2:** We thank for the reviewer's suggestion. The key novelty lies in **analyzing the discretized transformation law**, as shown in Equation 7 in our paper, and conditioning on any finite set of observed times. While we acknowledge that our approach shares similarities with prior works like [2] in addressing this well-described problem, our method introduces a new perspective tailored to continuous-time systems.
>
>
> >Q3: This paper mentions that the one major motivation is practicability: most linear systems in practice are continuous time. But does this paper consider other practical issues, such as uneven time sampling?
>
> **A3:** We appreciate the reviewer's feedback. We also considered using dynamic time intervals during our analysis. We find that **changing the time interval gap does not significantly improve performance**. As stated in Theorem 2 regarding the lower bound, the estimation error **is determined by the total running time $T$, rather than the interval gap**. This is one of the key differences between discrete and continuous systems.
>
>
>
> >Q4: Another practical issue is the imperfect state observation. This is easier than output feedback but this is also a common issue in real-world applications. How can the proposed algorithm be generalized to handle imperfect state observations?
>
> **A4:** We agree that imperfect state observation is an important scenario to address. If the observed state serves as an unbiased estimator of the true state, it can be directly incorporated into our method. Even when the observation error is stochastic, our system identification approach remains effective. By setting $U_{t}=KX_{t}+u_{k}$ for $t\in[kh,(k+1)h]$, the observed state transitions as:
> $$X_{(k+1)h}=AX_{kh}+BU_{kh}+e_{k}.$$
>
> where $e_{k}$ depends on the observation error and the stochastic noise $W_t$. Since $e_k$ has zero mean, we can apply similar analytical techniques as in our method to handle this case.
>
>
>
>
> Finally, we thank the reviewer once again for the effort in providing us with valuable and helpful suggestions. We will continue to provide clarifications if the reviewer has any further questions.
>
>
> **References**
>
> [1] Kopp, R.E. and Orford, R.J., 1963. Linear regression applied to system identification for adaptive control systems. Aiaa Journal, 1(10), pp.2300-2306.
>
> [2] Simchowitz, M., Mania, H., Tu, S., Jordan, M.I. and Recht, B., 2018, July. Learning without mixing: Towards a sharp analysis of linear system identification. In Conference On Learning Theory (pp. 439-473). PMLR.

---

> ### Author Response · Authors · 2024-12-01
>
> Thanks again for your valuable feedback! Could you please let us know whether your concerns have been addressed? We are happy to make further updates if you have any other questions or suggestions.

---

### Author Response · Authors · 2024-11-21
**General Response**

We sincerely thank all the reviewers for their valuable suggestions. In the updated PDF, we have replaced the proof sketch of the regret bound with experimental results in Subsection 5.3. Other key updates are highlighted in blue.

To address the concern regarding the lack of numerical experiments, we have added three experiments in Subsection 5.3. The first two experiments validate Theorems 1 and 3. The third experiment compares our online control algorithm with the baseline algorithm proposed in [1]. For more details, please refer to Subsection 5.3.

We have also revised the discussions on technical challenges and the novelty of our work in Sections 4 and 5. All these changes are highlighted in blue.

For responses to other questions, please refer to the individual sections.

**References**

[1] Faradonbeh, Mohamad Kazem Shirani, and Mohamad Sadegh Shirani Faradonbeh. "Online Reinforcement Learning in Stochastic Continuous-Time Systems." The Thirty Sixth Annual Conference on Learning Theory. PMLR, 2023.

---

### Author Response · Authors · 2024-11-23

We would like to express our sincere gratitude for the reviewer's constructive suggestions and comments. Since the deadline is approaching, we sincerely hope the reviewers can read our response. Please let us know if the reviewers have any comments about our response or any other additional concerns. We are eager to provide any further clarifications and discussions to help the evaluation.

---

### Meta-Review · Area_Chair_RfAd · 2024-12-22

**Metareview:**

Although I appreciate the authors' rebuttal efforts -- including adding new simulation results -- the reviewers still have lingering concerns on the technical challenges brought forth by the proposed system ID method for the continuous systems. In comparison to the existing control literature, the O(\sqrt{T}) regret is the same, with an improvement of T^{1/2} for sampling complexity. My reading of the cause of reviewers' lukewarm assessment (pre and post rebuttal) is how significant this improvement is. Tied to this assessment is also the perceived straightforwardness of the technical analysis associated with the approach. I suggest the authors rewrite the paper to make a convincing case on why this improvement is not incremental, before submitting to the next venue.

**Additional Comments On Reviewer Discussion:**

NA

---

### Decision · Program_Chairs · 2025-01-22

Reject